# Rational correction of pathogenic conformational defects in HTRA1

Nathalie Beaufort[1,17], Linda Ingendahl[2,17], Melisa Merdanovic[2], Andree Schmidt[3,4], David Podlesainski[2], Tim Richter[2], Thorben Neumann[5], Michael Kuszner[6], Ingrid R. Vetter[7], Patricia Stege[7], Steven G. Burston[8], Anto Filipovic[2], Yasser B. Ruiz-Blanco[2], Kenny Bravo-Rodriguez[2,7], Joel Mieres-Perez[2,9], Christine Beuck[2], Stephan Uebel[10], Monika Zobawa[10], Jasmin Schillinger[2], Rainer Malik[1], Katalin Todorov-Völgyi[1], Juliana Rey[2], Annabell Roberti[2], Birte Hagemeier[2], Benedikt Wefers[3,11], Stephan A. Müller[3,12], Wolfgang Wurst[3,11,13,14], Elsa Sanchez-Garcia[9], Alexander Zimmermann[6], Xiao-Yu Hu[15], Tim Clausen[16], Robert Huber[2,10], Stefan F. Lichtenthaler[3,12,13], Carsten Schmuck[6,18], Michael Giese[5], Markus Kaiser[2], Michael Ehrmann[2,19] ✉ & Martin Dichgans[1,3,13,19] ✉

Loss-of-function mutations in the homotrimeric serine protease HTRA1 cause cerebral vasculopathy. Here, we establish independent approaches to achieve the functional correction of trimer assembly defects. Focusing on the prototypical R274Q mutation, we identify an HTRA1 variant that promotes trimer formation thus restoring enzymatic activity in vitro. Genetic experiments in *Htra1^R274Q* mice further demonstrate that expression of this protein-based corrector in *trans* is sufficient to stabilize HtrA1-R274Q and restore the proteomic signature of the brain vasculature. An alternative approach employs supramolecular chemical ligands that shift the monomer-trimer equilibrium towards proteolytically active trimers. Moreover, we identify a peptidic ligand that activates HTRA1 monomers. Our findings open perspectives for tailored protein repair strategies.

High requirement temperature protein A1 (HTRA1) is a highly conserved PDZ-serine protease. It is implicated in the homeostasis of the extracellular matrix (ECM), and involved in cell signaling, differentiation, and survival[1–3]. Dysregulation of HTRA1 has been linked to various disorders including age-related macular degeneration[4–6], Alzheimer's disease[7–9], arthritis[2,10], and malignancies[1]. Biallelic loss-of-function mutations in *HTRA1* cause cerebral autosomal recessive arteriopathy with subcortical infarcts and leukoencephalopathy (CARASIL), a disabling and fatal condition that manifests with stroke and dementia and is characterized by a sclerotic degeneration of the brain vasculature[11,12]. HTRA1 exists in a dynamic equilibrium between monomers and trimers. Its activity is tightly controlled by an inter-protomer activation cascade enabling a reversible allosteric switch between the resting and active conformations[13,14]. Like other HtrAs, HTRA1 displays a composite

activation domain consisting of loops L1, L2, L3, and LD[1,13–15]. Activation is initiated by an interaction of the sensor loop L3 with the substrate bound to the active site or the PDZ domain[13,16]. Subsequently, the loop L3 interacts with loop LD of an adjacent protomer. This interaction induces the sequential proper positioning of loops L1 and L2 and thus of the catalytic site. Consequently, the multimeric assembly of HTRA1 is indispensable for proteolytic activity[13,14,17]. Notably, a subset of mutations implicated in familial vasculopathy impairs trimerization and thus proteolytic activity of HTRA1[18,19]. Here, we establish various molecular approaches to achieve the conformational and functional correction of these pathogenic variants. In an initial attempt, we take advantage of protein *trans*-complementation. Specifically, we identify an HTRA1 variant that reconstitutes the multimeric assembly and proteolytic activity of a prototypical disease-causing mutant (R274Q)

---

in vitro. In addition, we establish that the genetic delivery of this biologic is sufficient to restore the pathologically altered proteome of brain vessels from *Htra1*<sup>*R274Q*</sup> mice. To further explore the rescue of HTRA1 mutants causing assembly defects, we devise two alternative correction strategies. The first approach relies on supramolecular chemical ligands that shift the equilibrium towards active trimers, while the second approach is based on a natural peptide that activates HTRA1 monomers.

## Results

### Biochemical characterization of pathogenic assembly defects in HTRA1

The majority of pathogenic *HTRA1* mutations are missense mutations targeting the protease domain and disrupting its enzymatic activity[2,11,20]. To devise protein repair strategies, we selected a set of four disease-causing mutations distributed across the protease domain for biochemical characterization (Fig. 1a). While R166H, A173T and R274Q locate at the protomer-protomer interface, A252T locates at a distance from the interface (Fig. 1b). All mutations strongly reduced HTRA1 protease activity compared to wild-type (wt) HTRA1 in cell-free β-casein degradation assays (Fig. 1c and Supplementary Fig. 1a). Molecular dynamics simulations and free energy perturbation calculations predicted that R166H, A173T and R274Q destabilize the HTRA1 trimer (Supplementary Fig. 2). In accord with previous results from the Onodera lab[18,19], all three interface mutants were mostly detected as monomers in size

exclusion chromatography (SEC), SEC multi-angle light scattering (SEC-MALS) and chemical crosslinking, while wt HTRA1 and A252T were trimers (Fig. 1d, Supplementary Fig. 1b, c and Supplementary Fig. 3a). Of note, the retention time of A252T trimers in SEC and SEC-MALS was increased compared to wt HTRA1 trimers (Fig. 1d and Supplementary Fig. 1b, c). While the reason for the different retention times is unknown, the MALS data indicate that the mutant is a trimer. Native mass spectrometry (nMS) experiments, including titration assays, as well as sedimentation velocity analytical ultracentrifugation (AUC) further indicated that in R274Q, R166H, and A173T but not in A252T the monomer-trimer equilibrium was shifted towards monomers (Fig. 1e, Supplementary Fig. 1d, e). The small amounts of trimers account for the residual levels of enzymatic activity detected in interface mutants. A173T exhibited the lowest relative trimer abundance (Supplementary Fig. 1d, e) in accord with its markedly reduced activity compared to R274Q and R166H (Fig. 1c). Given these observations, we hypothesized that restoring the multimeric assembly would rescue protease function.

### Restoration of HTRA1 assembly and activity via protein-based complementation

We initially devised a targeted protein repair strategy focusing on the archetypical interface mutation R274Q, which is one of the few CARASIL-causing mutations identified in more than one pedigree (Supplementary Table 1). In wt HTRA1, R274 forms salt bridges with D174 and E177 on adjacent protomers (Fig. 2a, panel 2'). In

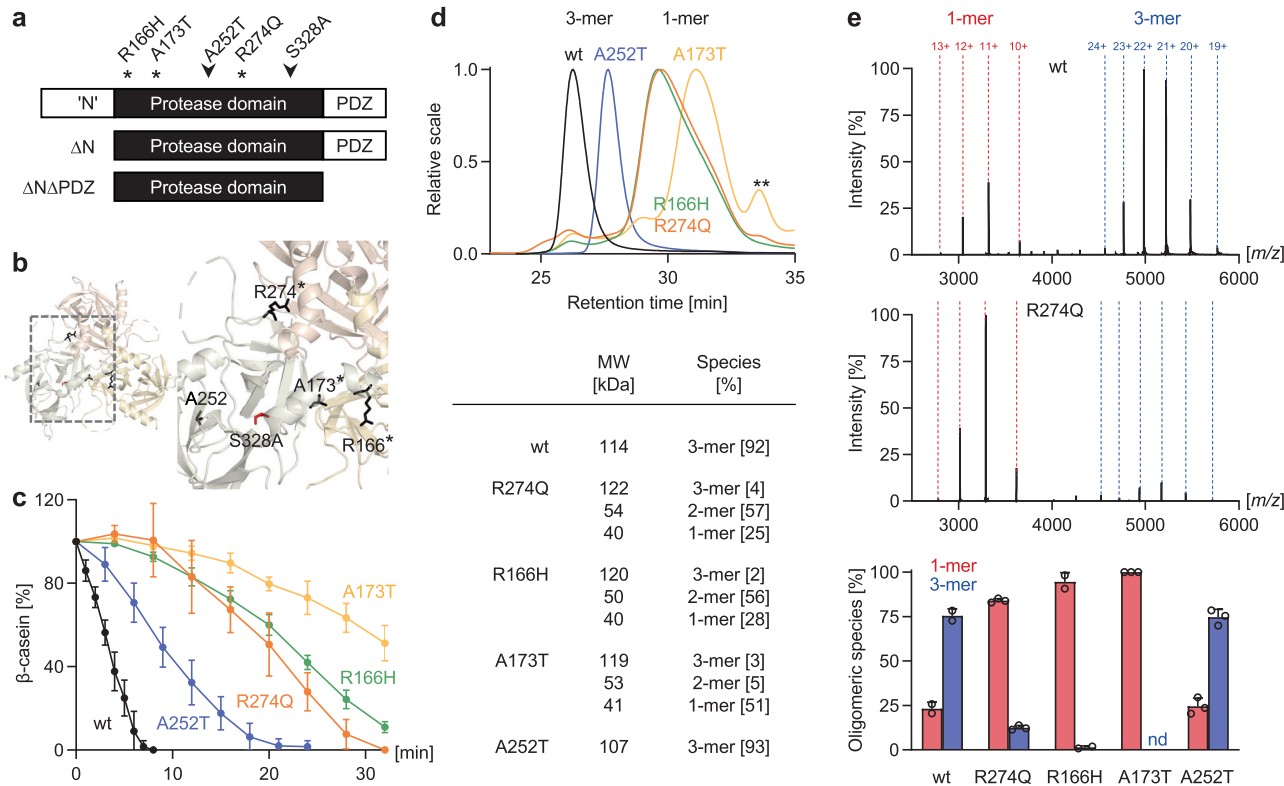

**Fig. 1 | HTRA1 interface mutants exhibit oligomeric assembly defects.**
**a** Schematic representation of HTRA1 domain organization and position of selected pathogenic mutations and the active site mutation S328A. Mutations located at the protomer-protomer interface are marked by an asterisk. The N-terminal ('N') domain consists of a fragmented IGFBP7 domain where neither the incomplete IGFBP binds insulin nor the incomplete Kazal-like domain functions as a protease inhibitor. **b** Position of pathogenic mutations in the HTRA1 trimer (PDB ID 3TJO). **c** Proteolytic activity of wt and mutant HTRA1s (1 μM) using β-casein (20 μM) as a substrate. The graph depicts the relative β-casein signal intensity (average ± SD of 3 independent experimental data). **d** Oligomeric states of wt and mutant HTRA1s

evaluated by SEC-MALS. Upper panel: representative UV chromatograms of wt and mutant HTRA1s depicted in overlay. Lower panel: calculated molecular weights and relative abundance of mono-, di- and trimeric HTRA1 species. **: degradation products or impurities. **e** Oligomeric states of wt and mutant HTRA1s evaluated by native MS. Upper panels: representative spectra of HTRA1 wt and R274Q (5 μM). Mono- and trimeric species and charge states are labeled. Lower panel: relative abundance of mono- and trimeric HTRA1 species (5 μM; nd: not detected; mean + SD of 2 [wt, R166H] or 3 [R274Q, A173T, A252T] independent datasets; empty circles: individual data points). Dimers (relative abundance <5%) are not depicted. Source data are provided as a Source Data file.

accordance with the computational modeling, the crystal structure of R274Q revealed that both interactions are disrupted (Fig. 2a, panel 2"). Owing to the loss of the bidentate ion pair, the interaction between adjacent protomers should be markedly weakened, thus shifting the monomer-trimer equilibrium towards inactive monomers, as observed in our in vitro analyses. To stabilize trimers and achieve functional repair, we designed a corrector HTRA1 variant as follows. First, we screened for a compensatory mutation at the opposite interface of residue 274 by expressing double-mutant proteins bearing R274Q in *cis* with various D174X mutations (Supplementary Fig. 4a). Replacement of Asp174 by Arg, Lys or Trp resulted in a gain of protease activity thus indicating functional *cis*-complementation (Fig. 2b, c and Supplementary Fig. 4a). In contrast, replacement by His, Ser, Tyr, Gly, or Val had little or no influence on HTRA1 activity. We then examined whether complementation in *trans* would suffice to correct the conformation and function of R274Q within mixed trimers (Fig. 2c). To this end, we generated a proteolytically inactive HTRA1 variant (D174R-S328A) bearing the corrector mutation alongside with an active site mutation (i.e., S328A, replacing the catalytic Ser by Ala). Mixing the proteins in various ratios caused a dose-dependent restoration of R274Q activity (Fig. 2b and Supplementary Fig. 5a). In-depth analyses of oligomeric states by nMS, SEC, SEC-MALS, AUC, and crosslinking revealed that the functional repair of R274Q involves the restoration of HTRA1 multimeric assembly within mixed R274Q/D174R-S328A trimers (Fig. 2c–e, Supplementary Fig. 3b and Supplementary Fig. 5b–d). Moreover, D174R-S328A did not interfere with the proteolytic activity of wt HTRA1 following mixed trimer formation (Supplementary Fig. 4b and Supplementary Fig. 3c). The crystal structure of D174R-R274Q further uncovered the molecular basis of conformational repair (Fig. 2a, panel 2"). Specifically, R174 reoriented towards the protomer at the opposite interface (Fig. 2a, panel 2") resulting in an overall increased trimer interface as compared to the single R274Q mutant (buried surface area per protomer: wt [PDB ID 3TJO]: 947 Å$^2$; R274Q [PDB ID 6Z0E]: 908 Å$^2$; D174R-R274Q [PDB ID 6Z0X; 6Z0Y]: 925–957 Å$^2$).

## *Htra1$^{R274Q}$* mice show an altered cerebrovascular proteome

To explore the relevance of our findings in vivo, we generated *Htra1$^{R274Q}$* knock-in mice by introducing an AGA>CAG substitution in the codon of R274 (Fig. 3a). To examine the consequences of R274Q on the vasculature and obtain a detailed readout for rescue experiments, we isolated mouse brain vessels followed by quantitative mass spectrometry (MS) analysis. Overall, we identified >6500 proteins. The levels of 268 proteins were altered in homozygous *Htra1$^{R274Q}$* compared to wild-type mice (Supplementary Fig. 6a). Among the most noticeable abnormalities was a >7.5-fold depletion of HtrA1 protein despite comparable *htra1* mRNA levels in *Htra1$^{R274Q}$* compared to *Htra1$^{wt}$* vessels (Supplementary Fig. 6b, c) indicating defects in folding and stability of HtrA1-R274Q. Of note, unique tryptic peptides matching with the canonical sequence of HtrA1 included an Arg274-containing peptide that was detected in lysates from *Htra1$^{wt}$* vessels while being absent in *Htra1$^{R274Q}$* vessels (Fig. 3b and Supplementary Fig. 7a, b). Expanding the search database with the sequence of HtrA1-R274Q resulted in the identification of a Gln274-containing peptide in brain vessels from *Htra1$^{R274Q}$* (3 out of 5 samples) but not *Htra1$^{wt}$* mice (Fig. 3b and Supplementary Fig. 7a, b). The signal intensity of this peptide was substantially reduced in heterozygous *Htra1$^{wt/R274Q}$* mice compared to homozygous *Htra1$^{R274Q/R274Q}$* mice (Supplementary Fig. 7b). Thus, analyzing this peptide is sufficient to selectively detect and quantify HtrA1-R274Q. We further found a marked accumulation of secreted and ECM-associated proteins overlapping with those enriched in the cerebrovasculature of *Htra1$^{KO}$* mice[3,21] (Supplementary Fig. 6d-e). Mapping of individual tryptic peptides to the respective protein

sequences revealed that Ltbp4 and Prss23, two proteins degraded by HTRA1 in vitro[21,22], displayed domain-specific differences in peptide abundance in *Htra1$^{R274Q}$* compared to *Htra1$^{wt}$* vessels (Supplementary Fig. 7c). Specifically, 7 peptides covering the N-terminal (Nt) region of Ltbp4 showed an >17-fold enrichment in *Htra1$^{R274Q}$* samples based on LFQ values, whereas enrichment of the 75 C-terminal (Ct) peptides was much less pronounced (<1.5-fold) (Fig. 3c) indicating that the Nt region of Ltbp4 is extensively processed in *Htra1$^{wt}$* but not in *Htra1$^{R274Q}$* vessels. Accordingly, immunoblots (IB) indicated an accumulation of high molecular mass species of Ltbp4 in *Htra1$^{R274Q}$* compared to *Htra1$^{wt}$* vessels (Fig. 3d). Moreover, biochemical cleavage assays using purified proteins confirmed processing of LTBP4 by HTRA1 (Fig. 3e). Immunohistochemical (IHC) analysis of mouse brain further showed a significant >1.5-fold increase of Ltbp4 signal intensity within the arterial wall of *Htra1$^{R274Q}$* mice whereas there was no difference in staining for laminin, another constituent of the vascular ECM (Fig. 3f). Analysis of Prss23-derived peptides revealed a > 14-fold enrichment of the Ct region, covered by 11 peptides peptides, but not of the Nt peptide in *Htra1$^{R274Q}$* vessels (Fig. 3c) suggesting proteolytic degradation of the Ct region of Prss23 in *Htra1$^{wt}$* but not in *Htra1$^{R274Q}$* vessels.

Given these observations, we quantified protein abundance in *Htra1$^{R274Q}$* vs *Htra1$^{wt}$* mice as follows: levels of Pan-HtrA1 were determined excluding Arg274-containing peptides; levels of Ltbp4 were quantified separately for the N- and C-terminal regions; levels of Prss23 were quantified by focusing on the C-terminal region (Fig. 3g). This signature was considered as the primary read-out to evaluate the functional correction of HtrA1 in vivo.

## Protein-based functional correction of HtrA1 in vivo

We next set out to rescue HtrA1 function and attenuate the molecular phenotype of the brain vasculature in *Htra1$^{R274Q}$* mice. To achieve long-term expression of the HtrA1 corrector (D174R-S328A) at relevant sites and at sufficient levels, we used a genetic approach. Specifically, we generated double mutant mice expressing D174R-S328A at the endogenous locus (Fig. 4a). Overall, the consequences of D174R-S328A on the cerebrovascular proteome resembled those of R274Q (Fig. 4b and Supplementary Fig. 8). In fact, *Htra1$^{D174R-S328A}$* vessels exhibited an even stronger accumulation of key proteins (Fig. 4b) consistent with D174R-S328A being completely inactive whereas R274Q displayed residual protease activity as demonstrated in vitro (Fig. 2b). Next, we crossed *Htra1$^{R274Q}$* with *Htra1$^{D174R-S328A}$* mice to obtain heterozygous animals with the pathogenic and corrector *Htra1* alleles in *trans* (Fig. 4c). Of note, the proteomic phenotype of heterozygous *Htra1$^{wt/R274Q}$* mice was less pronounced than the phenotype seen in homozygous *Htra1$^{R274Q/R274Q}$* animals (Supplementary Fig. 9). Hence, we postulated that the functional correction of a single R274Q allele in *Htra1$^{R274Q/D174R-S328A}$* mice would be sufficient to rescue the proteomic phenotype. Indeed, the key abnormalities detected in the vasculature of *Htra1$^{R274Q}$* mice were all attenuated in *Htra1$^{R274Q/D174R-S328A}$* brain vessels (Fig. 4d and Supplementary Fig. 10). Specifically, we found a > 11-fold increase in the levels of Pan-HtrA1 in *Htra1$^{R274Q/D174R-S328A}$* compared to *Htra1$^{R274Q}$* mice (Fig. 4d). Moreover, the signal intensity of the isoform specific Gln274-containing peptide was substantially increased in brain vessels of *Htra1$^{R274Q/D174R-S328A}$* mice compared to both *Htra1$^{wt/R274Q}$* (>4.5-fold) and *Htra1$^{R274Q/R274Q}$* (>3-fold) mice (Fig. 4d). These observations point to stabilization of HtrA1-R274Q in the presence of D174R-S328A. In addition, we found a significant reduction in the levels of both Ltbp4-Nt (>3.2-fold) and Prss23-Ct (>2.4-fold) in *Htra1$^{R274Q/D174R-S328A}$* compared to *Htra1$^{R274Q}$* mice (Fig. 4d). Also, immunoblots demonstrated that the proteolytic processing of Ltbp4, which was impaired in *Htra1$^{R274Q}$* and *Htra1$^{D174R-S328A}$* vessels, was restored in *Htra1$^{R274Q/D174R-S328A}$* vessels (Fig. 4e). IHC confirmed a significant reduction of Ltbp4 signal intensity in the vasculature of *Htra1$^{R274Q/D174R-S328A}$* compared to *Htra1$^{R274Q}$* mice (Fig. 4f). We further applied principal component analysis and found

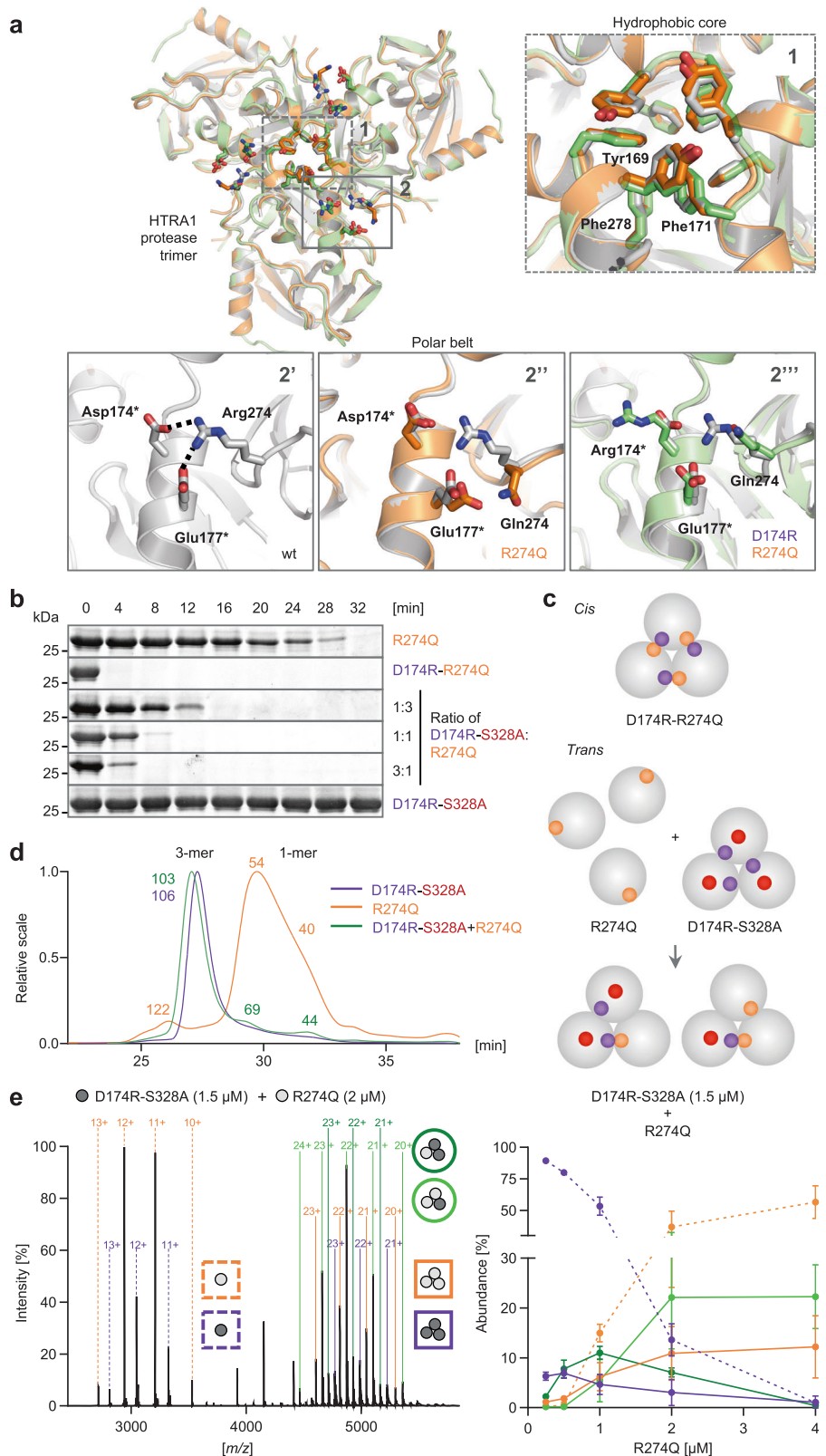

the protein abundance profile of *Htra1^R274Q* and *Htra1^D174R-S328A* vessels to segregate from the profile of *Htra1^wt* vessels (Fig. 4g). Proteome changes in *Htra1^R274Q/D174R-S328A* vessels exhibited a distinct profile which overlapped with the profile of Htra1^wt/D174R-S328A vessels, indicating functional rescue of the single *Htra1^R274Q* allele. Collectively, our findings demonstrate that HtrA1-D174R-S328A is a potent corrector of HtrA1-R274Q in vivo.

## HTRA1 assembly and activity in the presence of supramolecular ligands

To devise an alternative strategy that would target multiple HTRA1 interface mutants, we reasoned that supramolecular ligands exposing positive charges on either end might bind anionic hotspots on adjacent protomers thus stabilizing trimers. For this purpose, guanidinio-carbonyl pyrroles (GCP), synthetic arginine analogs that bind

**Fig. 2 | Restoration of HTRA1 assembly and activity via protein-based complementation. a** Structures of wt HTRA1 (gray; PDB ID 3TJO), R274Q (orange; PDB ID 6Z0E), and D174R-R274Q (green; PDB ID 6Z0X; 6Z0Y) in S328A background. Mutants are shown in an overlay with wt. Panel 1: similar to wt HTRA1, R274Q and D174R-R274Q were crystallized in their trimeric state, superimposing well as seen for the aligned aromatic residues of the central hydrophobic core. Panels 2′–2‴: salt bridges in the HTRA1 protomer interface. Asterisks denote residues on adjacent protomers. The side chains at positions 174, 177, and 274 were well defined by electron density allowing their direct comparison. **b** Proteolytic activity (β-casein degradation) of D174R-R274Q (1 μM), and of D174R-S328A (0.33–3 μM) and R274Q (1 μM) mixed at increasing molar ratios. The activity of R274Q (1 μM) and D174R-S328A (1 μM) serve as controls. Data are representative of 2 independent experiments. **c** Cartoons of *cis-* and *trans*-complementation. D174R-R274Q assembles as an active homotrimer (complementation in *cis*); R274Q (monomeric, inactive) and D174R-S328A (trimeric, inactive) assemble as proteolytically active heterotrimers (complementation in *trans*). **d, e** Oligomeric states of R274Q and D174R-S328A alone or in combination. **d** Representative SEC-MALS UV chromatograms of R274Q, D174R-S328A and D174R-S328A + R274Q (mixed at a 1:1 ratio) depicted in overlay. The calculated molecular weight of mono-, di- and trimeric HTRA1 species is indicated. **e** Left panel: representative native MS spectrum of D174R-S328A (1.5 μM) mixed with R274Q (2 μM). Mono- and trimeric species and charge states are labeled. Right panel: titration of D174R-S328A (1.5 μM) with R274Q. R274Q dimers (<9%) are not depicted. The mean abundance of oligomeric species is depicted and error bars indicate SD ($n$ = 3 independent datasets). Source data are provided as a Source Data file.

oxyanions[23], were connected by flexible linkers (Fig. 5a). Compounds MK1 and 2 carrying two GCP groups connected to a central tris(2-aminoethyl)amine core via peptidic linkers of different sizes yielded a 2 to 2.8-fold increase in proteolytic activity of R274Q, R166H and A173T in biochemical assays (Fig. 5b). Wt HTRA1 was mildly affected, if at all (up to 1.2-fold change), while A252T was inhibited by both compounds. We suspect that MKs stabilize an inactive conformation of HTRA1-A252T by an as yet unknown mechanism. As UV-based protein detection or chemical crosslinking are not feasible for GCPs, the assembly of HTRA1 was followed via NMR diffusion ordered spectroscopy (DOSY) experiments[24]. Stejskal-Tanner plots indicated that for R274Q, a complete shift from trimer to monomer occurs at 25 μM, while the addition of MK2 stabilized trimers, demonstrating that gain in activity correlates with restored trimer assembly (Fig. 5c). In addition, SDS-PAGE of SEC fractions revealed that the elution profile of R274Q was shifted towards trimers in the presence of MK1 (Supplementary Fig. 11a, b). Of note, related ligands with only one GCP unit and an unpolar sterically demanding side arm (AZ21, AZ25) or based on a rigidified peptide backbone (HXY23) appeared inefficient with respect to HTRA1 activation (Supplementary Fig. 11c, d). To evaluate the contribution of polyvalency, we tested a compound in which one GCP unit was removed, while keeping the number of charges constant (TNMK09) and its derivative harboring a non-charged group at the non-GCP end (TNMK27) (Fig. 5a). These compounds caused lower levels of activation of trimer interface mutants (Fig. 5b, right panel). Each specific point mutation is likely to cause structural changes beyond trimer formation, which could explain the non-uniform effects of TNMK09 and TNMK27 on individual interface mutants. We therefore hypothesize that binding of GCPs to as yet unidentified one or more binding site(s) triggers structural rearrangements ultimately causing the stabilization of trimers.

## Allosteric activation of monomeric HTRA1 by peptidic modulators

An alternative and independent strategy for rescuing the activity of interface mutants consists of activating monomeric HTRA1 (Fig. 6a). Monomers are proteolytically inactive because the activation domain is incomplete as loop L3 does not reach in from an adjacent protomer. Therefore, we hypothesized that peptides shifting the activation domain into an active conformation might serve as repair factors. This concept was addressed by identifying suitable activating peptides, validation of the monomeric state, determining enzyme kinetics and presenting an initial model of mechanism.

Screening of an in-house peptide library consisting of the C-termini of human proteins resulted in two peptides representing the C-termini of the Voltage Dependent Anion-selective Channel (VDAC) isoforms 2 and 3 that activated wt HTRA1, R274Q, R166H, and A173T in Tau degradation assays employing purified proteins (Supplementary Fig. 12a). In contrast, peptides that strongly activate wt HTRA1 by binding to the PDZ domain[16] had no effect on the activity of the R274Q mutant. The same effect was observed for a peptide (CAPN2.1) that activates wt HTRA1 in a PDZ-independent manner[16] (Supplementary Fig. 12a). These data suggested a distinct mode of activation of monomeric HTRA1 variants. Moreover, HTRA1 and R274Q lacking the PDZ domain were activated by VDAC2 and 3 peptides, confirming that these peptides bind to the protease domain (Supplementary Fig. 12a).

To characterize the oligomeric state of HTRA1 and its interaction with VDAC2, we carried out nMS titration assays (Fig. 6b and Supplementary Fig. 13a, b). These experiments confirmed the binding of VDAC2 peptide to wt and mutant HTRA1 and revealed an occupancy of one per protomer. In addition, active monomers were confirmed because VDAC2 peptide-induced trimers of R274Q, R166H or A173T were not detected in 3 independent experimental approaches i.e., nMS, chemical crosslinking and NMR DOSY (Fig. 6c, Supplementary Fig. 3d, Supplementary Fig. 13a).

For kinetic studies, an 11-mer peptidic fluorescence-quenched substrate replaced protein substrates that can be difficult to assay consistently due to their conformational heterogeneity. The ability of VDAC2 peptide to activate HTRA1 was initially characterized by titration assays at a fixed concentration of 2 μM substrate (Fig. 6d, Supplementary Fig. 14a). For wt HTRA1, non-cooperative binding of VDAC2 peptide with a $K_d$ = 284 nM was observed, resulting in an up to 4-fold increase in HTRA1 activity. The affinity of R274Q and R166H for VDAC2 peptide was lower ($K_d$ = 12 μM for R274Q; $K_d$ = 6 μM for R166H) but the peptide resulted in a significant activation by up to 5 and 12-fold for R274Q and R166H, respectively. Titration of A173T with VDAC2 peptide showed a lower activation (up to 2-2.4 fold) with 10-fold weaker binding ($K_d$ = 2 μM) compared to the wt protein. To determine whether VDAC2 peptide acts as an allosteric activator, we titrated the substrate in the absence or presence of peptide (Supplementary Fig. 14b). In both conditions, the activity of wt HTRA1 increased up to 5–6 μM substrate before decreasing at higher concentrations. Although the precise mechanism of this inhibition is unknown, it was assumed to be the result of substrate inhibition affecting the allosteric communication within the HTRA1 trimer. The presence of VDAC2 peptide caused the binding of the substrate to be tighter ($K_m$ decreased from 4.7 to 2.2 μM) as well as more cooperative (Hill constant $n$ increased from 1.7 to 2.5). In the case of R274Q and R166H, a hyperbolic dependence on substrate concentration was observed in the presence and absence of VDAC2 peptide, while VDAC2 peptide resulted in a shift to a tighter substrate binding, indicative of allosteric activation. Substrate titration with A173T was hyperbolic but in the presence of VDAC2 peptide, a marked activation of the protease without any apparent tightening of A173T affinity for the substrate was observed. Since some substrate inhibition was observed in the presence of peptide, the data were fitted to a steady-state model including non-cooperative substrate inhibition. Together these data demonstrate the ability of VDAC2 peptide to allosterically activate wt and mutant HTRA1.

Given the composition and sequence specificity of the activation domain, a certain degree of selectivity of the VDAC2 peptide would be expected. The corresponding assays indicated that indeed, HTRA2 and other serine proteases sharing the chymotrypsin-fold (trypsin,

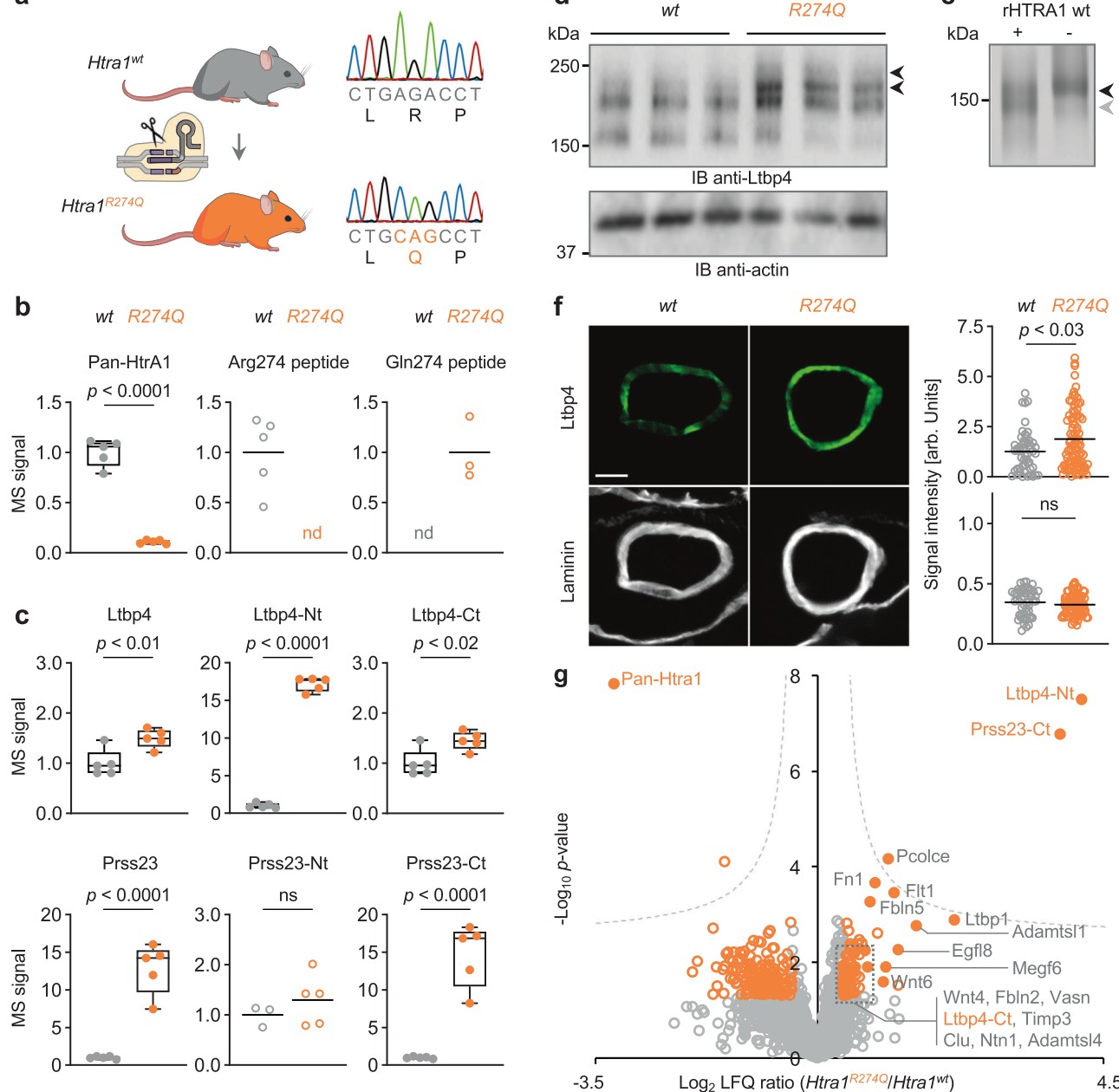

**Fig. 3 | Htra1^R274Q mice show an altered cerebrovascular proteome. a** Generation of *Htra1^R274Q* mice by CRISPR/Cas9-mediated genome editing. Created with BioRender.com released under a Creative Commons Attribution-NonCommercial-NoDerivs 4.0 International license https://creativecommons.org/licenses/by-nc-nd/4.0/deed.en. Isoform-specific detection of HtrA1 (**b**) and region-specific analysis of Ltbp4 and Prss23 (**c**) by mass spectrometry (MS). LFQ values as box-and-whisker plots (centerline: median; limits: 25th and 75th percentile; whiskers: minimum and maximum) with filled circles. MS peptide intensity as scatter plots with empty circles (black bars: mean; nd: not detected). Data points represent individual mice (n = 5 per genotype). The mean signal intensity in *Htra1^wt* or *Htra1^R274Q* samples was set to 1 as appropriate. Significance was tested by two-sided unpaired *t* test (Pan-HtrA1: *p* = 4.2E-07; Ltbp4: *p* = 9.6E-03; Ltbp4-Nt: *p* = 3.0E-10; Ltbp4-Ct: *p* = 1.5E-02; Prss23: *p* = 4.8E-05; Prss23-Nt: *p* = 3.8E-01; Prss23-Ct: *p* = 8.4E-05). **d** Detection of Ltbp4 by immunoblot (IB) in brain vessels from n = 3 mice per genotype. Actin serves as loading control. The multiple Ltbp4 bands most likely account for post-translational modifications and splice variants. Black arrowheads: Ltbp4 species enriched in *Htra1^R274Q* compared to *Htra1^wt* vessels. **e** Processing of LTBP4 by wt HTRA1 in vitro for comparison (purified proteins; SDS-PAGE and silver staining). Black and gray arrowheads: intact and cleaved LTBP4, respectively. Data are representative of 2 independent experiments. **f** Detection of Ltbp4 and laminin by immunohistochemistry (IHC) in brain arteries. Left panels: representative images (scale bar: 25 μm). Right panels: quantification of fluorescence signal intensity as scatter plots (black bars: mean). Data points represent individual vessels (wt: n = 52 arteries from 3 mice; R274Q: n = 87 arteries from 4 mice). Significance was tested by two-sided unpaired Mann-Whitney *U*-test (Ltbp4: *p* = 2.6E-02; laminin: *p* = 2.1E-01). **g** Volcano plot of all proteins quantified by MS. Orange circles: deregulated proteins; filled circles: proteins of interest (see text for explanations); hyperbolic curve: permutation-based FDR estimation. Proteins are labeled with gene names. Source data are provided as a Source Data file.

chymotrypsin and elastase) were not affected, while HTRA3 was slightly inhibited at and above peptide concentrations of 10 μM (Supplementary Fig. 12b). To initially address the mechanism of activation, co-crystallization of HTRA1 and bound peptide was attempted but failed. We, therefore, switched to bioinformatic modeling. Blind docking analyses followed by molecular dynamics simulations suggested two potential binding sites, the active site (a substrate-like binding mode) and an alternative site situated near the trimer tip

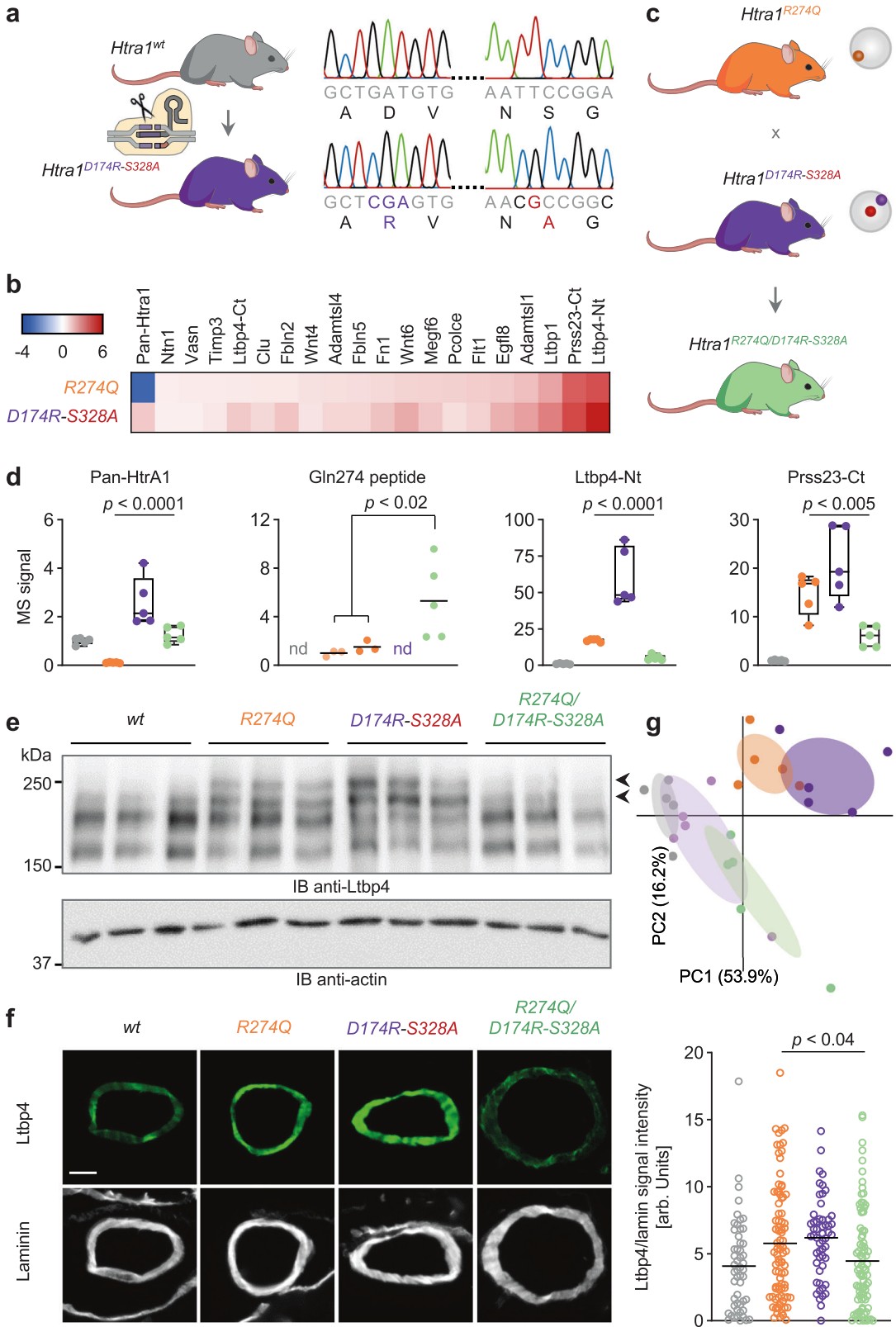

(Supplementary Fig. 15a–c). To distinguish between these modes of action, we tested a VDAC2 derivative (VDAC2-2) which was rationally optimized for active site-binding[16]. This peptide fully activated wt HTRA1 but not R274Q (Supplementary Fig. 15d), supporting the model that functional restoration of monomeric HTRA1 variants requires binding to the alternative site. Here, computational modeling proposes that the VDAC2 peptide interacts simultaneously with an N-terminal helix located at the far tip of the protease domain and the activation loop LD (Fig. 6e, f and Supplementary Fig. 15e). Modeling also suggested that the VDAC2 peptide's P2-Val residue (Fig. 6e, right panel) might be deeply buried within a hydrophobic cluster comprised of Y169, I172 and A173 (Fig. 6e). This model is supported by a substitution of P2-Val by Arg that strongly reduced activation (VDAC2-3 peptide in Supplementary Fig. 15d). Collectively, our observations

**Fig. 4 | Protein-based functional correction of HtrA1 in vivo. a** Generation of *Htra1^{D174R-S328A}* mice. **b,** MS analysis of brain vessels: log₂ LFQ ratio-based heatmap of protein deregulation in *Htra1^{R274Q}* and *Htra1^{D174R-S328A}* vessels. Depicted are the proteins of interest. **c** *Htra1^{R274Q}* mice were crossed with *Htra1^{D174R-S328A}* mice to generate heterozygous *Htra1^{R274Q/D174R-S328A}* animals. **a, c** Created with BioRender.com released under a Creative Commons Attribution-NonCommercial-NoDerivs 4.0 International license https://creativecommons.org/licenses/by-nc-nd/4.0/deed.en. **d** MS signal intensity of Pan-HtrA1, Gln274-containing peptide, Ltbp4-Nt and Prss23-Ct in brain vessels (n = 5 mice per genotype). Colors are as in (**a**) and (**c**); light orange: *Htra1^{wt/R274Q}*. Pan-HtrA1, Ltbp4-Nt, Prss23-Ct: LFQ values as box-and-whisker plots (centerline: median; limits: 25th and 75th percentile; whiskers: minimum and maximum) with filled circles. The mean value in *Htra1^{wt}* samples was set to 1. Gln274 peptide: MS peptide intensity as scatter plots with filled circles (black bars: mean; nd: not detected). The mean value in *Htra1^{wt/R274Q}* vessels was set to 1. Since this peptide was not detected in 2 out of 5 samples in *Htra1^{wt/R274Q}* and *Htra1^{R274Q}* vessels, these groups were combined for statistical analysis. Significance was tested by two-

sided unpaired *t*-test; no correction for multiple comparison was performed (Pan-HtrA1: *p* = 5.9E-05; Gln274 peptide: *p* = 1.2E-02; Ltbp4-Nt: *p* = 1.3E-06; Prss23-Ct: *p* = 3.4E-03). **e** Ltbp4 and actin detected by IB in brain vessels (n = 3 mice per genotype). **f** Ltbp4 and laminin detected by IHC in brain arteries. Scale bar in left panel: 25 μm. Right panel: Ltbp4/laminin signal intensity as scatter plots (black bars: mean). Data points represent individual vessels (wt: n = 52 arteries from 3 mice; R274Q: n = 87 arteries from 4 mice; D174R-S328A: n = 54 arteries from 4 mice, R274Q/D174R-S328A: 87 arteries from 4 mice). Significance was tested by two-sided unpaired Mann-Whitney *U*-test; no correction for multiple comparison was performed (*p* = 3.0E-02). **g** Following MS analysis, principal component (PC) analysis of protein abundance was applied to the proteins of interest, excluding HtrA1 which exhibited opposite behaviors in *Htra1^{R274Q}* and *Htra1^{D174R-S328A}* vessels (filled circles: individual mice, n = 5 per genotype). PC1 and PC2, explaining most of the variance, are depicted. Colors are as in (**a**), (**c**), (**d**); light purple: *Htra1^{wt/D174R-S328A}*. Source data are provided as a Source Data file.

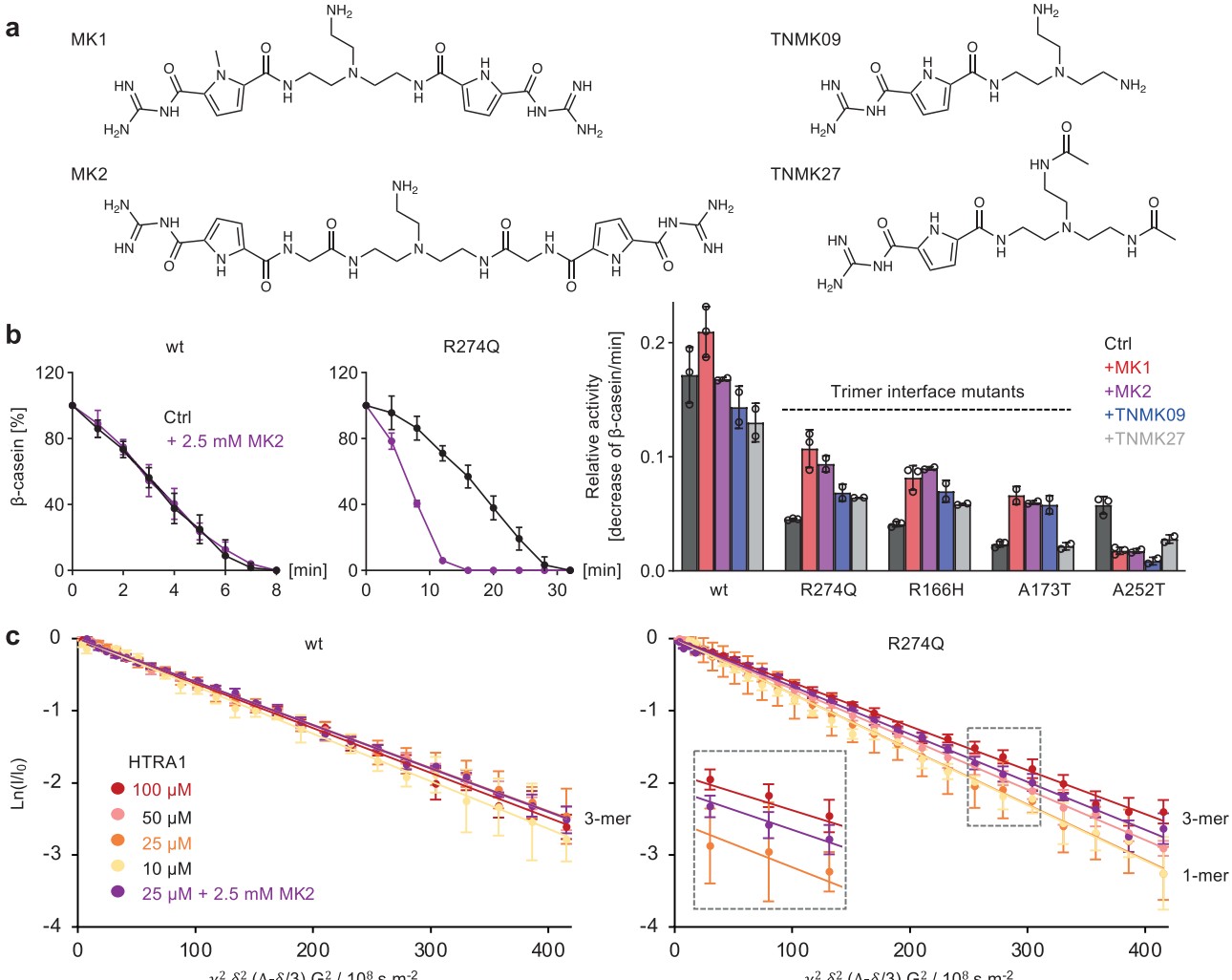

**Fig. 5 | HTRA1 assembly and activity in the presence of supramolecular ligands.**
**a** Structure of guanidiniocarbonyl pyrroles (GCPs). **b** β-casein degradation by wt and mutant HTRA1 (1 μM) in the absence (Ctrl) or presence of MK2 (2.5 mM). Left panels: activity of HTRA1 wt or R274Q. Graphs depict the relative loss of β-casein signal (mean ± SD of 2 [MK2] or 3 [Ctrl] independent experimental data). Right panel: relative activity of wt or mutant HTRA1. The rates of β-casein cleavage are presented as the maximum gradient of β-casein degradation after acceleration and before substrate exhaustion (mean ± SD of 2 [MK2, TNMK09, TNMK27] or 3 [Ctrl,

MK1] independent experimental data; empty circles: individual data points).
**c** Oligomeric states of wt and mutant HTRA1 in the absence or presence of MK2 analyzed via NMR DOSY experiments. Inlet: 25 μM R274Q (monomeric), 100 μM R274Q (trimeric) and 25 μM R274Q + MK2 2.5 mM (mean ± SD of 6 technical replicates). The slope in the Stejskal-Tanner plots represents the negative diffusion coefficient. Small molecules exhibit a larger diffusion coefficient and thus a steeper slope, while large molecules with a smaller diffusion coefficient show a shallower slope. Source data are provided as a Source Data file.

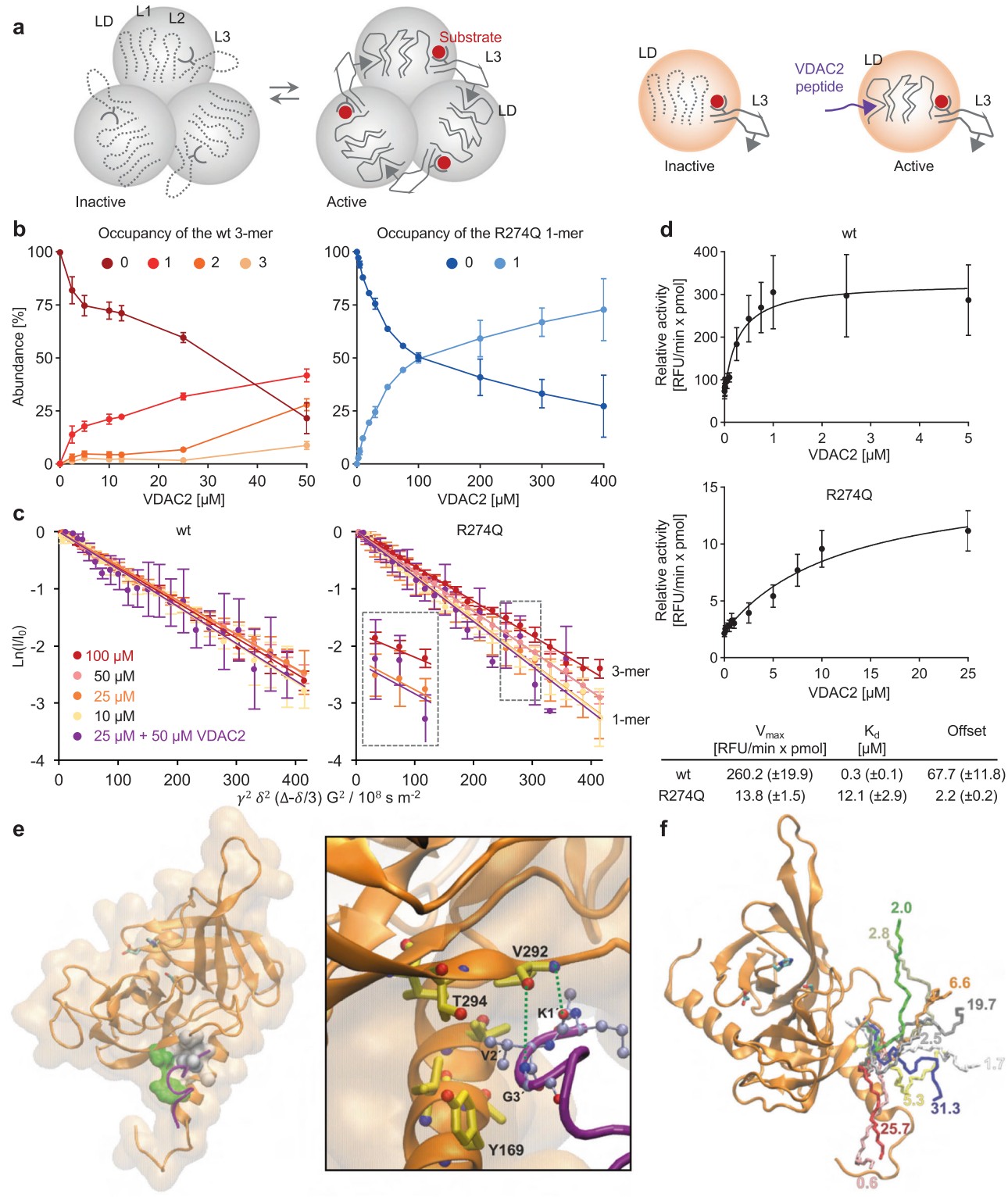

| | $V_{max}$ [RFU/min x pmol] | $K_d$ [µM] | Offset |
|---|---|---|---|
| wt | 260.2 (±19.9) | 0.3 (±0.1) | 67.7 (±11.8) |
| R274Q | 13.8 (±1.5) | 12.1 (±2.9) | 2.2 (±0.2) |

suggest that the VDAC2 peptide might act as a surrogate of the missing loop L3 of the activation domain that in wt HTRA1 is provided by an adjacent protomer (Fig. 6a). This model also explains why both monomeric HTRA1 and trimeric wt HTRA1 are activated by the peptides.

## Discussion

Pathogenic protein conformations resulting in loss of function phenotypes are difficult to address by conventional pharmaceutical compounds such as small molecule or biologic modulators targeting a

defined allosteric pocket or the active site of an enzyme. Combining in vitro, in silico and in vivo analyses, we established mechanistically distinct protein repair approaches to reverse the deleterious effects of pathogenic point mutations interfering with the assembly and catalytic function of HTRA1.

Molecular correction of pathogenic protein conformations including the stabilization of protein-protein interactions recently emerged as a promising therapeutic strategy in genetic diseases[25–30]. As a distinct strength, this approach directly targets the underlying molecular defect. A notable example that has reached clinical

**Fig. 6 | Allosteric activation of monomeric HTRA1 by peptidic modulators.**
**a** Model of activation of trimeric and monomeric HTRA1. Left panel: in wt HTRA1, the substrate bound to the active site interacts with loop L3, followed by an inter-protomer loop L3-LD interaction. This interaction mediates the proper positioning of loops L1 and L2, and of the catalytic site. Right panel: in monomeric mutant HTRA1, loop L3-LD interaction is disrupted. VDAC2 acts as a surrogate of the missing loop L3 from an adjacent protomer leading to the proper positioning of loop LD and thus of loops L1 and L2. Therefore, in active monomeric HTRA1, the activation domain resembles that of classic monomeric serine proteases[61,62].
**b** Binding of VDAC2 to HTRA1 wt trimers or to R274Q monomers analyzed by native MS. Graphs depict the occupancy (average ± SD of 4 independent datasets).
**c** Oligomeric states of HTRA1 wt and R274Q in the absence (Ctrl) or presence of VDAC2 analyzed by NMR DOSY (mean ± SD of 6 technical replicates). **d** Normalized cleavage rates of peptidic fluorescence-quenched substrate (2 μM) by HTRA1 wt

(30 nM) and R274Q (300 nM) plotted vs. VDAC2 concentration (mean ± SD of 3 independent datasets with technical duplicates). Data were fitted to the hyperbolic weak-binding equation to obtain $V_{max}$ and $K_d$ for peptide binding; brackets: SE of the fit. **e** Model of the binding mode of VDAC2 at the N-terminal tip of HTRA1-R274Q. Backbones of HTRA1 (orange) and VDAC2 (magenta), sidechains of the catalytic triad's residues shown with sticks (licorice style). Green: hydrophilic residues; gray: hydrophobic residues. Surface representation is used for HTRA1 (transparent surface). Inset: close-up view of selected HTRA1's residues nearby VDAC2. Sidechains of HTRA1's residues depicted with sticks (licorice style) and colored in yellow except oxygen (red) and nitrogen (blue). Sidechains of VDAC2 residues are shown with balls and sticks (CPK style) and colored in light blue except oxygen (red) and nitrogen (blue). **f** Representative structures of the 10 largest clusters of the advanced sampling simulations (three replicas combined). Numbers: relative population of each cluster (percent of the simulation's frames).

development is based on rescuing the consequences of the common CFTR Phe508 deletion in cystic fibrosis patients[31]. Here, we focused on pathogenic missense mutations in *HTRA1* that impair its oligomeric assembly and function. To correct the conformation and reconstitute the catalytic activity, we devised rational biologic and chemical biology approaches. In each case, the molecular mechanism of repair relies on the restoration of the integrity of the composite activation domain[32]. One approach targets the R274Q mutation by a compensatory mutant protein that is supplied in *trans*. As revealed by structural analysis, functional correction involves an overall increased surface area between protomers, which we consider as a remodeling rather than pairing between defined novel partner residues, thus underscoring the plasticity of protomer interfaces. Another approach employs supra-molecular GCP ligands to stabilize trimers. In their current form, the affinity of GCPs is rather weak and their specificity low. Nevertheless, their properties could be improved e.g., by adding properly positioned additional units. An alternative strategy involves the allosteric activation of HTRA1 monomers by decameric peptides representing the C-termini of VDAC2 and 3. Our data suggest a regulatory mechanism that leaves the known ligand binding sites i.e., the active site and the allosteric PDZ domain, unoccupied. Future efforts should be directed to improve critical parameters of these peptides such as solubility and stability in complex biological systems. HtrA proteases are regulated by the mechanism of ligand-induced activation, which can be allosteric[8,16,33]. In bacteria, such ligands include the C-termini of β-barrel outer membrane porins serving as signals of protein folding stress[33]. Therefore, the activation of HTRA1 by peptides derived from the C-termini of mitochondrial β-barrel outer membrane channels VDAC2 and 3 seems evolutionary conserved.

The majority of patients with cerebral small vessel disease, which is highly prevalent in the elderly population, do not harbor known pathogenic variants in *HTRA1* or other genes. However, there is a broader role of rare *HTRA1* variants beyond CARASIL, which is caused by biallelic *HTRA1* mutations: First, monoallelic mutations, which are linked to a milder vasculopathy with later onset, account for up to 5% of familial small vessel disease cases of unknown etiology[19,20]. The respective variants include interface mutations that destabilize the trimer such as R166C, R166L and A173P[20,34]. Second, trimer interface mutations including R166H and R274Q have been identified in cancer genome projects[35,36] thus opening additional perspectives for our correctors.

Importantly, we provide in vivo data for the consequences of R274Q and the efficacy of the complementation-based corrector (D174R-S328A). The advantage of an enzymatically inactive corrector is that restoration of activity is conditional on the presence of the pathogenic mutant thereby limiting off-target effects. Yet a practical limitation of this approach is the difficulty to deliver biologics to the brain vasculature. This limitation could be addressed by e.g., adeno-associated vectors specifically targeting the brain vasculature that have recently become available[37]. To uncover the proteomic signature

associated with R274Q and obtain a detailed readout for molecular repair, we applied mass spectrometry to the brain vasculature. As a distinct finding, we demonstrate that HtrA1-D174R-S328A substantially improved the stability of the HtrA1-R274Q protein, in accord with the principle of ligand-promoted protein folding by biased kinetic partitioning[38]. Concomitantly, the cerebrovascular accumulation of HtrA1 substrates was reduced, thus validating conformational and functional HtrA1 correction in vivo.

Collectively, our findings exemplify the feasibility of rationale strategies to correct pathogenic protein conformations thus opening perspectives for targeted protein repair.

## Methods
### Antibodies, peptides, and proteins
Mouse monoclonal antibodies against Strep-tag and His-tag were obtained from Qiagen (#34850, IB dilution 1/1000; #34670, IB dilution 1/2500). Mouse monoclonal antibodies against HTRA1 and goat poly-clonal antibodies against Ltbp4 were from R&D Systems (#MAB2916, IB dilution 1/5,000; #AF2885, IB dilution 1/1000, IHC dilution 1/100). Rabbit polyclonal anti-laminin antibodies were from Dako (#L9393, IHC dilution 1/50). Rabbit polyclonal anti-actin antibodies were from Sigma-Aldrich (#A2066, IB dilution 1/500). Alkaline phosphatase-coupled secondary antibodies were purchased from Sigma-Aldrich (#A1418, IB dilution 1/20,000), horseradish peroxidase-coupled sec-ondary antibodies from Dako (#P0447, #P0448, #P0449, IB dilution 1/10,000) and Alexa Fluor 488- or Cy3-coupled secondary antibodies from Jackson (#711-165-152, #705-546-147, IHC dilution 1/400). Pep-tides were purchased from Intavis peptides at >95% purity. Bovine β-casein and bovine serum albumin (BSA) were purchased from Sigma-Aldrich, recombinant human LTBP4s were from R&D Systems.

Note that since we and others have not detected any effect of the N-terminus of HTRA1 on oligomerization and protease function[13,14], in vitro experiments were conducted with HTRA1 proteins lacking the IGFBP7-like fragment (ΔN) except crystallography, NMR DOSY assays, some data presented in Supplementary Figs. 12a and 14a which were obtained using ΔNΔPDZ variants, and data presented in Supplemen-tary Fig. 4a which were obtained using culture medium from HEK cells transfected to overexpress full-length HTRA1.

### Plasmids
Bacterial expression plasmids based on the pET21d (Novagen) for purification of recombinant 6 × His-tagged ΔNHTRA1 wt, ΔNΔPDZHTRA1 wt, ΔNHTRA1-S328A and ΔNΔPDZ-S328A were pub-lished previously[13]. ΔNHTRA1 and ΔNΔPDZHTRA1 comprise residues 158-480 and 158-375, respectively. Expression plasmids to obtain untagged ΔNΔPDZHTRA1 used for NMR DOSY experiments were constructed by cloning the above sequence encoding ΔNΔPDZ-S328A into a modified pET41b vector harboring a GST-tag followed by a PreScission instead of a thrombin cleavage site via standard cloning techniques. Expression plasmids encoding the CARASIL-relevant or

corrector HTRA1 mutants were obtained via standard PCR mutagenesis techniques using the above plasmids as template. Expression plasmids for purification of Strep II-tagged HTRA1 comprise residues 161−480 and were constructed by PCR amplification of the N-terminally shortened sequences from the respective HTRA1 His-tagged expression vectors and subsequent cloning into a modified pET21d vector backbone containing an N-terminal Strep II-tag. Plasmids used for crystallography experiments were constructed by cloning an N-terminally shortened sequence of ΔNΔPDZ HTRA1 (residues 161−375) into a pET28a vector backbone and subsequent PCR mutagenesis. The mutational screening presented in Supplementary Fig. 4a used eukaryotic expression plasmids encoding human full-length HTRA1 (residues 1-480) fused to a C-terminal Myc-His tag[2]. Mutagenesis was conducted using the QuickChange Lightning Site-Directed Mutagenesis kit (Agilent Technologies).

### Purification of recombinant human HTRA1 and Tau

For expression of HTRA1 wt and mutant and Tau, BL21 DE3 cells were transformed with pET vectors containing the respective HTRA1 sequences or Tau sequence and grown overnight at 37 °C in 250 ml SOB medium supplemented with appropriate antibiotic. The overnight cultures were used to inoculate expression cultures. Cells were grown at 37 °C to an OD600 of ∼0.8 before induction with 0.1 mM IPTG. One hour before induction, the cultures were transferred to 25 °C and expression was carried out for 5 h, 25 °C, 180 rpm. Cells were harvested at 8000 $g$, 4 °C for 15 min.

For protease activity assays, crosslinking assays, SEC, SEC-MALS, AUC and native MS, as well as crystallography experiments, the 6× His-tagged and Strep II-tagged variants of human HTRA1 were purified as described below. His-tagged HTRA1 wt and mutants were purified as described previously[13]. After Ni-NTA affinity chromatography, proteins were further purified by SEC using Superdex 200 26/60 preparation grade columns (GE Healthcare) with 50 mM NaPi, 100 mM NaCl pH 8.0 as mobile phase. Strep II-tagged HTRA1 was purified via Strep-Tactin affinity column (IBA lifesciences), followed by hydroxyapatite (HAP) column and subsequent SEC.

Briefly, bacterial lysates containing overexpressed proteins were resuspended in equilibration buffer (50 mM $NaH_2PO_4$, 300 mM NaCl pH 8.0) and lysed using French press. Lysate was cleared by centrifugation (50,000 $g$, 4 °C, 40 min) and supernatant was applied on pre-equilibrated Strep-Tactin column. Column was washed in three successive steps with 5 column volume (CV) of equilibration buffer, 8 CVs of wash buffer (50 mM $NaH_2PO_4$, 1 M NaCl pH 8.0) and 3 CVs of equilibration buffer. HTRA1 wt and mutant proteins were eluted with equilibration buffer supplemented with 2.5 mM desthiobiotin. Eluted proteins were then diluted with 2 volumes of HAP equilibration buffer (50 mM HEPES, 50 mM NaCl, 5 mM KPi pH 7.8), applied on pre-equilibrated HAP column and washed in successive gradient steps with equilibration buffer (3 CV) containing 10% (5 CV), 15% (5 CV), 17%, 18% and 19% (3 CV each) of HAP elution buffer (50 mM HEPES, 50 mM NaCl, 476.6 mM $K_2HPO_4$, 23.4 mM $KH_2PO_4$ pH 7.8). HTRA1 wt and mutants were eluted with linear gradient containing 19−50% elution buffer. After HAP chromatography, HTRA1 samples were further purified by SEC using Superdex 200 26/60 preparation grade columns (GE Healthcare) with 100 mM NaPi, 150 mM NaCl pH 8.0 as mobile phase.

To obtain untagged ΔNΔPDZ HTRA1 utilized in NMR DOSY experiments, GST-tagged constructs were bound onto a glutathione sepharose (GSH) 4 Fast Flow column (GE Healthcare) equilibrated with buffer A (100 mM Tris-HCl, 150 mM NaCl pH 7.5), washed with 4 CV of buffer A and 3 CV of buffer B (100 mM Tris-HCl, 1 M NaCl pH 7.5) followed by an on-column cleavage step using PreScission protease. Cleaved HTRA1 was eluted from the column using 3 CVs of buffer A. Following a reverse GSH column in buffer A, samples were concentrated and subjected to SEC on a Superdex 75 preparation grade column with 50 mM NaPi, 100 mM NaCl pH 8 as mobile phase.

Protein concentrations were determined by Bradford assays, UV−vis and SDS-PAGE. The concentrations of wt and mutant HTRA1 refer to the monomer.

Human 4R wt Tau (residues 1−441) was purified by boiling of cleared bacterial lysates followed by cation exchange chromatography, HAP chromatography and SEC as follows. Bacterial cell pellets containing overexpressed Tau were resuspended in lysis buffer (50 mM HEPES, 100 mM NaCl, 2 mM dithiothreitol [DTT] pH 7.5) and lysed using French press. Subsequently, lysate was cleared by centrifugation (50,000 $g$, 4 °C, 40 min) and supernatant was boiled for 15 min in water bath. After an additional centrifugation step (21,000 $g$, 4 °C, 30 min), supernatant containing Tau protein was supplemented with fresh DTT and applied on a POROS HS50 column (Thermo Fisher scientific) equilibrated with lysis buffer. Column was washed with 3 CV of equilibration buffer and Tau was eluted with a linear gradient comprised of 0−30 % HS elution buffer (50 mM HEPES, 1 M NaCl, 2 mM DTT pH 7.5). Eluted fractions containing Tau were pooled and mixed with 3 volumes of HAP equilibration buffer (100 mM HEPES, 10 mM KPi, 2 mM DTT pH 7.6) and applied on an equilibrated HAP column. Column was washed with 3 CVs of equilibration buffer and 3 CVs of 10% elution buffer (100 mM HEPES, 10 mM KPi, 1 M NaCl, 2 mM DTT pH 7.6). Tau was eluted with a linear gradient from 10−50% elution buffer. Eluted Tau was concentrated and further purified by SEC using Superdex 200 26/60 preparation grade columns (GE Healthcare) with 10 mM HEPES, 50 mM $(NH_4)SO_4$, 2 mM TCEP pH 7.5 as mobile phase.

Protein concentrations were determined by UV absorption at $A_{280}$ and Bradford assays.

### In vitro proteolysis assays

To follow the degradation of the substrates ß-casein and Tau, 1 μM HTRA1 was incubated with 20 μM casein or 3 μM Tau respectively at 37 °C. ß-casein degradation was performed in 20 mM NaPi, 20 mM NaCl pH 8.0. In cases where effects of GCP compounds were tested, HTRA1 was incubated with GCP compounds at 37 °C for 30 min prior to ß-casein addition. Tau degradation was performed in 50 mM Tris-HCl pH 8.0. In cases where effects of peptidic activators were tested, HTRA1 was incubated with peptidic activator at 37 °C for 10 min prior to Tau addition. At the indicated time points aliquots were taken and the reaction was stopped by adding 5x SDS loading dye. Subsequently, aliquots were incubated for 5 min at 95 °C and analyzed by SDS-PAGE. Proteins were detected by Coomassie Brilliant Blue staining. Gels were subsequently scanned and the intensity of ß-casein or Tau bands was quantified with TotalLab Quant software (TotalLab) using raw, unmodified gel images.

For BSA degradation assays, human embryonic kidney (HEK) 293T cells were transfected with Lipofectamine 2000 (Invitrogen). Cells were maintained for 48 h in serum-free Dulbecco's Modified Eagle Medium (Gibco) before culture medium was collected and centrifuged for 10 min at 400 $g$, 4 °C. 5 μg BSA were slightly denatured by addition of 1.5 mM DTT, then exposed to 20 μl culture medium from HTRA1-overexpressing cells for 24 h at 37 °C. BSA degradation was evaluated by SDS-PAGE followed by Coomassie staining.

To evaluate the processing of LTBP4 by HTRA1, 1 μg LTBP4s was treated with 175 ng of wt HTRA1 for 24 h at 37 °C in a final volume of 10 μl 50 mM Tris 150 mM NaCl, pH 8.0. The reaction mixture was analyzed by SDS-PAGE and silver staining or immunoblot as indicated.

Uncropped gels are presented in the Source Data file.

### Enzyme assays with chromogenic and fluorescence-quenched substrates

Activity assays of wt and mutant HTRA1 with the synthetic peptide substrate IRRVSYSFKKC labeled with the fluorescent DY649P1 label and the quencher DYQ661 at the N- and C-terminus respectively (Intavis), were performed in 20 μl of 50 mM HEPES pH 8.0 at 37 °C in 384-well low-binding plates (Corning #4514). IRRVSYSFKKC was first

reported by ref. 39 as an HTRA2 substrate and was described as an HTRA1 substrate by ref. 14. Before addition of substrate, proteins were preincubated with either DMSO or VDAC2 peptide for 5 min at 37 °C. Cleavage of the labeled peptide by HTRA1 results in increased fluorescence intensity as the average distance between DY649P1 and its quencher greatly increases. Such assays have been extensively used in previous studies for measuring proteolytic activity, however, it is not without technical drawbacks. The substrate peptide is initially dissolved in DMSO prior to dilution into the proteolysis reaction mix, resulting in a final DMSO concentration of <2%. The rapid change of solvent conditions can result in a decrease in solubility resulting in a background of aggregation-induced emission, especially when the final concentration of the substrate is above 10–15 μM[40,41]. After initiating the reaction, the change in fluorescence intensity was recorded continuously over 30 min at 677 nm with excitation at 649 nm at 1 min intervals (Molecular Devices Spectramax 5e reader).

Optimal enzyme concentrations were determined with a substrate peptide concentration of 2 μM and varying enzyme concentrations. Following enzyme concentrations were employed in all other assays: wt: 30 nM; R274Q: 300 nM; R166H: 500 nM; A173T: 100 nM. Higher concentrations of mutants had to be used in the assays to obtain a good signal-to-noise ratio due to their relative inactivity compared to the wild-type protein.

The effects of VDAC2 on the activity of wt and mutant HTRA1 proteins were initially analyzed at constant substrate concentration (2 μM) by titrating VDAC2 concentrations (0–25 μM). VDAC2 concentrations >25 μM could not be employed in these assays due to low solubility. To further determine whether VDAC2 acts as an allosteric activator, substrate titrations were performed in either absence or presence of fixed VDAC2 concentrations.

In general, to obtain the rate of the reaction, the first derivative of each reaction result was calculated and the derivatives from 6 to 12 reactions were averaged. For some of the mutants, an acceleration phase could be observed before the rate of proteolysis reached a maximum. In the absence of VDAC2, this was markedly observed with A173T, however, rate acceleration was observed during proteolysis assays in the presence of VDAC2 with R274Q, R166H, A173T and to a much smaller extent with wt HTRA1. In each case, the maximum rate (normalized for protein concentration) was recorded and plotted as a function of either VDAC2 or substrate concentration, depending on the experiment. The precise underlying cause for the acceleration is beyond the scope of this manuscript, but could be due to a slow conformational transition resulting from the binding of substrate to a mutant in which the mutation has compromised the integrity of the folded monomer to some degree, as well as disrupting the assembly of the trimer.

Data obtained from VDAC2 titration at fixed substrate concentrations were fitted to a weak-binding hyperbola with offset.

Data obtained from substrate titration in the absence or presence of fixed VDAC2 concentrations were fitted as follows: In case of wt HTRA1 after initial increase in activity up to 5–6 μM substrate, a decrease was observed at higher substrate concentrations. The mechanism of inhibition was assumed to result from substrate inhibition affecting the allosteric communication within the HTRA1 trimer. Therefore, data were fitted to a modified Hill model in which substrate can bind cooperatively and act as both a substrate and an inhibitor and this allowed the determination of steady-state kinetic parameters that could be compared when VDAC2 was added to the assay[42]. The model is over-parameterized for the quantity of data being fitted, especially since the endpoint at high substrate concentrations is inaccessible due to technical limitations. Therefore, the Hill constant for the binding of the substrate as an inhibitor (z) was fixed at 3 (HTRA1 being a trimer) which allowed the most optimal fit, albeit with large errors in $V_{max}$ and $V_i$. Fixing at lower or higher values resulted in larger errors. Although this is not ideal, visual inspection of these data makes it clear that the presence of VDAC2 enhances the maximal activity of wt HTRA1 and shifts the mid-point to a lower substrate concentration. Data for R274Q and R166H were fitted to the Michaelis-Menten equation. In case of R166H, to obtain a fit to the dataset without VDAC2 $V_{max}$ was fixed at levels observed in the presence of VDAC2. Data obtained from substrate titration with A173T were also fitted with Michaelis-Menten model. In case of A173T, in presence of VDAC2 substrate inhibition was observed at higher substrate concentrations and therefore that dataset was fitted to a modified Michaelis-Menten model including non-cooperative substrate inhibition.

The effects of VDAC2 on reference serine proteases were tested as follows. Following preincubation with VDAC2 (0.05–100 μM), proteases were incubated with their respective substrates at 37 °C. The activity of bovine pancreatic trypsin (50 nM, Sigma-Aldrich, #T1426) was measured against Bz-DL-R-pNA (0.5 mM, Sigma-Aldrich #B4875) in 50 mM HEPES pH 8.0. The activity of human pancreatic chymotrypsin (100 nM, Sigma-Aldrich #SRP6509) was measured against AAF-pNA (0.5 mM, Bachem #4002753) in 100 mM HEPES, 500 mM NaCl, 0.05% Tween 20 pH 7.25. The activity of human neutrophil elastase (100 nM, Enzolifescience, #BML-SE284) was measured against MeOSuc-AAPV-pNA (0.5 mM, Sigma-Aldrich, #M4765) in 100 mM HEPES, 500 mM NaCl, 0.05% Tween 20 pH 7.25. The activity of human HTRA2 (100 nM, our lab) was measured against Mca-IRRVSYSF(Dnp)KK (10 μM) in 50 mM HEPES, pH 8.0. The activity of human HTRA3 (30 nM, our lab) was measured using DY649P1-IRRVSYSFKKC-DYQ661 (2 μM, Intavis) in 50 mM HEPES pH 8.0 buffer. HTRA2 lacking a transmembrane segment and HTRA3 lacking the IGFBP-Kazal domain were produced as previously described for HTRA2[39] except that we omitted Glycerin in Ni-column Elution buffer and performed an additional SEC run (50 mM HEPES, 150 mM NaCl pH 8.0). Absorbance or fluorescence was measured continuously at 1 min intervals (Molecular Devices Spectramax 5e reader).

### Glutaraldehyde crosslinking
0.5 μM HTRA1 were incubated in reaction buffer (50 mM NaPi, 50 mM NaCl pH 8.0) at 350 rpm, 37 °C for 10 min prior to the crosslinking reaction. Subsequently 0.5% glutaraldehyde was added and samples were incubated for additional 2 min at 37 °C. The reaction was stopped via incubation with 75 mM TRIS for 15 min at 500 rpm, room temperature and subsequent incubation at 40 °C for 3 min. Samples were analyzed via electrophoresis on NuPage Novex TRIS Acetate gels (3–8%) followed by silver staining or Western Blot analysis.

### Analytical SEC
Analytical SEC was carried out on an Agilent 1260 Infinity II LC system equipped with a G7115A DAD detector, using an Agilent AdvanceBio SEC 300 Å 2.7 μm column in a 4.6 × 300 mm dimension column and using 2×PBS (made from Roti-Stock 10xPBS, Roth) as the mobile phase with a flow rate of 0.35 ml/min. 10 μl of 10 mg/ml protein solution (273 μM monomer equivalent) was applied, except for A173T where protein concentration was 6.5 mg/ml (178 μM monomer equivalent).

The effect of MK1 compound on the oligomeric state of R274Q was analyzed on a ÄktaMicro FPLC system (GE Healthcare) using a Superdex 200 Increase column in 3.2 × 300 mm dimensions and 20 mM NaPi, 50 mM NaCl pH 8.0 with or without 2.5 mM MK1 as mobile phase with a flow rate of 0.05 ml/min. Prior to SEC, protein samples were incubated at 37 °C for 30 min (with or without MK1) and subsequently centrifuged at 21,000 g for 10 min. 10 μl of 3.6 mg/ml protein solution (100 μM monomer equivalent) was applied. Eluted samples were collected in 50 μl fractions and analyzed using SDS-PAGE followed by Coomassie staining.

Astra 8 software was used for system control, data collection and data analysis.

## SEC-MALS

SEC was carried out on an Agilent 1260 Infinity LC system using a Superdex 200 Increase column in $10 \times 300$ mm dimension and using 2x PBS as the mobile phase with a flow rate of 0.5 ml/min. 56 μl of 10 mg/ml protein solution (273 μM monomer equivalent) was applied, except for A173T where protein concentration was 6.5 mg/ml (178 μM monomer equivalent). The effluent was detected by an Agilent G7114A VWD UV detector at 280 nm for proteins, followed by a DAWN HELEOS 8 + MALS detector (Wyatt Technology Europe GmbH) and an Optilab T-rEX differential refractive index (RI) detector. Astra 8 software was used for system control, data collection and data analysis. A *dn/dc* value of 0.185 ml/g was used for proteins. The species distribution was analyzed using the peak analysis tool of the Origin software.

## Analytical ultracentrifugation

Sedimentation velocity experiments were performed on an Optima XL-I analytical centrifuge (Beckman Inc.) using an 60 Ti rotor and double-sector 12 mm centrepieces. The proteins were used in 100 mM $NaH_2PO_4$, 150 mM NaCl pH 8.0 buffer at 2.8 mg/mL. Buffer density was measured to 1.01316 kg/L using a DMA 5000 densitometer (Anton Paar). Protein concentration distribution was monitored at 280 nm, using 50,000 rpm rotor speed. Time-derivative analysis was computed using the SEDFIT software package[43], resulting in a c(s) distribution and an estimate for the molecular weight (from sedimentation coefficient and the diffusion coefficient, inferred from the peak width).

## Native mass spectrometry

The purified HTRA1 proteins were exchanged to 200 mM ammonium acetate buffer (pH 6.8) (A2706, Sigma-Aldrich, diluted in MS water 14263, Honeywell). To this end, a rebuffering procedure was six times applied: dilution of the protein sample in 200 mM ammonium acetate buffer (pH 6.8), followed by protein concentration via 10 kDa molecular weight cut-off spin-filter columns (UFC501096, Millipore)[44,45]. The resulting protein concentration was quantified via NanoDrop measurements. Protein complex formation was measured at a protein concentration range from 0.5 to 25 μM. Binding experiments were performed at a protein concentration of 5 μM with addition of the described molar ratios of the ligand.

Samples were ionized using a TriVersa NanoMate nanoESI System equipped with 5 μm diameter nozzle spray chips (Advion). A total sample volume of 2 μL was picked out of 96-well plates and the ESI spray was generated using 0.8 psi nitrogen backpressure combined with a positive nozzle chip voltage of 1.7 kV with spray sensing turned on (15 s threshold). MS spectra were recorded for 2 min in positive EMR mode on an Exactive Plus EMR Orbitrap mass spectrometer (ThermoFisher) calibrated with CsI (2 mg mL$^{-1}$, 192820010, Thermo Scientific). The used mass spectrometer parameters are listed in Supplementary Table 2 and were obtained by optimization of the parameters reported by ref. 46. All samples were measured in technical triplicates (for complex formation measurements) or quadruplicates (for binding studies)[46].

Mass deconvolution as well as data analysis was performed using UniDec software (version 5.0.2)[47]. The measurements of a concentration series were merged into a single HDF5 file via HDF5 Import Wizard. These files were analyzed separately via MetaUniDec, resulting in the deconvoluted mass and intensity values for each condition. Standard UniDec settings with the exceptions described in Supplementary Table 2 were used. For data visualization, GraphPad (version 9.3.1) was used.

## Mouse lines

Knock-in mice were generated by CRISPR/Cas9-assisted gene targeting in mouse zygotes. Pronuclear stage zygotes were obtained by mating C57BL/6J males with superovulated C57BL/6J females (Charles River) and *Htra1*-specific Cas9 ribonucleoproteins (RNP, Supplementary Table 3) were either microinjected or electroporated.

*Htra1R274Q* RNP consisted of 50 ng/μl SpCas9 protein (PNA Bio), 1 μM crRNA_R274 (IDT), 1 μM tracrRNA (IDT), and 50 ng/μl mutagenic single-stranded oligodeoxynucleotide (ssODN) ssHtra1_R274Q, comprising the R274Q substitution and three additional silent mutations for protospacer disruption (IDT, see Supplementary Table 3). *Htra1R274Q* RNP was microinjected into the male pronucleus[48] and treated zygotes were transferred to pseudopregnant foster animals.

*Htra1D174R-S328A* RNP consisted of 200 ng/μl SpCas9 protein (IDT), 3 μM crRNA_D174 (IDT), 3 μM crRNA_S328 (IDT), 6 μM tracrRNA (IDT), and two mutagenic ssODNs ssHtra1_D174R and ssHtra1_S328A (150 ng/μl each), comprising the D174R and S328A substitution and additional silent mutations for protospacer disruption or genotyping purposes (IDT, Supplementary Table 3). *Htra1D174R-S328A* RNP was electroporated into zygotes using the NEPA21 electroporator and a 1 mm electrode and treated zygotes were transferred to pseudopregnant foster animals.

For genotyping each line, genomic DNA was prepared from ear punches and the locus harboring the knock in mutation was amplified by PCR using primers Exon4-F/4-R, Exon2-F/2-R, and Exon5-F/-R (Supplementary Table 3). Amplicons were analyzed by Sanger sequencing and/or restriction digest using PstI, XhoI, or NaeI (New England Biolabs), respectively. To exclude additional unwanted modifications, putative off target sites were predicted using the online tool CRISPOR[49]. Genomic DNA from $F_1$ animals was then PCR-amplified and verified by Sanger sequencing and did not show additional sequence variation.

Mice were maintained on a C57BL/6J background. Animals were kept under standard conditions in a specific pathogen-free facility at 20–24 °C and 45–65% humidity on a 12-h light/dark cycle and had access to food and water *ad libitum*. All mouse-based experiments were performed in accordance with the German Animal Welfare Law (§4 TSchG) and in compliance with the Government of Upper Bavaria (approval number: ROB-55.2Vet-2532.Vet-02-16-121). Experiments reported in this study were all performed on 6-month-old mice and used 3–5 sex-matched (mixed males and females) animals per genotype, as indicated in figure legends. The study was not designed to perform sex-based analyses and is not suitable for this purpose due to low sample size.

## Preparation of mouse brain vessel extracts

Mice were anesthetized with Ketamine (100 mg/kg) and Xylazine (10 mg/kg) and transcardially perfused with 20 ml PBS, followed by 2 ml of 2% Evans blue (w/v) prepared in PBS. After brain harvest, vessels were collected with micro forceps under an M205 A dissection microscope (Leica) and snap-frozen on dry ice.

Vessels were mixed with 4% (w/v) SDS, 100 mM DTT, 100 mM Tris–HCl, pH 7.6 and homogenized in a TissueLyser LT bead mill (Qiagen, $2 \times 3$ min at 50 Hz). After heating for 3 min at 95 °C, samples were sonicated 5 times for 30 s at 4 °C in a VialTweeter sonicator (Hielscher, amplitude 100%, duty cycle 50%), then centrifuged at 16,000 g for 15 min. The supernatant was stored at −80 °C before further processing. For MS analysis, proteins were precipitated overnight at −80 °C after adding NaCl and ethanol to a final concentration of 50 mM and 90% (v/v), respectively. Precipitated proteins were washed once with 90% EtOH and the precipitate was dissolved in 50 mM Tris pH 7.5, 150 mM NaCl, 2 mM EDTA, 1% Triton X-100.

## Mass spectrometry

An amount of 25 units of Benzonase (Sigma-Aldrich) was added to the entire sample volume and samples were incubated for 30 min at 37 °C at 1400 rpm in a Thermomixer (Eppendorf) to remove remaining DNA. Afterwards, samples were digested with LysC and trypsin, using single-pot solid-phase-enhanced sample preparation (SP3)[50]. Proteolytic

peptides were dried by vacuum centrifugation and dissolved in 20 µl 0.1% (v/v) formic acid. 1.2 µg of peptide mixture was separated on a nanoLC system (EASY-nLC 1200, Thermo Fisher Scientific) using an in-house packed C18 column (30 cm × 75 µm ID, ReproSil-Pur 120 C18-AQ, 1.9 µm, Dr. Maisch GmbH) with a binary gradient of water and 80% acetonitrile (ACN) containing 0.1% formic acid (0 min, 3% ACN; 3.5 min, 6% ACN; 137.5 min, 30% ACN; 168.5 min, 44% ACN; 182.5 min, 75% ACN; 185 min, 99% ACN; 200 min, 99% ACN) at 50 °C column temperature. The nanoLC was coupled online via a nanospray flex ion source equipped with a column oven (Sonation) to a Q-Exactive HF mass spectrometer (Thermo Fisher Scientific). Full MS spectra were acquired at a resolution of 120,000 and a $m/z$ range from 300 to 1400. The top 15 peptide ions were chosen for collision induced dissociation (resolution: 15,000, isolation width 1.6 m/z, AGC target: 1E + 5, NCE: 26%). A dynamic exclusion of 120 s was used for peptide fragmentation.

The raw data were analyzed with the Maxquant software (max-quant.org, Mack-Planck Institute Munich[51]) version 1.5.5.1 and searched against a modified, reviewed isoform FASTA database of *Mus musculus* (UniProt, 26.11.2018, 25,217 entries, Supplementary Data 1). Where indicated, the FASTA file was implemented with the sequences of HtrA1-R274Q and HtrA1-D174R-S328A (for analysis of individual tryptic peptides, Supplementary Data 2) or was modified as follows (Supplementary Data 3): the canonical HTRA1 sequence was replaced with Pan-HtrA1 sequence (excluding Arg274); Ltbp4 was replaced with two sequences, which separately depict its N- and C-terminal parts (aa 1–143 and aa 144–1666, respectively); Prss23 sequence was shortened (aa 50-382) to only include its C-terminal part. Two missed trypsin cleavages were allowed. Oxidation of methionine and N-terminal acetylation were set as variable, carbamidomethylation of cysteine as static modifications. For the main search peptide and peptide fragment mass tolerances were set to 4.5 and 20 ppm, respectively. The false discovery rate (FDR) for both peptides and proteins was adjusted to less than 1% using a target and decoy (reversed sequences) search strategy. Label-free quantification (LFQ) of proteins required at least two ratio counts of unique peptides. Unique and razor peptides were used for quantification. Relative quantification and statistical analysis were performed for all proteins identified in at least 3 samples of each group. LFQ intensities were $\log_2$-transformed and a two-sided Student's t test was used to evaluate the significance of proteins with changed abundance. A $\log_2$ fold change $<-0.4$ or $>0.4$ and a $p$ value < 0.05 were set as significance thresholds. A permutation-based FDR estimation was used to account for multiple hypotheses ($p = 5\%$; s0 = 0.1) using the software Perseus (version 1.6.2.1)[52,53]. Mouse brain vessel proteomic data have been deposited to the ProteomeXchange Consortium via the PRIDE[54] partner repository with the dataset identifier PXD024683. Supplementary Data 1–3 and Source data provide detailed information on the proteomic data analyses depicted in the Main and Supplementary Figs..

After imputation of missing values using the "*mice*" package, principal components analysis (PCA) was performed using "*prcomp*". PCA results were visualized using "*fviz_pca_ind*" from the "*factoextra*" package. All analyses were performed in R version 4.0.2.

## Quantitative real-time PCR

Mouse brain vasculature was isolated and prepared exactly as previously described[3] by tissue homogenization at 4 °C in a glass tissue grinder (Wheaton) in 15 ml minimum essential medium, Ficoll gradient centrifugation at 6000 *g* for 20 min at 4 °C (Ficoll 400 from Sigma-Aldrich was used at a final concentration of 15% w/v), and filtration on a 40-µm nylon mesh (Corning). Brain vessels were mixed with QIAzol Lysis Reagent and homogenized in a TissueLyser LT bead mill (all from Qiagen). Total RNA was extracted using the RNeasy Tissue Mini Kit (Qiagen). A DNase I digestion step (Qiagen) was included. RNA concentration was measured on a Nanodrop ND-1000 spectrophotometer

(Peqlab). cDNA was synthetized from 1 µg RNA using the Omniscript RT kit (Qiagen) and oligo-dT Primers (Metabion). qRT-PCR was performed in a LightCycler 480 (Roche Diagnostics) using Brilliant II SYBR Green MasterMix (Agilent Technolgies) and intron spanning primer pairs (Metabion, primer sequences are provided in Supplementary Table 3). Measurements were performed in triplicates. Gapdh was used as a housekeeping gene and relative expression levels were determined using the ΔΔCT method.

## SDS-PAGE and immunoblot

Proteins were separated by SDS-PAGE and silver-stained. Alternatively, proteins were transferred onto *polyvinylidenfluoride* membranes (Immobilon-P, Millipore). Membranes were blocked in Tris-buffered saline (TBS) containing 0.2% (v/v) Tween 20 and 4% (w/v) skim milk and probed with the primary antibodies prepared in TBS/Tween/milk. Detection was performed with alkaline phosphatase- (Sigma-Aldrich) or horseradish peroxidase-coupled secondary antibodies (Dako) and appropriate substrates. Uncropped blots are presented in the Source Data file. Signals were quantified using the Image J (Fiji) Software.

## Immunohistochemistry

Mice were transcardially perfused with PBS followed by 4% (w/v) par-aformaldehyde. Brains were harvested and post-fixed overnight in paraformaldehyde. After washing and dehydration in EtOH, tissue was embedded in paraffin using a spin tissue processor (STP120, Thermo Scientific). 7 µm-thick sagittal sections were prepared on a HM 340E microtome (Thermo Scientific). After deparaffinization and rehydration, tissue was treated with PBS containing 20 µg/ml proteinase K (Sigma-Aldrich) for 10 min at room temperature to retrieve antigens. Sections were permeabilized and blocked in PBS added with 0.1% (v/v) Triton-X100 and 5% (w/v) BSA for 1 h at room temperature. Samples were probed overnight at 4 °C with the primary antibodies diluted in PBS/Triton/BSA, washed with PBS and incubated with the secondary antibodies diluted in PBS for 1 h at room temperature. After washing, tissue was mounted with Fluoromount (Sigma-Aldrich) and images were acquired on an Axio Observer Z1 fluorescence microscope using the Zen user interface (both Zeiss). Individual arteries were analyzed using the Image J (Fiji) Software. Laminin staining was used to select the area corresponding to the arterial wall and the mean fluorescence intensity was measured.

## Determination of diffusion coefficients via NMR DOSY experiments

NMR samples contained HTRA1 alone (10 µM, 25 µM, 50 µM, 100 µM), HTRA1 (25 µM) with ligand MK2 (2.5 mM), or HTRA1 (25 µM) with VDAC2 peptide (50 µM) in 50 mM potassium phosphate buffer (pH 6.7) with 10% $D_2O$ in 3 mm sample tubes (200 µL). No DSS as an internal chemical shift standard was used because the DSS propane sulfonic acid signals overlap with the protein signals used for DOSY analysis.

Diffusion Ordered Spectroscopy (DOSY) experiments were performed at 25 °C (298 K) on a Bruker 700 MHz Avance Ultrashield NMR spectrometer (Bruker) equipped with a 5 mm CPTCI $^1$H-$^{13}$C/$^{15}$N/D cryoprobe with z-gradient and Topspin 3.5 software for data acquisition and processing. The variable gradient diffusion data was acquired as a pseudo-2D data set using a stimulated echo pulse sequence (stegp1s) from the Bruker pulse program library, which was modified with a presaturation water suppression, with a diffusion delay (Δ) of 100 ms and a diffusion gradient length (δ) of 6 ms. Thirty-two gradient experiments were acquired for each data set with the gradient strength incremented from 5 to 95% of the maximum gradient strength (50.4 G/cm for a smoothed square gradient pulse) in 32 steps using a linear ramp. The number of transients was chosen according to protein concentration (100 µM HTRA1: 32 transients, 50 µM HTRA1: 96 transients, 25 µM HTRA1: 256 transients, 10 µM HTRA1: 640 transients, 25 µM HTRA1 + ligand: 800 transients).

The spectra were Fourier transformed and phased in Topspin 3.5. The integration of signals as a function of gradient strength was performed using the built-in T1T2 Dynamics module. The first data point was excluded due to irregular phasing and the last 4 data points were excluded due to low signal/noise. For free HTRA1 protein, the normalized $I/I_0$ values of four signals (3.0 ppm, 1.6 ppm, 1.4 ppm and 0.8 ppm) were averaged. For the VDAC2 peptide, only two signals (3.2 ppm and 0.7 ppm) and for ligand MK2 three signals (1.6 ppm, 1.4 ppm and 0.8 ppm) could be averaged because other protein signals overlapped with the signals of ligands or impurities found in the ligand preparations.

Plotting and fitting of the linearized data according to the Stejskal-Tanner equation[24,55] was performed in GraphPad Prism 5. Error bars for the averaged data points represent the standard deviation. The error of the diffusion coefficient represents the standard error of the fit.

$$\ln\left(\frac{I}{I_0}\right) = -\gamma^2\delta^2\left(\Delta - \frac{\delta}{3}\right)\cdot D \cdot G^2$$

$I$ = signal intensity $I_0$ = signal intensity at gradient strength 0
$\gamma$ = gyromagnetic ratio of the ${}^1$H nucleus
$\delta$ = diffusion gradient pulse length
$\Delta$ = diffusion delay
$G$ = gradient strength
$D$ = translational diffusion coefficient.

## X-ray methods

HTRA1ΔNΔPDZ derivatives were used for crystallization because attempts using constructs that include the PDZ domain were not successful, probably due to the well-known mobility of the PDZ domain[13,14]. In addition, proteolytically inactive variants, carrying a Ser328Ala mutation of the active site, were used to avoid autodegradation. Purification was carried out as described[13]. Both constructs used for crystallization comprise the protease domain of human HTRA1 (residues 161-375) and have an N-terminal His-Tag followed by a thrombin cleavage site (in bold) with the sequence MGSS-HHHHHH-SSG-**LVPRGS**-HM that precedes the HTRA1 sequence.

HTRA1ΔNΔPDZ S328A R274Q was crystallized by mixing 9 mg/ml protein in 20 mM TRIS/HCL pH 7.5, 300 mM NaCl and 5% glycerol with a reservoir solution of 100 mM sodium citrate pH 5, 1 M LiCl, and 20% PEG6000 (Nextal screen JCSG Core 2 F11, Qiagen, Hilden). The mutant HTRA1ΔNΔPDZ S328A D174R R274Q was crystallized by mixing a solution of either a protein solution of 15 mg/ml in 50 mM Hepes pH 8.0 and 400 mM NaCl ("crystal 1"), or of 10 mg/ml protein in 10 mM HEPES pH 8.0 and 150 mM NaCl ("crystal 2"), respectively, with a reservoir solution of 1.6 M ammonium sulfate, 100 mM Citrate pH 4 (Nextal screen JCSG Core 3 H9, Qiagen, Hilden). The crystals were flash-frozen in liquid nitrogen without additional cryoprotectant.

Data was collected at 100 K using a Pilatus 6 M detector at the X10SA beamline of the SLS synchrotron in Villigen, Switzerland. Data sets were integrated using XDS or DIALS[56,57] and scaled with XSCALE[56].

The structures were solved via molecular replacement with PHA-SER (CCP4 suite)[58] using PDB ID 3TJO as a template. HTRA1ΔNΔPDZ S328A R274Q crystallized in space group H3 with two molecules per asymmetric unit and a unit cell of 109.9 Å, 109.9 Å, 114.4 Å, 90°, 90°, 120°, whereas the double mutant HTRA1ΔNΔPDZ S328A D174R R274Q crystallized in space group P6₃ with three molecules per asymmetric unit and unit cells of 101.4 Å, 101.4 Å, 144.9 Å, 90°, 90°, 120° ("crystal 1"), and 103.5 Å, 103.5 Å, 147.3 Å, 90°, 90°,120° ("crystal 2"), respectively. Crystal 1 was merohedrally twinned, but since it had much better resolution and better-defined density than the (untwinned) structure of crystal 2, we decided to refine both for comparison. Residues visible in the electron density are given in Supplementary Table 4.

Refinement with REFMAC (CCP4 suite[58]) and PHENIX[59] and model building with COOT[60] resulted in models with good geometries. NCS restraints were used in the beginning, but had to be relaxed due to the significant differences between the molecules in the asymmetric unit, especially at the termini and in the loop region of residues 190–200. Twin refinement of crystal 1 using the twin operator h,-h-k,-l reduced the R factors by 12% (twin fraction 0.48). In this case, the free *R* value is not indicative of the quality of the model, so we used the untwined structure of crystal 2 to ensure the correctness of the structure. Furthermore, an omit map was used to check and correct for model bias. The r.m.s.d values between the monomers in each structure are in a similar range as those between the three structures and the wild type structure (3TJO) and range from 0.369 Å to 0.475 Å (1077 to 1186 atoms, calculated with PYMOL). As expected from the high Wilson B-factors, the B-factors for all structures are also quite high, probably due to inhomogeneous packing, since TLS refinement did not lower the R factors significantly.

Although the mutants did not crystallize with the canonical trimer (as found e.g., in 3TJO) in the asymmetric unit, the same trimeric arrangement as for the wild type is seen between the crystallographically related molecules, where e.g., three monomers "A" of the crystal structure form the canonical trimer, and three monomers "B" etc. When comparing the trimers of the mutants and the wild type, the monomer positions are very similar, with the largest differences at the outer parts of the monomers, indicating a slight rotational movement around the center of the trimers. The averaged buried surface area for the two trimers of the R274Q mutant is 908 Å$^2$, for the three trimers in each of the two double mutants 925 Å$^2$ and 957 Å$^2$, and for the wild type (PDB ID 3TJO) 947Å$^2$ (calculated with PISA, CCP4 suite[58]).

The coordinates and structure factors have been deposited with the RCSB with the accession codes 6Z0X (D174R R274Q), 6Z0E (R274Q) and 6Z0Y (D174R R274Q twinned).

### Computational modeling and chemistry

Methods related to computational modeling and GCP chemistry are described in Supplementary Information. The input for MD simulations and the coordinates of starting and ending structures are provided as Supplementary Data 4-13.

### Statistical analysis

Statistical analysis was performed as described in the figure legends.

### Reporting summary

Further information on research design is available in the Nature Portfolio Reporting Summary linked to this article.

## Data availability

The structures of HTRA1 R274Q and D174R R274Q have been deposited in the Protein Data Bank (PDB) with the accession codes 6Z0E, 6Z0X and 6Z0Y. Mouse brain vessel proteomic data have been deposited to the ProteomeXchange Consortium via the PRIDE[54] partner repository with the dataset identifier PXD024683. The processed mouse brain vessel proteomic data are provided in Supplementary Data 1-3. Uniprot is publicly available at https://www.uniprot.org. The input for MD simulations and the coordinates of starting and ending structures are provided as Supplementary Data 4-13. Source data underlying Figs.1, 2, 3, 4, 5, 6, Supplementary Fig. 1, 2, 5, 6, 7, 8, 9, 10, 11, 12, 13, 14, 15 and Supplementary Table 1 are provided as a Source Data file. Source data are provided with this paper.

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

## Acknowledgements

We thank P. Binz, A. Fasano, A. Gerhard, M. Schneider, and A. Weingart for their technical support. We thank the beamline staff of X10SA at the Swiss Light Source Paul Scherrer Institute, Villigen, CH, for support, and our colleagues of MPI Dortmund and the TU Dortmund for help with the data collection. Mouse drawings in Figs. 3a, 4a and 4c were adapted from "Mouse (lateral 2)" by BioRender.com (2023), retrieved from https://app.biorender.com/biorender-templates. This work was supported by Deutsche Forschungsgemeinschaft (EH 100/18–1, SFB1093 and SFB1430 – Project-ID 424228829 to I.V., C.B., E.S.-G., C.S., M.Ka. and M.E., DI 722/13-1 to M.D. and BE 6169 1-1 to N.B.), the Fondation Leducq (Transatlantic Network of Excellence on the Pathogenesis of Small Vessel Disease of the Brain to M.D.), EU Horizon 2020 project SVDs@target (grant agreement N°666881 to M.D.), Munich Cluster for Systems Neurology (EXC 2145 SyNergy – ID 390857198 to M.D.) and the Vascular Dementia Research Foundation (to M.D.). This work was also funded by Deutsche Forschungsgemeinschaft through Munich Cluster for Systems Neurology (EXC 2145 SyNergy – ID 390857198, to M.D. and S.F.L.), by Bundesministerium für Bildung und Forschung through CLINSPECT-M (FKZ161L0214C to M.D. and S.F.L.) and Bundesministerium für Bildung und Forschung for grant JPND PMG-AD (to S.F.L.). E.S.-G. was supported by Germany's Excellence Strategy – EXC 2033—390677874—RESOLV and acknowledges a Plus-3 grant of the Boehringer Ingelheim Foundation and the Gauss Centre for Supercomputing e.V. (www.gauss-centre.eu) for providing computing time on the GCS Supercomputer JUQUEEN at Jülich Supercomputing Centre (JSC). The IMP is supported by Boehringer Ingelheim.

## Author contributions

N.B., L.I., M.M., A.F., J.S., J.R., A.R. and B.H. designed, performed and analyzed biochemical experiments. Y.B.R.-B., K.B.-R., J.M.-P. and E.S.-G. carried out modeling, designed, analyzed and wrote the computational results. S.U. and M.Z. performed and analyzed SEC, SEC-MALS and AUC experiments. D.P., T.R. and M.Ka. performed nMS experiments. D.P. and M.Ka. designed and analyzed nMS experiments. I.R.V. and P.S. performed X-ray crystallography. B.W. and W.W. generated mutant mouse lines. N.B. and K.T.-V. performed and analyzed animal experiments. A.S. performed MS experiments. N.B., A.S., S.A.M., and S.F.L. designed and analyzed MS experiments. R.M. performed statistical analysis. T.N., M.Ku., A.Z., X.-Y.H., C.S. (deceased) and M.G. designed and synthesized compounds. C.B. designed, performed and analyzed NMR experiments. S.G.B. performed kinetic analyses of enzymatic data. N.B., L.I., M.M., S.G.B., T.C., R.H., M.G., M.E. and M.D. analyzed data. M.E. and M.D. designed the study. N.B., M.M., L.I., M.E. and M.D. wrote the manuscript with contributions from all authors.

## Competing interests

The authors declare no competing interests. Carsten Schmuck is deceased.

## Additional information

[1]Institute for Stroke and Dementia Research (ISD), University Hospital, Ludwig Maximilian University of Munich, Munich, Germany. [2]Center of Medical Biotechnology, Faculty of Biology, University Duisburg-Essen, Essen, Germany. [3]German Center for Neurodegenerative Diseases (DZNE), Munich, Germany. [4]Graduate School of Systemic Neurosciences (GSN), LMU Munich, Munich, Germany. [5]Organic Chemistry, Faculty of Chemistry, University Duisburg-Essen, Essen, Germany. [6]Center of Medical Biotechnology, Faculty of Chemistry, University Duisburg-Essen, Essen, Germany. [7]Max-Planck-Institute of Molecular Physiology, Dortmund, Germany. [8]School of Biochemistry, University of Bristol, Biomedical Sciences Building, Bristol, UK. [9]Department of Biochemical and Chemical Engineering, Technical University Dortmund, Dortmund, Germany. [10]Max-Planck-Institute of Biochemistry, Martinsried, Germany. [11]Institute of Developmental Genetics (IDG), Helmholtz Zentrum München, Neuherberg, Germany. [12]Neuroproteomics, School of Medicine, Klinikum rechts der Isar, Technical University of Munich, Munich, Germany. [13]Munich Cluster for Systems Neurology (SyNergy), Munich, Germany. [14]Technische Universität München-Weihenstephan, Freising, Germany. [15]College of Material Science and Technology, Nanjing University of Aeronautics and Astronautics, Nanjing, China. [16]Research Institute of Molecular Pathology (IMP), Vienna, Austria. [17]These authors contributed equally: Nathalie Beaufort, Linda Ingendahl. [18]Deceased: Carsten Schmuck. [19]These authors jointly supervised this work: Michael Ehrmann, Martin Dichgans. ✉e-mail: Michael.Ehrmann@uni-due.de; dichgansmartin@gmail.com

