## [Peer Review File · Nature Communications]

Rational correction of pathogenic conformational defects in HTRA1REVIEWER COMMENTS

Reviewer #1 (Remarks to the Author):

CARASIL is an inherited vascular dementia caused by loss-of-function mutations of HTRA1 whose trimeric assembly is essential for its protease activity. Authors chose a prototypical disease-causing and trimerization defect mutant, R274Q as a model. First, they indicated that both trimer formation and protease activity of R274Q were restored in vitro by mixing with the other protease defective mutant, D174R-S328A, which had a compensatory mutation at the protomer interaction surface and stabilized trimer formation. Second, they indicated that R274Q mutant was rescued by D174R-S328A mutant in vivo by making knocked-in mice of each mutant and mating them. Since loss-of-function mutant of Htra1 in mice have no clear phenotype, they showed the restoration of Htra1 protease activity by quantitative mass spectrometry (MS) analysis and degradation of substrates of Htra1. Next, they indicated that supramolecular ligands, GCPs increased protease activities of trimer defective mutants by stabilizing trimer formation. Finally, they found that peptides derived from VDAC2 and VDAC3 allosterically activated monomeric Htra1 mutants without shifting to trimer conformation. Their trials to correct prototypical disease-causing HTRA1 mutants by proteins, GCPs and, peptides are very novel. They succeed in indication that those approaches are effective not only in vitro but also in vivo by utilizing various kinds of experiments. However, several points are not clearly shown.

1) Although R166H, A173T, and R274Q were all trimer defect mutants, their protease activities vary as shown in Fig. 1 b and d. Authors should explain the reason why the difference was generated. Is that explained by the data in Sup Fig.1 e? If so, more detailed explanations should be added.

2) In Fig.2c, left side panel, the bands of His protein without GA at the bottom of each lane are not equal, although an equal concentration (0.5 μ M) of R274Q was used in all experiments. The left-most lane in which no protein was loaded showed a band. Authors should explain the reason.

3) In Fig.3d and e, the size of intact and cleaved LTBP4 are different.

For Fig.3d, loading control such as laminin and the explanation about the band around 150kDa should be added. And some quantitative analysis of LTBP4 bands such as intact or cleaved ones is required.

4) Comment 4) also applies to Fig.4e.

5) In Fig.5c, activation of R166H mutant was lower than the other two mutants.

Some explanations about that should be added.

6) In Fig.5, the authors showed evidence that GCPs corrected the conformation of the mutants only by NMR DOSY experiments. However other experiments such as gel filtration or non-denaturing electrophoresis to show trimerization of mutants may be possible. A252T which forms trimer should be used as a negative control, showing DCPs do not activate its protease activity.

7) In Fig.6b, description about – and + are necessary.

8) In Fig.6a, the degree of activation of mutants by VDAC2 or VDAC3 peptides were different. The explanation about the reason should be added.

I wonder how the peptides affect the activity of A252T mutant.

9) Is there any biological significance that the peptides derived from VDAC2 and VDAC3 allosterically activate wt HtrA1? Some description about that should be added.

10) In this study, all HtrA1 proteins used in in vitro experiments have deletion in the N-terminal (~152) domain. Authors should discuss if the N-terminal domain affects trimerization and protease activity.

Reviewer #2 (Remarks to the Author):

The manuscript by Dichgans and coworkers looks into a structural mutation in serine protease HtrA1 that has been identified in cerebral vasculopathy.

The authors have looked into the mutation both from structural and functional perspectives to understand the cause for its pathogenicity using in vitro and in vivo tools. They have also devised strategies to rescue this phenotype using structure-guided design and allosteric modulation.

The manuscript is reasonably well-written and provides an interesting approach to devise therapeutic strategies against diseases where HtrA1 oligomerization might play a prominent role.

However, there are few important concerns that need to be addressed.

1. The protein is shown as a monomer (one very important aspect of the manuscript) using crosslinking studies. Other independent studies such as size exclusion chromatography and most importantly Analytical ultracentrifugation should be done. This will also give information about the stability of the trimeric ensemble, with a precise kd value as well as the stoichiometry of monomeric and trimeric species.

2. One of the rescue experiments is based on the protein's allosteric property. However, the manuscript does not provide any enzyme kinetics studies of HtrA1 wt versus the mutant (residual activity is present according to the manuscript). This, along with structural studies might be important toward providing a proper structural roadmap for understanding of its conformational dynamics that leads to loss of activity.

3. Although, the authors have said that HtrA1 R274Q mutant is a monomer, they found some activity in the mutant. No explanation has been provided whatsoever for this observation.

4. Most importantly, the disease is extremely rare and is multi-factorial; HtrA1 mutations might not be the only or prime reason. While, choosing the mutation, the authors did not provide any rationale behind picking this particular mutation. Moreover, how many other HtrA1 mutations are found in the same disease, their positions, their frequencies, ethnic backgrounds of the patients, and their oligomeric properties with relevant references (a table would be helpful) is important. Furthermore, what is the percentage of patients with cerebral vasculopathy who do not harbor any HtrA1 mutation? How different is the severity between these two subsets of patients? In this respect, I would consider the manuscript to be a quite incomplete and lacks significance of its further clinical applications. Moreover, this information should be provided to help understand how general their approach would be to rescue the pathogenic phenotype when mutations can be any.

Reviewer #3 (Remarks to the Author):

The manuscript titled "Rational correction of pathogenic conformational defects in HTRA1" by Beaufort et al, describes a broad study aimed at correcting effects of the R274Q

substitution in the human HtrA1 protease both in vitro and in vivo. This substitution is a member of a group of the HtrA1 substitutions inactivating the protease and linked to cerebral autosomal dominant arteriopathy with subcortical infarcts and leukoencephalopathy (CARASIL).

The authors found that the loss-of-function mutations cause changes at the interface of the protomers of the HtrA1 trimer which prevent trimerization and thus lead to the enzyme inactivity.

They set out to achieve the functional correction of the mutated variant by several approaches. Firstly, they identified a compensatory substitution (D174R) which suppressed inactivity of the R274Q variant in cis by restoring the enzyme ability to trimerize.

Interestingly, they were able to show that the suppression was possible when the compensatory substitution was present in trans, i. e. in another HtrA1 protein. The suppression was documented in vitro by showing that the enzyme was active and formed trimers. Moreover, the suppression in trans worked also in vivo, in the mouse model. Since the HtrA1 deficient mice have no obvious phenotype, including no CARASIL symptoms, the Authors monitored HtrA1 activity by comparing proteomes of mouse brain vessels using the genetically-edited mice (by CRISPRCas9 method). This rather laborious approach showed many differences between the wt and mutant mice, especially among the extracellular proteins. The HtrA3 candidate substrates, LTB4 and PRSS23 proteins, served as indicators to monitor the effects of supplementing the mice carrying the R274Q HtrA1 protein with the compensatory D174R variant. Indeed, the HtrA1 D174R efficiently suppressed the R274Q mutation. This was in my opinion a very elegant experiment showing complementation of disease-causing protein by another protein.

Secondly, as protein delivery in humans poses many problems, the Authors searched for other ways of correcting the defective HtrA1. One approach was to use supramolecular ligands to induce trimerization of the R274Q HtrA1. Another one was to find peptides which would allosterically activate the monomeric R274Q variant. In both cases they were able to identify ligands which increased activity of the defective HtrA1 in vitro.

In conclusion, this work shows several ways of correcting malfunctioning disease-linked protein by influencing its conformation.

Concerns

1. In the experiment in which trimerization of the R274Q HtrA1 variant was induced by GCPs, relatively high concentrations (2.5 mM) of the ligands were used with 25 μ M HtrA1, which gives the protein/ligand ratio of 1:100. As the general aim of the work was to design ways of correcting malfunctioning proteins in humans, would the Authors comment on this? Could such excess of a therapeutic ligand be achieved? Secondly, the GPC ligands bind oxyanions, so they may bind a lot of proteins, not only HtrA1. The specificity problem should be discussed.

2. The Authors used peptidic modulators to correct function of the inactive HtrA1 variants. The best of them, the VDAC2 efficiently stimulated activity of the mutant HtrA1 variants against purified tau protein. However, it also stimulated very efficiently activity of wt HtrA3 and, on the other hand, inhibited trypsin and chymotrypsin activity. This again raises a question of the ligand specificity. This should be pointed out and discussed.

3. The Authors did a lot of work in silico to characterize interaction of the VDAC2 activating peptide with the HtrA1 R274Q and wt proteins, found places of the ligand binding and concluded that VDAC2 is an allosteric activator binding mainly to the active site and alternative site situated at the trimer tip. However, the Authors should support their in silico results by kinetic experiments showing that VDAC2 is indeed an allosteric activator. This could be easily done, since fluorogenic substrate peptides for HtrA1 are available.

4. In my opinion, the model of HtrA1 activation (shown in the Extended Data Fig.10) should be placed in the main text, possibly as an additional panel of Fig. 6. This would help the general readers who may not be acquainted with the scheme of the HtrA proteins regulation to understand the text without going to the ExtendedData part.

5. Fig. 3d and Supplementary Fig. 4d. What does the 150 kDa band represent? Is this a degradation product of LTB4?

Reviewer #4 (Remarks to the Author):

The manuscript by Ehrmann, Dichgans and coworkers shows that certain loss-of-function mutants in the homotrimeric serine protease HTRA1, known to cause cerebral vasculopathy, affect the stability of the trimer by perturbing interactions across the protomer-protomer interface. The authors show that the enzymatically active trimer can be reconstituted by either introducing compensating mutations that stabilize the trimer, or binding protomer-

bridging low-molecular weight ligands, or binding target-derived peptides at an allosteric site. The compensating mutations can be introduced in cis or trans, as shown by in vitro experiments.

Using transgenic mice, the authors proceed to show that the same approach generates enzymatically active trimers in vivo when presented in the context of heterozygous animals containing the disease-causing mutant R274Q on one allele and the compensatory (and inactive) mutant D174R-S328A on the other. The manuscript represents a considerable amount of work involving, among other techniques, sophisticated genetic approaches and quantitative MS analyses comparing the proteome in brain vessels of wild-type and mutant mice.

Still, I am not at all convinced that this contribution will have any major impact for several reasons.

1. The work showing, in vitro and in vivo, that interactions between protomers are important for enzymatic activity is solid, but is an expected result in line with previous reports in the literature.
2. The authors highlight the suggested protein-based functional correction as a potential therapeutic strategy. It is entirely unclear to me why the proposed mutant D174R-S238A would offer a suitable genetic therapy of disease; how does introducing this mutant lead to improvements over the alternative to perform gene-editing back to the wild-type? What am I missing here?
3. It seems to me that the low-molecular weight ligands represent a more interesting approach, but these were not characterized in vivo. Would it be possible to test whether these ligands are effective treatments in the mouse model?
4. I also wonder why the in-vivo work did not use the active D174R mutant (rather than the inactive D174R-S238A)? The combinatorics of trimer formation predict that the active D174R mutant would rescue the R274Q mutant to yield 3/4 of the (homozygous) wild-type activity, whereas the rescue achieved by the inactive D174R-S238A would yield only 1/4

(assuming that each D174R or R274Q protomer is active in the trimer). Along the same lines, introducing the wild-type in one allele would yield 1/2 of the activity in non-diseased individuals.

5. In connection to point 4, it would be of interest to know the variation between mutants (and between mixtures of mutants) in trimer stability. I suggest that you measure the stability by biophysical methods.

6. Further, it would be of interest to measure the activity (in terms of LTBP4-Nt and PRSS23-Ct levels) in wt/R274Q mice, thereby addressing the issue of trimer combinatorics.

Other points (roughly in order of appearance):

The title claims "Rational correction..." but the chosen corrective mutant does not seem to behave as one might have naively expected (or rationally designed): the introduced arginine (R174) reaches over to the neighboring protomer where it forms stabilizing van der Waal's interactions, rather than re-establishing hydrogen bonding interactions to Q274. To this extent, the stabilizing effect of the chosen mutant seems rather serendipitous.

line 69: "...several loops that are shared between protomers." How can a loop be shared between protomers?

lines 153 and 215: the changes in levels of HTRA1 seem to suggest that the R274Q/D174R-S328A construct has 11/7.5 higher levels than wild-type(?) If correct: How do you explain this result?

Figs. 5d and 6c: I do not find that these plots provide convincing evidence of significant differences between the different cases, in part because it is hard to discern the 25 uM data among the different gray shades. Also please present for each case the extracted translational diffusion coefficient and its estimated error.

line 324-5: "...overall remodeling of the protomer-protomer interface, rather than pairing

with a defined partner residue..." This wording does not agree with Fig. 2, which shows a modest change in the orientation of just a few side chains.

line 338-40: Please explain why the inactive mutant is better than the active one (or the wild-type). Are you suggesting that the homozygous wild-type is causing detrimental off-target effects?

Reviewer #5 (Remarks to the Author):

In this manuscript, authors describe that certain disease causing mutations in the homotrimeric serine protease HTRA1 leads to loss of enzymatic activity by impairing multimeric assembly of HTRA1. Using a genetic approach, the authors show that a corrector variant is active in trans and rescue the molecular phenotype in vivo. Concurrently, authors developed two additional strategies based on small molecules to correct the activity of the mutants. On one side, the authors describe small-molecule stabilizers based on guanidinocarbonyl pyrroles that restore multimeric assembly of HTRA1 variants. The second approach describes short peptides that allosterically activate HTRA1 wt and variants in the monomeric form. Overall, the paper reports interesting observations and approaches, but not enough explanations in terms of biophysics and molecular interactions.

-It is not clear how the compensatory mutation (D174R) actually works. The authors mention multiple van-der-Waals contacts that could result in an overall increased trimer interface but these contacts are not specifically identified nor shown in Fig. 2a.

-Fig.5a is vague and we do not understand how GCP actually brings the two monomers together. It is unclear whether the image shown in Fig 5a is based on molecular modelling. If yes what are the key residue on the surface of HTRA1 that would interact with the GCP ligands. In the absence of additional evidence that the molecule is actually binding as suggested there is no reason why to suggest a particular mode of binding.

-The representations proposed for the interaction between VDAC2 and HTRA1 R274Q in Fig 6d/e could be improved by using surface representation for the protein, by using different colours for the side chain of the peptide and the protein as well as orthoscopic view.

Biophysical experiments (including affinity measurements) are needed to confirm the interaction between HTRA1 R274Q and VDAC2 as well as confirm the limited or absence of

interaction of HTRA1-R274Q with VDAC2-2 or VDAC2-3. Authors do not show experimental evidence for the location of the second binding site for allosteric activation. A mutant of wt HTRA1 that would incorporate a mutation in the putative alternative site should abrogate activation by VDAC2 but not by VDAC2-2 and would help validating this hypothesis. This would provide further evidence for a second binding site and this hypothesis could be reinforced by evaluating the effect of the mutation in HTRA1-R274Q

-Other points to be addressed :

1- How frequent are the mutations selected in this work among pathogenic HTRA1 mutations ? This information could be added in the text.

2- Isothermal titration calorimetry (ITC) could prove useful to further compare the ability of wt HTRA1 and mutants to self-assemble. This could be achieved by titrating a concentrated solution of HTRA1 into the same buffer solution, the calorimeter would then measure the heat variation upon shifting the monomer-oligomer equilibrium towards the monomer.

3- Modifications of the tris(2-aminoéthyl)amine linker (e.g. without the free amino group, or with longer or more rigid chain) in the MK series would help to gain information about the molecular determinants for bridging HTRA1 monomers.

4- The authors mention that GCP actually compare favourably with other compounds tested (p10, line 252 of the manuscript) but it looks as if compound MK99 with two Arg is as effective as MK1 when looking at Supplementary Fig. 5b. Authors should explain this discrepancy or remove the statement. The concentration of compounds tested should be reported in the legend of supplementary Fig. 5 for better comparison.

5- Molecular approaches aiming at stabilizing protein-protein interactions is expanding and useful literature is worth being cited on this specific topic (see for example Andrei et al. *Exp. Opin Drug Discov* 2017, 12, 925-940)

6- The legend for the two lines with + and - is missing in Fig. 6b

7- Copies of HPLC traces and MS spectra should be provided for compounds MK1-MK3 as well as control compounds shown in Supplementary Fig. 5.

8- I could not find details about the purity of peptides used to study allosteric activation of HTRA1. This information should be provided.

9- SI (p26, 27 and 29) please correct Fomc in Fmoc.

Reviewer #6 (Remarks to the Author):

Beaufort and colleagues report the biochemical defect, i.e. lack of trimerization, of naturally occurring mutations in the serine protease HtrA1 causing cerebral vasculopathy. They develop several strategies to rescue this defect. This includes the biological restoration by introducing a single mutation in HtrA1-R274Q, which was convincingly validated in mouse studies. As this strategy remains therapeutically challenging, the authors describe two alternative strategies to correct the pathogenic R274Q mutation, i.e. the identification of trimer-stabilizing supramolecular ligands and of allosteric activators. While these latter approaches restore enzymatic activity of HtrA1 mutants, they still lack sufficient specificity and in-vivo validation. Nevertheless, the presented work represents a stimulating and creative work to correct the pathogenic effects of Htra1 trimer interface mutations. However, there are some important shortcomings that need to be addressed. Among these are the misleading claim that the authors are the first to unravel the biochemical defects of the interface mutants, which in fact have been described previously by two publications, Nozaki H. et al (Ref 19) and Uemura M et al. (Ref 20). In addition, the methods used to show this defect and the rescue by ligands are not straightforward and need to be supported by additional experiments (SEC). Additional comments that need attention are detailed below.

Major Comments:

1.The authors brush over important published studies which have identified the trimerization defect for some of these mutants, including R274Q. Nozaki 2016 did elegant SEC experiments to show that R274Q as well as A252T exist as monomers and trimers, respectively (and w/o any A252T hexamers, which were shown in fig.1d by x-linking), while Uemura 2016 investigated R166C, R166L and A173T. The authors convey the perception that they have identified the mechanism for the first time, which is misleading and incorrect. This entire section needs revision and published data need to be explicitly quoted and their confirmatory data shown here need to go to supplemental section. Accordingly, the claim in the discussion section "we unraveled how a set of point mutations causing ...disrupt protease function" needs modification; the same goes for the abstract.

2.page 4. It was surprising to see that the authors decided to carry out chemical x-linking

and NMR to determine the oligomeric states. SEC or SEC-MALS seemed straightforward and accurate and has been used by other authors. Therefore, I would strongly encourage the authors to further validate the oligomeric state of some key HtrA1 mutants using SEC or SEC-MALS. This should also be applied with the identified MK supramolecular ligands and the allosteric activators.

3. page 11: why did the authors not stay with the casein cleavage assay, but instead switched to Tau? Did the casein assay give similar results? While the effect of these peptides is impressive (albeit surprising since they don't require the PDZ domain), the non-selective character is a significant obstacle in terms of therapeutic potential: Htra3 is activated and trypsin is inhibited. This may just be the tip of the iceberg regarding unwanted effects on the human proteome and needs to be better discussed in terms of therapeutic applications and limitations. The inhibitory activity on trypsin should raise major concerns to the authors in regards to proposing a clear mechanism by this compound: the Htra3 effect could be understood in terms of its similarity to htra1, but this doesn't hold for trypsin where the opposite effect (inhibition) is observed. In addition, the finding that the peptides act on the monomer is surprising and needs to be firmed up by a SEC (-MALS) experiment. The question on how the compound is able to activate both, the wt trimeric form AND the "inactive" mutant monomer, should be better discussed.

4. Affinities of individual peptides: The affinities of the most important ligands for HtrA1 should be quantified. For the MK peptides 2.5mM final peptide concentrations were used in enzyme assays, which is extremely high and indicates a very low affinity. However, the VDAC peptides should have a measurable affinity towards the HtrA1 monomer R166H, A173T or R274Q. Similarly, the contribution of different binding sites could be probed in regard to the mutants VDAC2-2 and VDAC2-3 further strengthening the different binding sites (active site vs. alternate site).

5. GCP supramolecular ligands: The authors demonstrate that bi-functional GCP supramolecular ligands can restore the oligomeric state of HtrA1 R274Q mutant. It would be important to show that a monovalent GCP molecule with a single GCP headgroup would fail to restore the trimerization. The authors only show branched monovalent GCP molecules

(AZ21 and AZ25) but not the direct derivative of MK1-3 with only a single GCP group. It might be possible that binding of GCP alone induces trimerization through structural rearrangement of the underlying unidentified binding site.

Minor Comments:

1. Protease cleavage assay: It would be helpful to show the levels of HtrA1 or the full uncropped gels in the supplement, to validate equal loading and activity across the HtrA1 wildtype and mutants. Are the concentrations of htra1 and mutants based on the monomer concentrations? This is important since some of the mutants, especially the R274N are in the monomeric state and a comparison of their activities needs to be based on the monomer concentration.

2. Htra1 constructs used: this needs better explanation. It took me a while to go through the cited literature to find out what constructs were used. I believe it is mainly the protease domain-PDZ construct and the protease domain construct. The cartoon in fig.1a represents the full length Htra1 (the mac25 domain should rather be called the IGFBP/Kazal-like domain), rather than any of the constructs used herein. This would be the ideal place for showing the cartoons of the constructs used.

3. page 3/line70: the sensor loop statement needs references.

4. page 5: what exactly was the rationale for designing the compensatory mutation? The authors invoke the Xray structure (fig.2a) as the rationale, but this structure was done based on the rationale/design, which remains obscure. Please clarify.

5. page 6, line 140: please show the vdW contacts of R174 in figure 2a; from the figure it looks like it contacts a positively charged patch, which cannot be true.

6. page 10: Please define the "inter protomer gap" more precisely. The fig.5a gives no information what we are looking at: please label some key residues. An additional ribbon model of the trimer would help in understanding the location and how this site is related to

the compensatory mutant?

7. page 11: the rationale of screening a C-terminal peptide library is totally obscure. I presume it was based on the assumption that these peptides would bind the PDZ domain to allosterically activate htra1? However, they seem to act in the absence of the PDZ domain. This is rather confusing and needs to be clarified.

8. Fig. 6a: The amounts of VDAC2 and VDAC3 in the Tau cleavage assays are not clear. Also, a detailed description of the Tau cleavage assay in presence of VDAC2/3 or other activators (like the MKs and casein) is missing in the Methods section.

9. Fig. 6b: I believe the labels +/- VDAC2 and +/- crosslinker are missing.

10. Fig. 6c: The R274Q mutant is missing the labels for the Y-axis.

11. Material and Methods: Please provide buffers whenever appropriate. No buffers have been listed in the M&M part for purifications, proteolytic cleavage assays or NMR experiments. "2N4R Tau was purified in house as described below", but there is no purification method shown.

Rational correction of pathogenic conformational defects in HTRA1

Nathalie Beaufort^{1,a}, Linda Ingendahl^{2,a}, Melisa Merdanovic², Andree Schmidt^{3,4}, David Podlesainski², Tim Richter², Thorben Neumann⁵, Michael Kuszner⁵, Ingrid R. Vetter⁶, Patricia Stege⁶, Steven G. Burston⁷, Anto Filipovic², Yasser B. Ruiz-Blanco², Kenny Bravo-Rodriguez^{2,6}, Joel Mieres-Perez^{2,13}, Christine Beuck², Stephan Uebel⁸, Monika Zobawa⁸, Jasmin Schillinger², Rainer Malik¹, Katalin Todorov-Völgyi¹, Juliana Rey², Annabell Roberti², Birte Hagemeyer², Benedikt Wefers^{3,9}, Stephan A. Müller^{3,10}, Wolfgang Wurst^{3,9,11,12}, Elsa Sanchez-Garcia¹³, Alexander Zimmermann⁵, Xiao-Yu Hu¹⁴, Tim Clausen¹⁵, Robert Huber^{2,8}, Stefan F. Lichtenthaler^{3,10,11}, Carsten Schmuck⁵, Michael Giese⁵, Markus Kaiser², Michael Ehrmann^{2,b,*}, Martin Dichgans^{1,3,11,b,*}

Point-to-point replies to the comments of the reviewers

REVIEWER 1:

CARASIL is an inherited vascular dementia caused by loss-of-function mutations of HTRA1 whose trimeric assembly is essential for its protease activity. Authors chose a prototypical disease-causing and trimerization defect mutant, R274Q as a model. First, they indicated that both trimer formation and protease activity of R274Q were restored in vitro by mixing with the other protease defective mutant, D174R-S328A, which had a compensatory mutation at the protomer interaction surface and stabilized trimer formation. Second, they indicated that R274Q mutant was rescued by D174R-S328A mutant in vivo by making knocked-in mice of each mutant and mating them. Since loss-of-function mutant of Htra1 in mice have no clear phenotype, they showed the restoration of Htra1 protease activity by quantitative mass spectrometry (MS) analysis and degradation of substrates of Htra1. Next, they indicated that supramolecular ligands, GCPs increased protease activities of trimer defective mutants by stabilizing trimer formation. Finally, they found that peptides derived from VDAC2 and VDAC3 allosterically activated monomeric Htra1 mutants without shifting to trimer conformation. Their trials to correct prototypical disease-causing HTRA1 mutants by proteins, GCPs and, peptides are very novel. They succeed in indication that those approaches are effective not only in vitro but also in vivo by utilizing various kinds of experiments. However, several points are not clearly shown.

AUTHORS: We thank the reviewer for the overall positive assessment of our manuscript and the helpful comments. **Page and line numbers refer to the clean version of the revised text.**

1) Although R166H, A173T, and R274Q were all trimer defect mutants, their protease activities vary as shown in Fig. 1 b and d. Authors should explain the reason why the difference was generated. Is that explained by the data in Sup Fig.1 e? If so, more detailed explanations should be added.

AUTHORS: In response to the comments we now performed additional experiments and go into more detail in terms of explaining the results: First, we quantified HTRA1 activity against β -casein. Second, we evaluated HTRA1 monomer/dimer/trimer stoichiometry by independent approaches including size exclusion chromatography (SEC), SEC multi-angle light scattering (SEC-MALS), native mass spectrometry (nMS) including titration assays, as well as sedimentation velocity analytical ultracentrifugation (AUC).

As illustrated in the revised Fig. 1d-e and in Supp. Fig. 1b-e, R274Q, R166H and A173T shift the monomer-trimer equilibrium towards inactive monomers but some (proteolytically active) trimers exist. These trimers are the basis for the residual levels of enzymatic activities detected. Mutant A173T exhibited the lowest relative trimer abundance (Supp. Fig. 1d-e), in accord with its markedly reduced activity compared to R274Q and R166H (Fig. 1c).

To clarify, we expanded on the text (see Results, page 4-5, lines 100-103 and 106-113):

“All three interface mutants were mostly detected as monomers in size exclusion chromatography (SEC), SEC multi-angle light scattering (SEC-MALS) and chemical crosslinking, while wt HTRA1 and A252T were trimers (Fig. 1d, Supp. Fig. 1b-c and Supp. Fig. 3a).”

“Native mass spectrometry (nMS) experiments, including titration assays, as well as sedimentation velocity analytical ultracentrifugation (AUC) further indicated that in R274Q, R166H, and A173T but not in A252T the monomer-trimer equilibrium was shifted towards monomers (Fig. 1e, Supp. Fig. 1d-e). The small amounts of trimers account for the residual levels of enzymatic activity detected in interface mutants. A173T exhibited the lowest relative trimer abundance (Supp. Fig. 1d-e) in accord with its markedly reduced activity compared to R274Q and R166H (Fig. 1c).”

Moreover, we scaled up and optimized the production of recombinant HTRA1 proteins, including an additional hydroxyapatite column (see Methods, page 2-4, lines 66-104). Under these conditions, A173T (for which we obtained lower protein amounts, suggesting folding

issues) exhibited a reduced protease activity compared to our previous dataset. The residual activity of R274Q, R166H and A252T was comparable to our previous dataset.

2) In Fig.2c, left side panel, the bands of His protein without GA at the bottom of each lane are not equal, although an equal concentration (0.5 μ M) of R274Q was used in all experiments. The left-most lane in which no protein was loaded showed a band. Authors should explain the reason.

AUTHORS: We would like to apologize for a labeling error in the left gel where the most right-hand lane should read “-” (upper) “0.5” (lower lane), i.e. labeling of both gels should be identical. The revised panel and legend are now presented in Supp. Fig. 3b:

Supp. Fig. 3b: Following incubation of R274Q in the absence or presence of D174-S328A, the oligomeric states of each protein were determined via chemical crosslinking and subsequent immunoblot analysis using anti-His and Strep antibodies. Note that the anti-His antibody recognizes the Strep tag (with Strep-tagged HTRA1 running slightly lower compared to His-tagged HTRA1), however only in the uncrosslinked protein.

As stated in the Figure legend, the anti-His antibody recognizes the Strep Tag (with Strep-tagged HTRA1 running slightly lower compared to His-tagged HTRA1), explaining the increase in band intensities. Please note that this is only true for the uncrosslinked protein. Following cross-linking, the issue does not exist as in the most right-hand lane of the +GA part of the left-hand gel the Western blot with the anti His antibody does not show any bands.

3) In Fig.3d and e, the size of intact and cleaved LTBP4 are different. For Fig.3d, loading control such as laminin and the explanation about the band around 150kDa should be added. And some quantitative analysis of LTBP4 bands such as intact or cleaved ones is required.

In accord with the reviewer’s suggestion, we included **a loading control:** actin, a suitable control for the analysis of full vessel lysates including cytosolic fractions, in Fig. 3d (see also the Figure below).

Concerning the **150 kDa band**, LTBP4 typically migrates as multiple bands on immunoblot analysis of mouse tissues (see¹ presenting analysis of tissues from *LTBP4^{wt}* and *LTBP4^{KO}* mice and using the anti-LTBP4 Ab AF2885 from R&D Systems, which we also used in our work). The exact identity of individual bands is unknown. They most likely account for post-translational modifications (e.g., glycosylation, proteolytic processing) and/or splice variants^{1,2}.

As requested, we **quantified the immunoreactive LTBP4 species** (see Source Data and the Figure below). High molecular weight LTBP4 species (L1 and L2) are enriched in *Htra1*^{R274Q} and *Htra1*^{D174R-S328A} vessels compared to *Htra1*^{wt} vessels. This accumulation is attenuated in *Htra1*^{R274Q/D174R-S328A} vessels.

related to
Fig. 3d

related to
Fig. 4e

related to
Supp. Fig. 9d

Quantification of LTBP4 bands detected by IB in mouse brain vessels. LTBP4 was detected by immunoblot in mouse brain vessels. Actin serves as loading control. Images duplicate the data presented in Fig. 3d, Fig. 4e and Supp. Fig. 9d. *: sample not depicted in Fig. 3d. Band intensity was determined using Image J and data are presented as box-and-whisker plots (median, minimum, maximum) with filled circles. Data points represent individual mice; significance was tested by two-sided unpaired *t*-test.

To account for the reviewer's comments, we adapted the text (see Results, page 7, lines 180-182):

“Accordingly, immunoblots indicated an accumulation of high molecular mass species of LTBP4 in Htra1^{R274Q} compared to Htra1^{wt} vessels.”

We further revised the legend to Fig. 3d:

“Detection of LTBP4 by immunoblot (IB) in brain vessels. Actin serves as loading control. Biological replicates are shown. The multiple LTBP4 bands most likely account for post-translational modifications and splice variants. Black arrowheads: LTBP4 species enriched in Htra1^{R274Q} compared to Htra1^{wt} vessels (see Source Data for quantification).”

Using conditioned medium from HEK cells overexpressing LTBP4, we confirmed that the **size of mouse brain vessel-derived (Fig. 3d) and purified recombinant LTBP4 (Fig. 4e, R&D Systems)** are different. This difference most likely reflects post-translational modifications and/or splice variants.

4) Comment 4) also applies to Fig.4e.

AUTHORS: We included a loading control (actin), quantified LTBP4 bands and revised the legend to Fig. 4: “LTBP4 and actin detected by immunoblots.”

5) In Fig.5c, activation of R166H mutant was lower than the other two mutants. Some explanations about that should be added.

AUTHORS: The level of activation of HTRA1 is often affected by the basal activity of the mutant enzyme. In case of R166H, it is higher compared to A173T explaining the slightly lower levels of activation.

	wt	R274Q	R166H	A173T
Relative activity in the absence of GCPs [slope of β -casein degradation]	0.17 \pm 0.02	0.05 \pm 0.00	0.04 \pm 0.00	0.02 \pm 0.00
Fold activation by MK1	1.2 \pm 0.1	2.4 \pm 0.4	2.0 \pm 0.3	2.8 \pm 0.3
Fold activation by MK2	1.0 \pm 0.0	2.1 \pm 0.2	2.2 \pm 0.0	2.6 \pm 0.1

Activation of interface HTRA1 mutants (1 μ M) by MK1 and MK2 (2.5 mM). β -casein (20 μ M) was used as a substrate. Activity was determined as the slope of β -casein degradation in the linear phase (mean \pm SD of 2-4 experimental data). Average relative activity or fold activation \pm SD are depicted.

6) In Fig.5, the authors showed evidence that GCPs corrected the conformation of the mutants only by NMR DOSY experiments. However other experiments such as gel filtration or non-denaturing electrophoresis to show trimerization of mutants may be possible.

A252T which forms trimer should be used as a negative control, showing DCPs do not activate its protease activity.

AUTHORS: As suggested by the reviewer, we collected and analyzed **SEC fractions** by SDS-PAGE. In accord with NMR DOSY assays using R274Q and MK2, R274Q was shifted towards trimers in the presence of MK1 (Supp. Fig. 11a-b).

We updated our manuscript (see Results, page 10, lines 251-253):

“In addition, SDS-PAGE of SEC fractions revealed that the elution profile of R274Q was shifted towards trimers in the presence of MK1 (Supp. Fig. 11a-b).”

We also investigated the impact of MK1 and MK2 on the **protease activity of A252T**. Contrasting with the activation of the interface mutants, A252T was inhibited by both compounds (see revised Fig. 5b). We suspect that MKs stabilize the inactive conformation of HTRA1-A252T. To clarify the mechanism of inhibition, structural studies would be required.

We revised the text accordingly (see Results, page 10, lines 243-246):

“Wt HTRA1 was mildly affected, if at all (up to 1.2-fold change) while A252T was inhibited by both compounds. We suspect that MKs stabilize an inactive conformation of HTRA1-A252T by an as yet unknown mechanism.”

7) In Fig.6b, description about – and + are necessary.

AUTHORS: Following the reviewer’s suggestion we labeled the +/- in the Figure and expanded on the figure legend. The revised panel is now presented in Supp. Fig. 3d:

Supp. Fig. 3d: Oligomeric states of wt and mutant HTRA1s in the absence or presence of 50 μM VDAC2 peptide analyzed by chemical GA-based crosslinking.

8) In Fig.6a, the degree of activation of mutants by VDAC2 or VDAC3 peptides were different. The explanation about the reason should be added. I wonder how the peptides affect the activity of A252T mutant

AUTHORS: It is not surprising that sequence variations are causing different levels of activation (see peptides VDAC2-2 and -3). Because we feel this is a rather trivial fact, we did not add a comment to the manuscript, which is already very extensive.

We have performed additional enzyme kinetics studies using an 11-mer peptidic FRET substrate. We carried out VDAC2 titration assays at a fixed concentration of substrate (see Fig. 6d, Supp. Fig. 14a and revised description of the results, page 12, lines 293-301). We further performed substrate titration assays in the absence or presence of VDAC2 (see Supp. Fig. 14b and revised description of the results, page 12-13, lines 302-318). These new data exemplify that the affinity of a peptidic modulator such as VDAC2 for the enzyme as well as its impact on V_{max} and K_m are mutant- and substrate-specific.

We have characterized the effects of VDAC2 peptide on A252T. As this mutant behaves like a "weak wt", we feel that not much can be learned from these data. We therefore did not include them into the manuscript, for reasons of space and focus.

Impact of VDAC2 on the activity of A252T. **a**, Time-dependent degradation of Tau (3 μM) by A252T (1 μM) in the absence (Ctrl) or presence of VDAC2 (50 μM). **b**, Normalized cleavage rates of peptidic

FRET substrate DY649P-IRRVSYSFKKC-DYQ661 (2 μ M) by the A252T plotted vs. concentration of VDAC2. Error bars indicate SD of 6 experimental data. Data were fitted to the weak-binding equation to yield V_{max} and K_d (black line). **c**, Normalized cleavage rates of DY649P-IRRVSYSFKKC-DYQ661 by A252T in absence (black circles) and in presence of fixed VDAC2 concentrations (red circles) plotted vs. substrate concentrations. Error bars indicate SD of 4 experimental data. **c-d**, brackets indicate SE of the fit.

9) Is there any biological significance that the peptides derived from VDAC2 and VDAC3 allosterically activate wt HtrA1? Some description about that should be added.

AUTHORS: Following the reviewer's comment, we expanded on the Discussion (see page 15, lines 375-380):

"HtrA proteases are regulated by the mechanism of ligand-induced activation, which can be allosteric^{3,4,5}. In bacteria, such ligands include the C-termini of β -barrel outer membrane porins serving as signals of protein folding stress³. Therefore, the activation of HTRA1 by peptides derived from the C-termini of mitochondrial β -barrel outer membrane channels VDAC2 and 3 seems evolutionary conserved."

10) In this study, all HtrA1 proteins used in in vitro experiments have deletion in the N-terminal (~152) domain. Authors should discuss if the N-terminal domain affects trimerization and protease activity.

AUTHORS: The N-terminus of HTRA1 represents a fragment of insulin-like growth factor binding protein 7 (IGFBP7). Because both constituents, insulin-like growth factor binding domain and the Kazal-like protease inhibitor domain are incomplete in HTRA1, they are non-functional. The crystal structure of this domain revealed the presence of eight disulfide bonds⁶ that are expected to contribute to protein stability and therefore to protein half-life in the extracellular space.

We therefore added the following statement (see Methods section, page 1, lines 28-33):

"Note that since we and others have not detected any effect of the N-terminus of HTRA1 on oligomerization and protease function^{6,7}, in vitro experiments were conducted with HTRA1 proteins lacking the IGFBP7-like fragment (ΔN) except crystallography, NMR DOSY assays, some data presented in Supp. Fig. 12a and 14a which were obtained using $\Delta N\Delta PDZ$ variants, and data presented in Supp. Fig. 4a which were obtained using culture medium from HEK cells transfected to overexpress full-length HTRA1."

REVIEWER 2:

The manuscript by Dichgans and coworkers looks into a structural mutation in serine protease HtrA1 that has been identified in cerebral vasculopathy. The authors have looked into the mutation both from structural and functional perspectives to understand the cause for its pathogenicity using in vitro and in vivo tools. They have also devised strategies to rescue this phenotype using structure-guided design and allosteric modulation.

The manuscript is reasonably well-written and provides an interesting approach to devise therapeutic strategies against diseases where HtrA1 oligomerization might play a prominent role. However, there are few important concerns that need to be addressed.

AUTHORS: We thank the reviewer for the overall positive evaluation of the manuscript and for the constructive suggestions. **Page and line numbers refer to the clean version of the revised text.**

1. The protein is shown as a monomer (one very important aspect of the manuscript) using crosslinking studies. Other independent studies such as size exclusion chromatography and most importantly Analytical ultracentrifugation should be done. This will also give information about the stability of the trimeric ensemble, with a precise K_d value as well as the stoichiometry of monomeric and trimeric species.

AUTHORS: We agree with the reviewer. We performed additional experimental work and investigated the oligomeric states of HTRA1 by independent approaches including size exclusion chromatography (SEC), SEC multi-angle light scattering (SEC-MALS), native mass spectrometry (nMS) including titration assays, as well as sedimentation velocity analytical ultracentrifugation [AUC]. Detailed analysis of the stoichiometry of mono-, di- and trimeric HTRA1 species is now presented in Fig. 1d-e, Fig. 2d-e, Supp. Fig. 1 and Supp. 5 and the corresponding results are described on page 4-5 and 6, lines 100-113 and 138-142.

Note that in AUC runs, the dilution range of protein concentrations necessary for K_d determination (which would require very low concentrations for wt and A252T and very high concentrations for R274Q, R166H and A173T) was not compatible with the dynamic sensitivity range. In native MS, saturation could not be reached with R274Q, R166H and A173T, excluding calculations of trimer stability. As an alternative approach, we also performed isothermal titration calorimetry assays, which did not work due to low signal to noise ratios. Although precise K_d values for subunit assembly into trimers cannot be extracted from the data, they do provide some initial estimates regarding the differences in terms of stability and self-assembly between individual HTRA1 mutants.

2. One of the rescue experiments is based on the protein's allosteric property. However, the manuscript does not provide any enzyme kinetics studies of HtrA1 wt versus the mutant (residual activity is present according to the manuscript). This, along with structural studies might be important toward providing a proper structural roadmap for understanding of its conformational dynamics that leads to loss of activity.

AUTHORS: We agree and have performed enzyme kinetics studies (Fig. 6d, Supp. Fig. 14a-b, and revised description of the results, page 12-13, lines 292-318). Using an 11-mer peptidic FRET substrate, we performed substrate titrations in absence and presence of VDAC2 peptide (Supp. Fig. 14b). Mutant HTRA1 showed 3-4 fold lower affinity for FRET substrate than wt (with the exception of A173T, showing 3-fold higher affinity) and a 5-15 fold decrease in cleavage rates. Furthermore, whereas data obtained for wt indicated a substrate induced cooperativity, consistent with its trimeric structure, interface mutants R274Q, R166H and A173T displayed a hyperbolic dependence on substrate concentration, typical for Michaelis-Menten kinetics. For wt HTRA1, R274Q and R166H, addition of VDAC2 peptide resulted in decrease in K_m and increase in cleavage rates. In case of A173T, addition of VDAC2 peptide resulted in a significant increase of protease activity without apparent tightening of A173T affinity for the substrate. These data demonstrate that VDAC2 peptide acts as an allosteric activator of both

wt and mutant HTRA1 variants. Additional titrations of VDAC2 peptide at fixed substrate concentration (Fig. 6d, Supp. Fig. 14a) indicate that VDAC2 peptide binds in non-cooperative manner, with a higher affinity for wt HTRA1 than for mutant proteins.

We have tried co-crystallization of HTRA1 and VDAC2 peptides. After analyzing many crystals of different space groups obtained at various conditions, we had to accept that none of them contained peptide. We included a statement into the text (see Results, page 13, lines 324-325):

"To initially address the mechanism of activation, co-crystallization of HTRA1 and bound peptide was attempted but failed."

3. Although, the authors have said that HtrA1 R274Q mutant is a monomer, they found some activity in the mutant. No explanation has been provided whatsoever for this observation.

AUTHORS: An explanation has been added to the text (see Results, page 4-5, lines 100-103 and 106-113):

"All three interface mutants were mostly detected as monomers in size exclusion chromatography (SEC), SEC multi-angle light scattering (SEC-MALS) and chemical crosslinking, while wt HTRA1 and A252T were trimers (Fig. 1d, Supp. Fig. 1b-c and Supp. Fig. 3a)."

"Native mass spectrometry (nMS) experiments, including titration assays, as well as sedimentation velocity analytical ultracentrifugation (AUC) further indicated that in R274Q, R166H, and A173T but not in A252T the monomer-trimer equilibrium was shifted towards monomers (Fig. 1e, Supp. Fig. 1d-e). The small amounts of trimers account for the residual levels of enzymatic activity detected in interface mutants. A173T exhibited the lowest relative trimer abundance (Supp. Fig. 1d-e) in accord with its markedly reduced activity compared to R274Q and R166H (Fig. 1c)."

4. Most importantly, the disease is extremely rare and is multi-factorial; HtrA1 mutations might not be the only or prime reason. While, choosing the mutation, the authors did not provide any rationale behind picking this particular mutation. Moreover, how many other HtrA1 mutations are found in the same disease, their positions, their frequencies, ethnic backgrounds of the patients, and their oligomeric properties with relevant references (a table would be helpful) is important. Furthermore, what is the percentage of patients with cerebral vasculopathy who do not harbor any HtrA1 mutation? How different is the severity between these two subsets of patients? In this respect, I would consider the manuscript to be a quite incomplete and lacks significance of its further clinical applications. Moreover, this information should be provided to help understand how general their approach would be to rescue the pathogenic phenotype when mutations can be any.

AUTHORS: Our work is a proof of concept study that explores distinct strategies to correct the assembly and function of rare HTRA1 variants linked to small vessel disease. Our study was not designed to identify a general strategy to therapeutically approach patients with cerebral vasculopathy.

As correctly stated by the reviewer, CARASIL is a rare disorder. In contrast to some other hereditary conditions such as cystic fibrosis, where >45% patients carry the same (CFTR F508del) mutation, the majority of pathogenic HTRA1 variants are private mutations affecting single or few families.

Following the Reviewer's suggestion, we now include detailed information on the spectrum of CARASIL-related HTRA1 mutations (see Supp. Table 1). Information includes the position of mutations, the number of afflicted families, the country of origin/ethnicity of the patients, as well as the enzymatic and oligomeric properties of the corresponding mutant proteins (where reported).

We selected R274Q because (i) R274Q is located at the trimer interface (Fig. 1b), (ii) R274 is directly involved in interprotomer salt bridges (based on modeling, Supp. Fig. 2) and (iii) R274Q is one of the few CARASIL-causing mutations identified in more than one pedigree (see above). Note that we also elaborated on two additional repair approaches that are not restricted to this variant.

To account for the reviewer's comment, we expanded the text (see Results, page 5, lines 118-120):

"We initially devised a targeted protein repair strategy focusing on the archetypical interface mutation R274Q, which is one of the few CARASIL-causing mutations identified in more than one pedigree (Supp. Table 1)."

Regarding the percentage of patients with cerebral vasculopathy who do not harbor any HTRA1 mutation and disease severity in those with HTRA1 mutations vs those without HTRA1 mutations: To our knowledge there has been no sufficiently large study that systematically explored the frequency of rare exonic HTRA1 mutations in sporadic small vessel disease (SVD). We acknowledge that the majority of patients with cerebral SVD, which is highly prevalent in the elderly population, do not harbor known pathogenic germ line mutations in HTRA1 or other genes. Still, there is a broader role of rare HTRA1 variants beyond CARASIL, which is caused by biallelic HTRA1 mutations (see statement below).

To account for the reviewer's comment we expanded on the Discussion (see page 15, lines 381-389):

"The majority of patients with cerebral small vessel disease, which is highly prevalent in the elderly population, do not harbor known pathogenic variants in HTRA1 or other genes. However, there is a broader role of rare HTRA1 variants beyond CARASIL, which is caused by biallelic HTRA1 mutations: First, monoallelic mutations, which are linked to a milder vasculopathy with later onset, account for up to 5 % of familial small vessel disease cases of unknown etiology^{8,9}. The respective variants include interface mutations that destabilize the trimer such as R166C, R166L and A173P^{8,10}. Second, trimer interface mutations including R166H and R274Q have been identified in cancer genome projects^{11,12} thus opening additional perspectives for our correctors."

REVIEWER 3:

The manuscript titled “Rational correction of pathogenic conformational defects in HTRA1” by Beaufort et al, describes a broad study aimed at correcting effects of the R274Q substitution in the human HtrA1 protease both in vitro and in vivo. This substitution is a member of a group of the HtrA1 substitutions inactivating the protease and linked to cerebral autosomal dominant arteriopathy with subcortical infarcts and leukoencephalopathy (CARASIL). The authors found that the loss-of-function mutations cause changes at the interface of the protomers of the HtrA1 trimer which prevent trimerization and thus lead to the enzyme inactivity. They set out to achieve the functional correction of the mutated variant by several approaches. Firstly, they identified a compensatory substitution (D174R) which suppressed inactivity of the R274Q variant in cis by restoring the enzyme ability to trimerize. Interestingly, they were able to show that the suppression was possible when the compensatory substitution was present in trans, i. e. in another HtrA1 protein. The suppression was documented in vitro by showing that the enzyme was active and formed trimers. Moreover, the suppression in trans worked also in vivo, in the mouse model. Since the HtrA1 deficient mice have no obvious phenotype, including no CARASIL symptoms, the Authors monitored HtrA1 activity by comparing proteomes of mouse brain vessels using the genetically-edited mice (by CRISPRCas9 method). This rather laborious approach showed many differences between the wt and mutant mice, especially among the extracellular proteins. The HtrA3 candidate substrates, LTBP4 and PRSS23 proteins, served as indicators to monitor the effects of supplementing the mice carrying the R274Q HtrA1 protein with the compensatory D174R variant. Indeed, the HtrA1 D174R efficiently suppressed the R274Q mutation. This was in my opinion a very elegant experiment showing complementation of disease-causing protein by another protein. Secondly, as protein delivery in humans poses many problems, the Authors searched for other ways of correcting the defective HtrA1. One approach was to use supramolecular ligands to induce trimerization of the R274Q HtrA1. Another one was to find peptides which would allosterically activate the monomeric R274Q variant. In both cases they were able to identify ligands which increased activity of the defective HtrA1 in vitro. In conclusion, this work shows several ways of correcting malfunctioning disease-linked protein by influencing its conformation.

AUTHORS: We thank the reviewer for the in-depth and overall positive evaluation of the manuscript. **Page and line numbers refer to the clean version of the revised text.**

Concerns

1. In the experiment in which trimerization of the R274Q HtrA1 variant was induced by GCPs, relatively high concentrations (2.5 mM) of the ligands were used with 25 μ M HtrA1, which gives the protein/ligand ratio of 1:100. As the general aim of the work was to design ways of correcting malfunctioning proteins in humans, would the Authors comment on this? Could such excess of a therapeutic ligand be achieved? Secondly, the GPC ligands bind oxyanions, so they may bind a lot of proteins, not only HtrA1. The specificity problem should be discussed.

AUTHORS: We agree, the affinity of GCPs for HTRA1 is rather weak and their specificity low. While GCPs were used in cell-based studies¹³, they are not suited for clinical use at this stage. However, their properties could be improved by adding properly positioned additional units as multivalency will likely improve their affinity and selectivity. Also, as a non-covalent binder, the off-target effects should be lower compared to covalent binders. Irrespective of their clinical utility, we feel there is a conceptual value in this class of compounds.

To explicitly address these aspects, we expanded on the Discussion (see page 15, lines 367-369):

“In their current form, the affinity of GCPs is rather weak and their specificity low. Nevertheless, their properties could be improved e.g. by adding properly positioned additional units.”

2. The Authors used peptidic modulators to correct function of the inactive HtrA1 variants. The best of them, the VDAC2 efficiently stimulated activity of the mutant HtrA1 variants against purified tau protein. However, it also stimulated very efficiently activity of wt HtrA3 and, on the other hand, inhibited trypsin and chymotrypsin activity. This again raises a question of the ligand specificity. This should be pointed out and discussed.

AUTHORS: We agree with the reviewer that this is an important aspect. We performed additional assays using reference synthetic substrates instead of Tau and found that VDAC2 does not interfere with the activity of other relevant serine proteases (trypsin, chymotrypsin, elastase and HTRA2) except a moderate inhibition of HTRA3 (see revised Supp. Fig.12b). Our previous observations suggesting an inhibition of trypsin and chymotrypsin by VDAC2 most likely reflect a competition between VDAC2 (50 μ M) and the substrate Tau (3 μ M).

We revised the Results section (page 13, lines 319-324) and expanded on the Discussion (page 15, lines 373-375):

“Given the composition and sequence specificity of the activation domain, a certain degree of selectivity of the VDAC2 peptide would be expected. The corresponding assays indicated that indeed, HTRA2 and other serine proteases sharing the chymotrypsin-fold (trypsin, chymotrypsin and elastase) were not affected, while HTRA3 was slightly inhibited at and above peptide concentrations of 10 μ M (Supp. Fig. 12b).”

“Future efforts should be directed to improve critical parameters of these peptides such as solubility and stability in complex biological systems.”

3. The Authors did a lot of work in silico to characterize interaction of the VDAC2 activating peptide with the HtrA1 R274Q and wt proteins, found places of the ligand binding and concluded that VDAC2 is an allosteric activator binding mainly to the active site and alternative site situated at the trimer tip. However, the Authors should support their in silico results by kinetic experiments showing that VDAC2 is indeed an allosteric activator. This could be easily done, since fluorogenic substrate peptides for HtrA1 are available.

AUTHORS: We agree and have performed additional enzyme kinetics studies. Using an 11-mer peptidic FRET substrate, we (i) titrated VDAC2 at a fixed concentration of substrate (see Fig. 6d, Supp. Fig. 14a and revised description of the results, page 12, lines 293-301) and (ii) titrated substrate in the absence or presence of VDAC2 (see Supp. Fig. 14b and revised description of the results, page 12-13, lines 302-318). As depicted in Supp. Fig. 14b, we found that VDAC2 reduces K_m (HTRA1 wt, R274Q, R166H), demonstrating allosteric activation.

4. In my opinion, the model of HtrA1 activation (shown in the Extended Data Fig.10) should be placed in the main text, possibly as an additional panel of Fig. 6. This would help the general readers who may not be acquainted with the scheme of the HtrA proteins regulation to understand the text without going to the Extended Data part.

AUTHORS: We agree. We inserted the model of HTRA1 activation in Fig. 6 and adapted the legend as shown below:

Fig. 6a: Model of activation of trimeric and monomeric HTRA1. Left panel: in wt HTRA1, the substrate bound to the active site interacts with loop L3, followed by an inter-protomer interaction of loop L3 with

loop LD. This interaction mediates the proper positioning of loops L1 and L2, and of the catalytic site. Right panel: in monomeric mutant HTRA1, loop L3-LD interaction is disrupted. The VDAC2 peptide acts as a surrogate of the missing loop L3 from an adjacent protomer leading to the proper positioning of loop LD and thus of loops L1 and L2. Therefore, in active monomeric HTRA1, the activation domain resembles that of classic monomeric serine proteases such as trypsin and chymotrypsin that have no loop L3¹⁴.

5. Fig. 3d and Supplementary Fig. 4d. What does the 150 kDa band represent? Is this a degradation product of LTB4?

AUTHORS: LTBP4 typically migrates as multiple bands on immunoblot analysis of mouse tissues (see¹ presenting analysis of tissues from *LTBP4*^{wt} and *LTBP4*^{KO} mice and using the anti-LTBP4 Ab AF2885 from R&D Systems, which we also used in our work). The exact identity of individual bands is unknown. They most likely account for post-translational modifications (e.g., glycosylation, proteolytic processing) and/or splice variants^{1,2}.

To account for the reviewer's comments, we expanded on the legend to Fig. 3d:

"The multiple LTBP4 bands most likely account for post-translational modifications and splice variants"

REVIEWER 4:

The manuscript by Ehrmann, Dichgans and coworkers shows that certain loss-of-function mutants in the homotrimeric serine protease HTRA1, known to cause cerebral vasculopathy, affect the stability of the trimer by perturbing interactions across the protomer-protomer interface. The authors show that the enzymatically active trimer can be reconstituted by either introducing compensating mutations that stabilize the trimer, or binding protomer-bridging low-molecular weight ligands, or binding target-derived peptides at an allosteric site. The compensating mutations can be introduced in cis or trans, as shown by in vitro experiments. Using transgenic mice, the authors proceed to show that the same approach generates enzymatically active trimers in vivo when presented in the context of heterozygous animals containing the disease-causing mutant R274Q on one allele and the compensatory (and inactive) mutant D174R-S328A on the other. The manuscript represents a considerable amount of work involving, among other techniques, sophisticated genetic approaches and quantitative MS analyses comparing the proteome in brain vessels of wild-type and mutant mice. Still, I am not at all convinced that this contribution will have any major impact for several reasons.

AUTHORS: We are grateful for the critical assessment of our work. **Page and line numbers refer to the clean version of the revised text.**

1. The work showing, in vitro and in vivo, that interactions between protomers are important for enzymatic activity is solid, but is an expected result in line with previous reports in the literature.

AUTHORS: We appreciate that activation by oligomerization of protomers is a classic and widely studied mechanism of enzyme activation. However, please note that this paper is about the restoration of mutationally perturbed interaction of protomers (and activation of protomers), a field that is still in its infancy.

2. The authors highlight the suggested protein-based functional correction as a potential therapeutic strategy. It is entirely unclear to me why the proposed mutant D174R-S238A would offer a suitable genetic therapy of disease; how does introducing this mutant lead to improvements over the alternative to perform gene-editing back to the wild-type? What am I missing here?

Given the fact that despite recent efforts, gene-editing is not an established procedure of treatment, we feel that additional concepts such as protein repair are of general interest. One of these strategies is to generate mixed and thus stable trimers by adding a protein that is able to correct assembly defects. The proteolytically inactive corrector variant was employed to ensure that the gain of activity of the mixed trimers is not attributable to the enzymatic activity of the corrector but exclusively results from the conformational repair of the pathogenic mutant R274Q.

Note that editing wt HTRA1 to D174R-S328A is not meant to be a genetic therapy of disease. We used this approach to circumvent the difficulty of expressing the corrector protein in mouse brain vasculature at relevant sites and at sufficient levels for a long-term period.

3. It seems to me that the low-molecular weight ligands represent a more interesting approach, but these were not characterized in vivo. Would it be possible to test whether these ligands are effective treatments in the mouse model?

AUTHORS: Testing the consequences of the low-molecular weight ligands on protein *repair in vivo* would require prior pharmaceutical studies including the capacity of compounds to cross the blood-brain barrier. We therefore couldn't address these aspects within the context of a revision.

4. I also wonder why the in-vivo work did not use the active D174R mutant (rather than the inactive D174R-S238A)? The combinatorics of trimer formation predict that the active D174R mutant would rescue the R274Q mutant to yield 3/4 of the (homozygous) wild-type activity, whereas the rescue achieved by the inactive D174R-S238A would yield only 1/4 (assuming that each D174R or R274Q protomer is active in the trimer). Along the same lines, introducing the wild-type in one allele would yield 1/2 of the activity in non-diseased individuals.

AUTHORS: We agree that both the active D174R and simply adding back wt HTRA1 would compensate the lack of HTRA1 activity. However, our aim was to demonstrate the feasibility of repairing the conformation and enzymatic activity of R274Q by protein-based complementation. Hence, there would have been no point in adding a repair factor that itself is enzymatically active. Also, from a therapeutic aspect, dosing of active HTRA1 protease would be challenging.

5. In connection to point 4, it would be of interest to know the variation between mutants (and between mixtures of mutants) in trimer stability. I suggest that you measure the stability by biophysical methods.

AUTHORS: This is a very interesting point, which we have attempted to address.

We investigated the oligomeric states of HTRA1 (wt, mutant and mixtures of R274Q and D174R-S328A) by independent approaches including size exclusion chromatography (SEC), SEC multi-angle light scattering (SEC-MALS), sedimentation velocity analytical ultracentrifugation (AUC) and native mass spectrometry (nMS), including titration assays. Detailed analysis of the stoichiometry of mono-, di- and trimeric HTRA1 species is now presented in Fig.1d-e, Fig. 2d-e, Supp. Fig. 1 and Supp. Fig. 5, and the corresponding results are described on page 4-5 and 6, lines 100-113 and 138-142.

In native MS, saturation could not be reached with R274Q, R166H and A173T, excluding calculations of trimer stability. In AUC runs, the dilution range of protein concentrations necessary for Kd determination was not compatible with the dynamic sensitivity range. As an alternative approach, we also performed isothermal titration calorimetry assays, which did not work due to low signal to noise ratios. Although precise Kd values for subunit assembly into trimers cannot be extracted from the data, they do provide some initial estimates regarding the differences in terms of stability and assembly.

6. Further, it would be of interest to measure the activity (in terms of LTBP4-Nt and PRSS23-Ct levels) in wt/R274Q mice, thereby addressing the issue of trimer combinatorics.

These data are depicted in Supp. Fig. 9c and below.

HTRA1, PRSS23-CT and LTBP4-NT levels in mouse brain vessels determined by MS. The mean abundance in *Htra1*^{wt} vessels was set to 1; results as box-and-whisker plots (median, minimum, maximum); data points represent individual mice; significance was tested by two-sided unpaired *t*-test.

We would like to point out that quantification of protease activity is not always straightforward in an *in vivo* model as it depends on a variety of factors including the levels of HTRA1 variants

available for forming mixed trimers, the affinity of individual cleavage sites and the accessibility and concentration of substrate, which will all affect substrate-enzyme interactions. We are therefore hesitant to expect e.g. a linear correlation between trimer assembly/combinatorics and substrate levels. However, there is a clear qualitative correlation between genotypes and the processing of substrates.

Other points (roughly in order of appearance):

The title claims "Rational correction..." but the chosen corrective mutant does not seem to behave as one might have naively expected (or rationally designed): the introduced arginine (R174) reaches over to the neighboring protomer where it forms stabilizing van der Waal's interactions, rather than re-establishing hydrogen bonding interactions to Q274. To this extent, the stabilizing effect of the chosen mutant seems rather serendipitous.

AUTHORS: From a stereochemistry perspective, it is unlikely that R174 adopts a rotamer conformation in which it could form a hydrogen bond to Q274, as the reviewer surmised, and was not expected. Thus, a discussion whether rationality or serendipity lead to the identification of the compensatory mutation is otiose and we would like to keep the title as it is. We would also like to point out that the original idea of generating proteolytically active mixed trimers by combining protomers that are by themselves inactive was based on reason.

line 69: "...several loops that are shared between protomers." How can a loop be shared between protomers?

AUTHORS: A loop is shared between protomers if it originates in one protomer and reaches over to another protomer to e.g. complete an activation domain.

To facilitate the reader's understanding, a scheme of HTRA1 activation (initially presented in Extended Data) has now been inserted in Fig. 6.

Fig. 6a: Model of activation of trimeric and monomeric HTRA1. Left panel: in wt HTRA1, the substrate bound to the active site interacts with loop L3, followed by an inter-protomer interaction of loop L3 with loop LD. This interaction mediates the proper positioning of loops L1 and L2, and of the catalytic site. Right panel: in monomeric mutant HTRA1, loop L3-LD interaction is disrupted. The VDAC2 peptide acts as a surrogate of the missing loop L3 from an adjacent protomer leading to the proper positioning of loop LD and thus of loops L1 and L2. Therefore, in active monomeric HTRA1, the activation domain resembles that of classic monomeric serine proteases such as trypsin and chymotrypsin that have no loop L3¹⁴.

lines 153 and 215: the changes in levels of HTRA1 seem to suggest that the R274Q/D174R-S328A construct has 11/7.5 higher levels than wild-type(?) If correct: How do you explain this result?

AUTHORS: Note that the 11-fold elevation of protein levels are compared to *Htra1*^{R274Q} and not the wt (see below). Also, the ratios provided in Line 153 and Line 215 refer to two different analyses of the entire MS dataset using distinct setups. Therefore, they cannot be directly compared. Specifically:

The >7.5-fold depletion of HTRA1 in *Htra1*^{R274Q} compared to *Htra1*^{wt} vessels (line 153; line 158 in the revised text) is derived from MS quantification of the entire proteome based on the reference mouse protein sequences.

In contrast, the > 11-fold enrichment of Pan-HTRA1 in *Htra1*^{R274Q/D174R-S328A} compared to *Htra1*^{R274Q} (line 215; line 216 in the revised text) is derived from quantification of the entire proteome using a search database that was modified as follows: (1) the canonical HTRA1 sequence was replaced by the Pan-HTRA1 sequence (excluding aa 274); (2) LTBP4 was replaced with two sequences, which separately depict its N- and C-terminal parts (aa 1-143 and aa 144-1,666, respectively); (3) PRSS23 sequence was shortened (aa 50-382) to only include its C-terminal part (see Methods section, page 12, lines 392-402 and Results, page 8, lines 191-196).

As depicted in Fig. 4d (panel “Pan-HTRA1”) and below, there is no significant accumulation of HTRA1 in *Htra1*^{R274Q/D174R-S328A} compared to *Htra1*^{wt} vessels (fold change 1.26, p=0.15; with MS quantification based on the custom protein sequences). This holds true when quantifying proteins based on the reference protein sequences (fold change 1.18, p=0.22).

HTRA1 protein levels in mouse brain vessels determined by MS. The mean abundance in *Htra1*^{wt} vessels was set to 1; results as box-and-whisker plots (median, minimum, maximum); data points represent individual mice; significance was tested by two-sided unpaired *t*-test. Ref-HTRA1: analysis of the entire proteome using the reference sequences; Pan-HTRA1: analysis of the entire proteome using a modified FASTA file (see text for explanations). Note the accumulation of D174R-S328A. In contrast to wt HTRA1, this inactive (but trimeric) variant is not prone to auto-degradation.

Figs. 5d and 6c: I do not find that these plots provide convincing evidence of significant differences between the different cases, in part because it is hard to discern the 25 μ M data among the different gray shades. Also please present for each case the extracted translational diffusion coefficient and its estimated error.

AUTHORS: To improve the **clarity of the graphs** we have adapted the plots and introduced insets depicting only 3 conditions: R274Q 25 μ M (monomeric), R274Q 100 μ M (trimeric) and R274Q 25 μ M + MK2 or VDAC2 (see below). In the revised manuscript, NMR-DOSY assays are presented in Fig. 5c and Fig. 6c.

Oligomeric states of HTRA1 analyzed via NMR DOSY experiments. The slope in the Stejskal-Tanner plots represents the negative diffusion coefficient. Small molecules exhibit a larger diffusion coefficient and thus a steeper slope, while large molecules with a smaller diffusion coefficient show a shallower slope. Upper panels (Fig. 5c): oligomeric states of wt and mutant HTRA1 in the absence or presence of MK2. Inlet: 25 μM R274Q (monomeric), 100 μM R274Q (trimeric) and 25 μM R274Q + 2.5 mM MK2. Lower panels (Fig. 6c): oligomeric states of wt and mutant HTRA1 in the absence or presence of VDAC2 peptide. Inlet: 25 μM R274Q (monomeric), 100 μM R274Q (trimeric) and 25 μM R274Q + 50 μM VDAC2.

In accord with the reviewer's request, the **extracted translational diffusion coefficient and estimated errors** are now presented in Source Data:

	wt	R274Q
c(HTRA1)	D / 10^{-10} m/s ²	D / 10^{-10} m/s ²
100 μM	0.62 \pm 0.01	0.61 \pm 0.01
50 μM	0.60 \pm 0.01	0.70 \pm 0.01
25 μM	0.60 \pm 0.01	0.77 \pm 0.01
10 μM	0.66 \pm 0.01	0.77 \pm 0.01
25 μM + 2.5 mM MK2	0.60 \pm 0.01	0.66 \pm 0.01
25 μM + 50 μM VDAC2	0.65 \pm 0.03	0.79 \pm 0.04

HTRA1 translational diffusion coefficients and estimated errors obtained from DOSY Stejskal-Tanner-Plots. The diffusion coefficient is the absolute value of the slope fitted from the linear Stejskal-Tanner-Plot.

line 324-5: "...overall remodeling of the protomer-protomer interface, rather than pairing with a defined partner residue..." This wording does not agree with Fig. 2, which shows a modest change in the orientation of just a few side chains.

AUTHORS: We rephrased the sentence (see Discussion, page 14-15, lines 363-366):

"As revealed by structural analysis, functional correction involves an overall increased surface area between protomers, which we consider as a remodeling rather than pairing between defined novel partner residues, thus underscoring the plasticity of protomer interfaces."

line 338-40: Please explain why the inactive mutant is better than the active one (or the wild-type). Are you suggesting that the homozygous wild-type is causing detrimental off-target effects?

AUTHORS: The advantage of an enzymatically inactive corrector is that restoration of activity is conditional on the presence of the pathogenic mutant thereby limiting off-target effects, which could emerge with simply adding wild-type HTRA1 protein (e.g. via systemic administration).

We don't suggest that the homozygous wild-type allele is causing detrimental off-target effects.

We adapted the Discussion (see page 15-16, lines 391-393):

“The advantage of an enzymatically inactive corrector is that restoration of activity is conditional on the presence of the pathogenic mutant thereby limiting off-target effects.”

REVIEWER 5:

In this manuscript, authors describe that certain disease causing mutations in the homotrimeric serine protease HTRA1 leads to loss of enzymatic activity by impairing multimeric assembly of HTRA1. Using a genetic approach, the authors show that a corrector variant is active in trans and rescue the molecular phenotype in vivo. Concurrently, authors developed two additional strategies based on small molecules to correct the activity of the mutants. On one side, the authors describe small-molecule stabilizers based on guanidinocarbonyl pyrroles that restore multimeric assembly of HTRA1 variants. The second approach describes short peptides that allosterically activate HTRA1 wt and variants in the monomeric form. Overall, the paper reports interesting observations and approaches, but not enough explanations in terms of biophysics and molecular interactions.

AUTHORS: We thank the reviewer for the in-depth assessment of our manuscript and for the constructive comments. **Page and line numbers refer to the clean version of the revised text.**

- It is not clear how the compensatory mutation (D174R) actually works. The authors mention multiple van-der-Waals contacts that could result in an overall increased trimer interface but these contacts are not specifically identified nor shown in Fig. 2a.

AUTHORS: We have removed the statement on van der Waals interactions because we feel that the large increase in the surface area is stabilized by multiple interactions that cannot be observed at 3.2 Å resolution.

The relevant section now reads (see Results, page 6, lines 146-149):

“Specifically, R174 reoriented towards the protomer at the opposite interface (Fig. 2a, panel 2”) resulting in an overall increased trimer interface as compared to the single R274Q mutant (buried surface area per protomer: wt [PDB ID 3TJO]: 947 Å²; R274Q [PDB ID 6Z0E]: 908 Å²; D174R-R274Q [PDB ID 6Z0X; 6Z0Y]: 925-957 Å²).”

- Fig.5a is vague and we do not understand how GCP actually brings the two monomers together. It is unclear whether the image shown in Fig 5a is based on molecular modelling. If yes what are the key residue on the surface of HTRA1 that would interact with the GCP ligands. In the absence of additional evidence that the molecule is actually binding as suggested there is no reason why to suggest a particular mode of binding.

AUTHORS: We do agree with the reviewer that we cannot suggest a particular mode of binding and have therefore removed Fig. 5a.

We revised the text accordingly (see Results, page 10 and 11, lines 236-238, 249-250, 262-264):

“To devise an alternative strategy that would target multiple HTRA1 interface mutants, we reasoned that supramolecular ligands exposing positive charges on either end might bind anionic hotspots on adjacent protomers and stabilize trimers.”

“...the addition of MK2 stabilized trimers.”

“We therefore hypothesize that binding of GCPs to as yet unidentified one or more binding site(s) triggers structural rearrangements ultimately causing the stabilization of trimers.”

-The representations proposed for the interaction between VDAC2 and HTRA1 R274Q in Fig 6d/e could be improved by using surface representation for the protein, by using different colours for the side chain of the peptide and the protein as well as orthoscopic view. Biophysical experiments (including affinity measurements) are needed to confirm the interaction between HTRA1 R274Q and VDAC2 as well as confirm the limited or absence of interaction of HTRA1-R274Q with VDAC2-2 or VDAC2-3. Authors do not show experimental evidence for the location of the second binding site for allosteric activation. A mutant of wt HTRA1 that would incorporate a mutation in the putative

alternative site should abrogate activation by VDAC2 but not by VDAC2-2 and would help validating this hypothesis. This would provide further evidence for a second binding site and this hypothesis could be reinforced by evaluating the effect of the mutation in HTRA1-R274Q

AUTHORS: As suggested, we adapted the **representation of the proposed interaction** of VDAC2 and HTRA1 R274Q (see revised Fig. 6e and below).

Fig. 6e: Model of the binding mode of VDAC2 peptide at the N-terminal tip of HTRA1-R274Q. Backbones of HTRA1 (orange) and VDAC2 peptide (magenta), sidechains of the catalytic triad's residues are shown with sticks (licorice style). Green: hydrophilic residues; grey: hydrophobic residues. In addition, surface representation is also used for HTRA1 (transparent surface). Inset: close-up view of HTRA1's residues within 4 Å of VDAC2. Sidechains of HTRA1's residues are depicted with sticks (licorice style and colored in yellow except oxygen and nitrogen atoms, which are colored as red and blue respectively). Sidechains of VDAC2 peptide's residues are shown with balls and sticks (CPK style and colored in light blue except oxygen and nitrogen atoms, which are colored as red and blue respectively).

ITC measurements for determining the **affinity of VDACs** failed because of solubility issues (we tested several buffers).

Enzyme kinetic studies, revealing an **allosteric mechanism of activation** were performed and are now included into the manuscript (see Fig. 6d, Supp. Fig. 14a-b, as well as the revised description of the results, page 12-13, 292-318).

We agree that a **mutation in the putative alternative site** should abrogate activation by VDAC2 but not by VDAC2-2 and would help validating this hypothesis. However, initial studies suggested that a single point mutation might not be enough or the one required is not straightforward to identify without causing folding problems.

We therefore decided to focus on native mass spectrometric characterization of peptide binding and occupancy as well as on the oligomeric states of HTRA1 (see Fig. 6b and Supp. Fig. 13 and revised description of the Results, page 11-12, lines 285-291).

-Other points to be addressed:

1- How frequent are the mutations selected in this work among pathogenic HTRA1 mutations? This information could be added in the text.

AUTHORS: The majority of pathogenic HTRA1 variants including those selected in the current work are private mutations affecting single or few families. To address the comments of this reviewer and those of reviewer 2 we added detailed information on the spectrum of pathogenic HTRA1 mutations focusing on variants implicated in CARASIL (see Supp. Table 1).

2- Isothermal titration calorimetry (ITC) could prove useful to further compare the ability of wt HTRA1 and mutants to self-assemble. This could be achieved by titrating a concentrated solution of HTRA1 into the same buffer solution, the calorimeter would then measure the heat variation upon shifting the monomer-oligomer equilibrium towards the monomer.

AUTHORS: As suggested by the reviewer, we performed ITC assays to evaluate HTRA1 trimer stability and determine K_d. These attempts did not work due to low signal to noise ratios.

We performed additional experimental work and investigated the oligomeric states of HTRA1 by independent approaches (size exclusion chromatography [SEC], SEC multi-angle light scattering [SEC-MALS], native mass spectrometry [nMS], including titration assays, as well as sedimentation velocity analytical ultracentrifugation [AUC]). Detailed analysis of the stoichiometry of mono-, di- and trimeric HTRA1 species is now presented in Fig. 1d-e, Fig. 2d-e, Supp. Fig. 1 and Supp. 5 and the corresponding results are described on page 4-5 and page 6, lines 100-113 and 138-142. Although precise K_d values for subunit assembly into trimers cannot be extracted from the data, they do provide some initial estimates regarding the differences in terms of stability and self-assembly between individual HTRA1 mutants.

3- Modifications of the tris(2-aminoéthyl)amine linker (e.g. without the free amino group, or with longer or more rigid chain) in the MK series would help to gain information about the molecular determinants for bridging HTRA1 monomers.

AUTHORS: Following the suggestions of the reviewer (and of reviewer 6), we tested additional compounds.

Overall, we now present data on MK1 and MK2 (with a slightly longer and more rigid chain compared to MK1), AZ21 and AZ25 (with only one GCP unit and an unpolar sterically demanding side arm, Supp. Fig. 11c-d), and HXY23 (based on a rigidified peptide backbone, Supp. Fig. 11c-d) as well as new data on TNMK09 (with a more rigid chain and carrying one GCP motif) and TNMK27 (a monovalent ligand exhibiting a non-charged group at the non-GCP end) (see revised Fig. 5a and the Figure below). We also synthesized additional compounds exhibiting shorter chains devoid of free amino group and carrying one or two GCP group (TNMK15, 28 and 29, see structures below) but these were not soluble in aqueous solution at 2.5 mM.

While the activation of all interface mutants by MK2 was comparable to that observed with MK1, AZ21, AZ25 and HXY23 appeared inefficient with respect to HTRA1 activation, whereas TNMK09 and TNMK27 caused lower levels of activation of trimer interface mutants (Fig. 5b and Supp. fig. 11d).

We revised the text accordingly (see Results, page 10-11, lines 240-244, 253-255, 255-262):

“Compounds MK1 and 2 carrying two GCP groups connected to a central tris(2-aminoéthyl)amine core via peptidic linkers of different sizes yielded a 2 to 2.8 fold increase in proteolytic activity of R274Q, R166H and A173T in biochemical assays (Fig. 5b). Wt HTRA1 was mildly affected, if at all (up to 1.2 fold change).”

“Of note, related ligands with only one GCP unit and an unpolar sterically demanding side arm (AZ21, AZ25) or based on a rigidified peptide backbone (HXY23) appeared inefficient with respect to HTRA1 activation (Supp. Fig. 11c-d).”

“To evaluate the contributions of polyvalency, we tested a compound in which one GCP unit was removed, while keeping the number of charges constant (TNMK09) and its derivative harboring a non-charged group at the non-GCP end (TNMK27) (Fig. 5a). These compounds caused lower levels of activation of trimer interface mutants (Fig. 5b). Each specific point mutation is likely to cause structural changes beyond trimer formation, which could explain the non-uniform effects of TNMK09 and TNMK27 on individual interface mutants.”

Structure of TNMK15, 09, 27, 28 and 29. The structure of MK1 and MK2 is depicted for comparison. *: TNMK15, 28 and 29 were not soluble in aqueous solution.

Note that we cannot exclude that binding of GCPs might stabilize trimers through structural rearrangements upon binding to an as yet unidentified site. Hence, we added the comment below, to account for this possibility (see Results, page 11, lines 262-264):

"We therefore hypothesize that binding of GCPs to as yet unidentified one or more binding site(s) triggers structural rearrangements ultimately causing the stabilization of trimers."

4- The authors mention that GCP actually compare favourably with other compounds tested (p10, line 252 of the manuscript) but it looks as if compound MK99 with two Arg is as effective as MK1 when looking at Supplementary Fig. 5b. Authors should explain this discrepancy or remove the statement. The concentration of compounds tested should be reported in the legend of supplementary Fig. 5 for better comparison.

AUTHORS: As stated in our reply to point 7, we realized the purity of several compounds was unsatisfactory. We therefore decided to remove the corresponding compounds (initially presented in Supp. Fig. 5).

The concentration of the compounds (i.e., 2.5 mM) is now provided in the legend to revised Fig. 5 and in Supp. Fig. 11.

5- Molecular approaches aiming at stabilizing protein-protein interactions is expanding and useful literature is worth being cited on this specific topic (see for example Andrei et al. Exp. Opin Drug Discov 2017, 12, 925-940)

AUTHORS: We agree and expanded on the Discussion (see page 14, lines 353-355):

"Molecular correction of pathogenic protein conformations including the stabilization of protein-protein interactions recently emerged as a promising therapeutic strategy in genetic diseases^{15,16,17,18,19,20}."

6- The legend for the two lines with + and - is missing in Fig. 6b

AUTHORS: We labeled the +/- in the Figure and expanded on the figure legend. The revised panel is presented in Supp. Fig. 3d as shown below:

Supp. Fig. 3d: Oligomeric states of wt and mutant HTRA1s in the absence or presence of 50 μ M VDAC2 peptide analyzed by chemical GA-based crosslinking.

7- Copies of HPLC traces and MS spectra should be provided for compounds MK1-MK3 as well as control compounds shown in Supplementary Fig. 5.

AUTHORS: We are grateful for this comment. We realized the purity of several compounds was unsatisfactory. This happened because of circumstances around the responsible PI becoming seriously ill and dying in Aug 2019 which has caused a lack of supervision of the graduate student.

As a consequence, we resynthesized the key compounds (MK1 and MK2) and those requested by the reviewers (TNMK09 and TNMK27) and now provide HPLC traces and MS spectra in Source Data. Importantly, we confirmed that MK1 and 2 stabilize R274Q trimers and activate R274Q, R166H and A173T.

8- I could not find details about the purity of peptides used to study allosteric activation of HTRA1. This information should be provided.

AUTHORS: The peptides were purchased from INTAVIS Peptide Services at >95% purity.

A relevant statement was included into the Methods section, page 1, line 25:

“Peptides were purchased from INTAVIS Peptide Services at >95% purity.”

9- SI (p26, 27 and 29) please correct Fomc in Fmoc.

AUTHORS: We corrected the text accordingly.

REVIEWER 6:

Beaufort and colleagues report the biochemical defect, i.e. lack of trimerization, of naturally occurring mutations in the serine protease HtrA1 causing cerebral vasculopathy. They develop several strategies to rescue this defect. This includes the biological restoration by introducing a single mutation in HtrA1-R274Q, which was convincingly validated in mouse studies. As this strategy remains therapeutically challenging, the authors describe two alternative strategies to correct the pathogenic R274Q mutation, i.e. the identification of trimer-stabilizing supramolecular ligands and of allosteric activators. While these latter approaches restore enzymatic activity of HtrA1 mutants, they still lack sufficient specificity and in-vivo validation. Nevertheless, the presented work represents a stimulating and creative work to correct the pathogenic effects of Htra1 trimer interface mutations. However, there are some important shortcomings that need to be addressed. Among these are the misleading claim that the authors are the first to unravel the biochemical defects of the interface mutants, which in fact have been described previously by two publications, Nozaki H. et al (Ref 19) and Uemura M et al. (Ref 20). In addition, the methods used to show this defect and the rescue by ligands are not straightforward and need to be supported by additional experiments (SEC). Additional comments that need attention are detailed below.

AUTHORS: We thank the reviewer for the detailed and overall positive assessment of our manuscript and for the helpful suggestions. **Page and line numbers refer to the clean version of the revised text.**

Major Comments:

1. The authors brush over important published studies which have identified the trimerization defect for some of these mutants, including R274Q. Nozaki 2016 did elegant SEC experiments to show that R274Q as well as A252T exist as monomers and trimers, respectively (and w/o any A252T hexamers, which were shown in fig.1d by x-linking), while Uemura 2016 investigated R166C, R166L and A173T. The authors convey the perception that they have identified the mechanism for the first time, which is misleading and incorrect. This entire section needs revision and published data need to be explicitly quoted and their confirmatory data shown here need to go to supplemental section. Accordingly, the claim in the discussion section "we unraveled how a set of point mutations causing ...disrupt protease function" needs modification; the same goes for the abstract.

AUTHORS: We agree with the reviewer that trimerization defects have been reported in two previous SEC-based studies by the Onodera lab. To account for these studies, we revised the text as detailed below.

However, we would like to point out that our revised manuscript (which presents new SEC, SEC multi-angle light scattering [SEC-MALS], native mass spectrometry [nMS] data including titration assays, as well as sedimentation velocity analytical ultracentrifugation [AUC]; see revised Fig. 1 and Supp. Fig. 1) provides an unprecedented in-depth analysis of HTRA1 assembly defects including structural analyses (i.e., modelling and/or crystallography) as well as a detailed biochemical assessment of the oligomeric ensemble including analysis of the monomer/dimer/trimer stoichiometry using complementary approaches.

Following the reviewer's comments, we made the following changes.

Abstract, lines 44-45: *"Here, we established independent approaches to achieve the functional correction of trimer assembly defects."*

Introduction, page 3, lines 75-77: *"A subset of mutations implicated in familial vasculopathy impairs trimerization and thus proteolytic activity of HTRA1^{9,21}. Here, we employed a variety of molecular approaches to achieve the functional correction of these pathogenic variants."*

Results, page 4, lines 99-103: *"In accord with previous results from the Onodera lab^{9,21} all three interface mutants were mostly detected as monomers in size exclusion chromatography"*

(SEC), SEC multi-angle light scattering (SEC-MALS) and chemical crosslinking, while wt HTRA1 and A252T were trimers (Fig. 1d, Supp. Fig. 1b-c and Supp. Fig. 3a).”

Discussion (page 14, lines 349-352): “Combining *in vitro*, *in silico* and *in vivo* analyses, we established mechanistically distinct protein repair approaches to reverse the deleterious effects of pathogenic point mutations interfering with the assembly and catalytic function of HTRA1.”

In accord with the reviewer’s comment we transferred chemical crosslinking results to the Supplementary section (see Supp. Fig. 3). Our new SEC and AUC data are also depicted as Supplementary Data (see Supp. Fig. 1), whereas SEC-MALS and nMS data (including quantifications) are presented in Fig. 1.

The detection of low abundance HTRA1 wt and A252T hexamers in glutaraldehyde-based crosslinking assays possibly reflects a crosslinking artifact.

2.page 4. It was surprising to see that the authors decided to carry out chemical x-linking and NMR to determine the oligomeric states. SEC or SEC-MALS seemed straightforward and accurate and has been used by other authors. Therefore, I would strongly encourage the authors to further validate the oligomeric state of some key HtrA1 mutants using SEC or SEC-MALS. This should also be applied with the identified MK supramolecular ligands and the allosteric activators.

AUTHORS: As requested, SEC and/or SEC-MALS experiments were performed for HTRA1 mutants and mixture of mutants as well as for R274Q with and without MK1. In addition, native mass spectrometry data are provided for HTRA1 mutants and mixture of mutants, as well as for HTRA1 with and without the allosteric activator VDAC2. See revised Fig.1d-e, Fig.2d-e, Fig. 6b and Supp. 1, 5, 11 and 13 as well as revised description of the Results, page 4-5, 6, 10 and 11-12, lines 100-113, 138-142, 251-253 and 285-292.

3.page 11: why did the authors not stay with the casein cleavage assay, but instead switched to Tau? Did the casein assay give similar results? While the effect of these peptides is impressive (albeit surprising since they don't require the PDZ domain), the non-selective character is a significant obstacle in terms of therapeutic potential: Htra3 is activated and trypsin is inhibited. This may just be the tip of the iceberg regarding unwanted effects on the human proteome and needs to be better discussed in terms of therapeutic applications and limitations. The inhibitory activity on trypsin should raise major concerns to the authors in regards to proposing a clear mechanism by this compound: the Htra3 effect could be understood in terms of its similarity to htra1, but this doesn't hold for trypsin where the opposite effect (inhibition) is observed. In addition, the finding that the peptides act on the monomer is surprising and needs to be firmed up by a SEC (-MALS) experiment. The question on how the compound is able to activate both, the wt trimeric form AND the "inactive" mutant monomer, should be better discussed.

AUTHORS: During the extensive revisions, several adjustments were made.

In the submitted form of the manuscript, **casein was replaced by Tau** because when testing peptides for HTRA1 activation, casein was not suitable because it or its cleavage products are strong activators of HTRA1, so activation by peptides can’t be measured. This is not the case with Tau.

For the newly included kinetic studies (Fig. 6d, Supp. Fig. 14), Tau was replaced by a peptidic FRET substrate for improved quantification. In addition, much lower HTRA1 (and thus VDAC2 peptide) concentrations are required compared to the previously used Tau protein.

This is also directly relates to the **selectivity data**. Using reference synthetic substrates instead of Tau, we found that VDAC2 does not interfere with the activity of other relevant serine proteases (trypsin, chymotrypsin, elastase and HTRA2) except a moderate inhibition of HTRA3 (see revised Supp. Fig.12b). Our previous observations suggesting an inhibition of trypsin and

chymotrypsin by VDAC2 most likely reflect a competition between VDAC2 (50 μM) and the substrate Tau (3 μM).

We revised the Results section (page 13, lines 320-324) accordingly:

"The corresponding assays indicated that indeed, HTRA2 and other serine proteases sharing the chymotrypsin-fold (trypsin, chymotrypsin and elastase) were not affected by VDAC2, while HTRA3 was slightly inhibited at and above concentrations of 10 μM (Supp. Fig. 12b)."

The reviewer's request that "the finding that the peptides act on the monomer is surprising and **needs to be firmed up by a SEC (-MALS) experiment**" was addressed by native mass spectrometry that revealed that the peptides bind to monomeric HTRA1 and an occupancy of 1 peptide per monomer/protomer (Fig. 6b and Supp. Fig. 13). In addition, active monomers were further confirmed by NMR DOSY and chemical crosslinking (Fig. 6c, Supp. Fig. 3d). See revised text, page 12, lines 288-291.

In response to the reviewer's request "the question on **how the compound is able to activate both, the wt trimeric form AND the "inactive" mutant monomer**, should be better discussed", we expanded on the text (see Results, page 13-14, lines 339-343):

"Collectively, our observations suggest that the VDAC2 peptide might act as a surrogate of the missing loop L3 of the activation domain that in wt HTRA1 is provided by an adjacent protomer (Fig. 6a). This model also explains why both monomeric HTRA1 and trimeric wt HTRA1 are activated by the peptides."

4. Affinities of individual peptides: The affinities of the most important ligands for HtrA1 should be quantified. For the MK peptides 2.5mM final peptide concentrations were used in enzyme assays, which is extremely high and indicates a very low affinity. However, the VDAC peptides should have a measurable affinity towards the HtrA1 monomer R166H, A173T or R274Q. Similarly, the contribution of different binding sites could be probed in regard to the mutants VDAC2-2 and VDAC2-3 further strengthening the different binding sites (active site vs. alternate site).

AUTHORS: Isothermal titration calorimetry measurements for determining the affinity of VDACs failed because of solubility issues (we tested several buffers).

We do hope, however, that the additional kinetic (Fig. 6d, Supp. Fig. 14) and native mass spectrometry (Fig. 6b, Supp. Fig. 13) data will at least in part make up for the reviewer's request, which is of course absolutely adequate.

5. GCP supramolecular ligands: The authors demonstrate that bi-functional GCP supramolecular ligands can restore the oligomeric state of HtrA1 R274Q mutant. It would be important to show that a monovalent GCP molecule with a single GCP headgroup would fail to restore the trimerization. The authors only show branched monovalent GCP molecules (AZ21 and AZ25) but not the direct derivative of MK1-3 with only a single GCP group. It might be possible that binding of GCP alone induces trimerization through structural rearrangement of the underlying unidentified binding site.

AUTHORS: We agree with the reviewer that binding of GCPs alone might stabilize trimers through structural rearrangements upon binding to an as yet unidentified site. We therefore included into the text (see Results, page 11, lines 262-264):

"We therefore hypothesize that binding of GCPs to as yet unidentified one or more binding site(s) triggers structural rearrangements ultimately causing the stabilization of trimers."

As suggested by the reviewer, we generated and tested TNMK09 (carrying a more rigid chain and one GCP motif) and TNMK27 (a monovalent ligand exhibiting a non-charged group at the non-GCP end) (see revised Fig. 5 and the Figure below). We also synthesized additional compounds exhibiting shorter chains devoid of free amino group and carrying one or two GCP group (TNMK15, 28 and 29) but these were not soluble in aqueous solution at 2.5 mM.

Structure of TNMK09, 27, 28 and 29 carrying a single GCP headgroup. *: TNMK28 and 29 are not soluble in aqueous solution.

TNMK09 and TNMK27 caused lower levels of activation of trimer interface mutants (Fig. 5b). We revised the text accordingly (see Results, page 10-11, 255-262):

“To evaluate the contribution of polyvalency, we tested a compound in which one GCP unit was removed, while keeping the number of charges constant (TNMK09) and its derivative harboring a non-charged group at the non-GCP end (TNMK27) (Fig. 5a). These compounds caused lower levels of activation of trimer interface mutants (Fig. 5b). Each specific point mutation is likely to cause structural changes beyond trimer formation, which could explain the non-uniform effects of TNMK09 and TNMK27 on individual interface mutants.”

Minor Comments:

1. Protease cleavage assay: It would be helpful to show the levels of HtrA1 or the full uncropped gels in the supplement, to validate equal loading and activity across the HtrA1 wildtype and mutants. Are the concentrations of htra1 and mutants based on the monomer concentrations? This is important since some of the mutants, especially the R274Q are in the monomeric state and a comparison of their activities needs to be based on the monomer concentration.

AUTHORS: We now provide enlarged images from the gels in Supp. Fig. 1a and 5a and in Source Data. We also revised the Methods section (see page 4, line 104):

“The concentrations of wt and mutant HTRA1 refer to the monomer.”

2. Htra1 constructs used: this needs better explanation. It took me a while to go through the cited literature to find out what constructs were used. I believe it is mainly the protease domain-PDZ construct and the protease domain construct. The cartoon in fig.1a represents the full length Htra1 (the mac25 domain should rather be called the IGFBP/Kazal-like domain), rather than any of the constructs used herein. This would be the ideal place for showing the cartoons of the constructs used.

AUTHORS: We thank the reviewer for pointing this out. We now include cartoons of the HTRA1 constructs used for the in vitro work in Fig. 1a and changed the nomenclature of the N-terminus of HTRA1 to IGFBP7 (-like domain).

Fig. 1a: Schematic representation of HTRA1 domain organization and position of selected pathogenic mutations and the active site mutation S328A. Mutations located at the protomer-protomer interface are marked by an asterisk.

In addition, we also included detailed information in the Methods section (page 1, lines 28-33):
“Note that since we and others have not detected any effect of the N-terminus of HTRA1 on oligomerization and protease function^{6,7}, in vitro experiments were conducted with HTRA1 proteins lacking the IGFBP7-like fragment (ΔN) except crystallography, NMR DOSY assays, some data presented in Supp. Fig. 12a and 14a which were obtained using $\Delta N\Delta PDZ$ variants, and data presented in Supp. Fig. 4a which were obtained using culture medium from HEK cells transfected to overexpress full-length HTRA1.”

3. page 3/line70: the sensor loop statement needs references.

AUTHORS: We updated the sentence and included references (see Introduction, page 3, lines 69-70):

“Activation is initiated by an interaction of the sensor loop L3 with the substrate bound to the active site or the PDZ domain^{5,7}.”

4. page 5: what exactly was the rationale for designing the compensatory mutation? The authors invoke the Xray structure (fig.2a) as the rationale, but this structure was done based on the rationale/design, which remains obscure. Please clarify.

AUTHORS: The rationale for choosing D174R as the compensatory mutation was not that we had the Xray structure of D174R-R274Q at hand, but was based on the structure of wt and R274Q, and on mutational screening. To identify a mutation capable to compensate R274Q, we considered residues located in proximity of R274Q, on an adjacent protomer. As shown in Fig. 2a, D174 interacts with R274 in wt HTRA1 (panel 2') and is located at the opposite interface of R274Q (panels 2' and 2''). Hence, we chose this residue for mutational screening. We expressed double-mutant proteins bearing R274Q in *cis* with various D174X mutations and found that replacement of Asp174 by Arg, Lys, or Trp resulted in a substantial gain of protease activity thus indicating functional *cis*-complementation (Supp. Fig. 4a). We focused on D174R for follow up analyses.

To clarify these aspects (i) we labeled the lower panels of Fig. 2a as 2' (wt), 2'' (R274Q) and 2''' (D174R-R274Q), (ii) we now present the results of our mutational screening in Supp. Fig. 4a and (iii) we adapted the text (see Results, page 5, lines 127-132):

“First, we screened for a compensatory mutation at the opposite interface of residue 274. For this, we expressed double-mutant proteins bearing R274Q in cis with various D174X mutations (Supp. Fig. 4a). Replacement of Asp174 by Arg, Lys or Trp resulted in a gain of protease activity thus indicating functional cis-complementation (Fig. 2b-c and Supp. Fig. 4a). In contrast, replacement by His, Ser, Tyr, Gly, or Val had little or no influence on HTRA1 activity.”

5. page 6, line 140: please show the vdW contacts of R174 in figure 2a; from the figure it looks like it contacts a positively charged patch, which cannot be true.

AUTHORS: The reviewer is correct about the positively charged patch making no sense. It has therefore been removed. We have also removed the statement on van der Waals interactions because we feel that the large increase in the surface area is stabilized by multiple interactions that cannot be observed at 3.2 Å resolution (see Results, page 6, lines 146-149):

“Specifically, R174 reoriented towards the protomer at the opposite interface (Fig. 2a, panel 2''') resulting in an overall increased trimer interface as compared to the single R274Q mutant (buried surface area per protomer: wt [PDB ID 3TJO]: 947 Å²; R274Q [PDB ID 6Z0E]: 908 Å²; D174R-R274Q [PDB ID 6Z0X; 6Z0Y]: 925-957 Å²).”

6. page 10: Please define the "inter protomer gap" more precisely. The fig.5a gives no information what we are looking at: please label some key residues. An additional ribbon model of the trimer would help in understanding the location and how this site is related to the compensatory mutant?

AUTHORS: We agree that “**inter protomer gap**” is confusing and adapted the text (see Results, page 10, lines 236-238):

“*To devise an alternative strategy that would target multiple HTRA1 interface mutants, we reasoned that supramolecular ligands exposing positive charges on either end might bind anionic hotspots on adjacent protomers thus stabilizing trimers.*”

Concerning Fig. 5a: we agree with the reviewer and deleted the corresponding panel as it might be misleading concerning the occupancy of GCP ligands. Specifically, it may suggest one bound ligand, while we cannot exclude and even expect the binding of multiple ligands.

7.page 11: the rationale of screening a C-terminal peptide library is totally obscure. I presume it was based on the assumption that these peptides would bind the PDZ domain to allosterically activate htra1? However, they seem to act in the absence of the PDZ domain. This is rather confusing and needs to be clarified.

AUTHORS: To clarify the rationale, we expanded on the text (see Results, page 11, lines 267-273) and inserted a scheme of HTRA1 activation in Fig. 6a:

“*An alternative and independent strategy for rescuing the activity of interface mutants consists of activating monomeric HTRA1 (Fig. 6a). Monomers are proteolytically inactive because the activation domain is incomplete as loop L3 does not reach in from an adjacent protomer. Therefore, we hypothesized that peptides shifting the activation domain into an active conformation might serve as repair factors. This concept was addressed by identifying suitable activating peptides, validation of the monomeric state, determining enzyme kinetics and presenting an initial model of mechanism.*”

Fig. 6a (initially presented in Extended Data): Model of activation of trimeric and monomeric HTRA1. Left panel: in wt HTRA1, the substrate bound to the active site interacts with loop L3, followed by an inter-protomer interaction of loop L3 with loop LD. This interaction mediates the proper positioning of loops L1 and L2, and of the catalytic site. Right panel: in monomeric mutant HTRA1, loop L3-LD interaction is disrupted. The VDAC2 peptide acts as a surrogate of the missing loop L3 from an adjacent protomer leading to the proper positioning of loop LD and thus of loops L1 and L2. Therefore, in active monomeric HTRA1, the activation domain resembles that of classic monomeric serine proteases such as trypsin and chymotrypsin that have no loop L3¹⁴.

8.Fig. 6a: The amounts of VDAC2 and VDAC3 in the Tau cleavage assays are not clear. Also, a detailed description of the Tau cleavage assay in presence of VDAC2/3 or other activators (like the MKs and casein) is missing in the Methods section.

AUTHORS: As suggested, we indicated the amounts of VDAC peptides in the Tau cleavage assays (see legend to Supp. Fig. 12a and Fig. 15d). We also revised and completed the Methods section.

9.Fig. 6b: I believe the labels +/- VDAC2 and +/- crosslinker are missing.

AUTHORS: We now labeled the +/- in the Figure and expanded on the figure legend. The revised panel is presented in Supp. Fig. 3d as shown below:

Supp. Fig. 3d: Oligomeric states of wt and mutant HTRA1 in the absence or presence of 50 μM VDAC2 peptide analyzed by chemical GA-based crosslinking.

10.Fig. 6c: The R274Q mutant is missing the labels for the Y-axis.

AUTHORS: We modified the graph accordingly.

Fig. 6c: Oligomeric states of wt and mutant HTRA1 in the absence or presence of VDAC2 peptide analyzed via NMR DOSY experiments. The slope in the Stejskal-Tanner plots represents the negative diffusion coefficient. Small molecules exhibit a larger diffusion coefficient and thus a steeper slope, while large molecules with a smaller diffusion coefficient show a shallower slope. Inlet: 25 μM R274Q (monomeric), 100 μM R274Q (trimeric) and 25 μM R274Q + 50 μM VDAC2.

11.Material and Methods: Please provide buffers whenever appropriate. No buffers have been listed in the M&M part for purifications, proteolytic cleavage assays or NMR experiments. “2N4R Tau was purified in house as described below”, but there is no purification method shown.

AUTHORS: As suggested, we have revised and completed the Methods section.

References

1. Bultmann-Mellin I, *et al.* Modeling autosomal recessive cutis laxa type 1C in mice reveals distinct functions for Ltbp-4 isoforms. *Dis Model Mech* **8**, 403-415 (2015).
2. Saharinen J, Taipale J, Monni O, Keski-Oja J. Identification and characterization of a new latent transforming growth factor-beta-binding protein, LTBP-4. *J Biol Chem* **273**, 18459-18469 (1998).
3. Meltzer M, *et al.* Allosteric activation of HtrA protease DegP by stress signals during bacterial protein quality control. *Angew Chem Int Ed Engl* **47**, 1332-1334 (2008).
4. Poepsel S, *et al.* Determinants of amyloid fibril degradation by the PDZ protease HTRA1. *Nat Chem Biol* **11**, 862-869 (2015).
5. Rey J, *et al.* An allosteric HTRA1-calpain 2 complex with restricted activation profile. *Proc Natl Acad Sci U S A* **119**, e2113520119 (2022).
6. Eigenbrot C, *et al.* Structural and functional analysis of HtrA1 and its subdomains. *Structure* **20**, 1040-1050 (2012).
7. Truebestein L, *et al.* Substrate-induced remodeling of the active site regulates human HTRA1 activity. *Nat Struct Mol Biol* **18**, 386-388 (2011).
8. Verdura E, *et al.* Heterozygous HTRA1 mutations are associated with autosomal dominant cerebral small vessel disease. *Brain* **138**, 2347-2358 (2015).
9. Uemura M, *et al.* HTRA1 Mutations Identified in Symptomatic Carriers Have the Property of Interfering the Trimer-Dependent Activation Cascade. *Front Neurol* **10**, 693 (2019).
10. Bougea A, *et al.* The first Greek case of heterozygous cerebral autosomal recessive arteriopathy with subcortical infarcts and leukoencephalopathy: An atypical clinico-radiological presentation. *Neuroradiol J* **30**, 583-585 (2017).
11. Ellis MJ, *et al.* Whole-genome analysis informs breast cancer response to aromatase inhibition. *Nature* **486**, 353-360 (2012).
12. Mouradov D, *et al.* Colorectal cancer cell lines are representative models of the main molecular subtypes of primary cancer. *Cancer Res* **74**, 3238-3247 (2014).
13. Vallet C, *et al.* Functional Disruption of the Cancer-Relevant Interaction between Survivin and Histone H3 with a Guanidiniocarbonyl Pyrrole Ligand. *Angew Chem Int Ed Engl* **59**, 5567-5571 (2020).
14. Hedstrom L. Serine protease mechanism and specificity. *Chem Rev* **102**, 4501-4524 (2002).
15. Okiyoneda T, *et al.* Mechanism-based corrector combination restores DeltaF508-CFTR folding and function. *Nat Chem Biol* **9**, 444-454 (2013).
16. Generoso SF, *et al.* Pharmacological folding chaperones act as allosteric ligands of Frizzled4. *Nat Chem Biol* **11**, 280-286 (2015).
17. Veit G, *et al.* Structure-guided combination therapy to potently improve the function of mutant CFTRs. *Nat Med* **24**, 1732-1742 (2018).
18. Wang C, *et al.* Gain of toxic apolipoprotein E4 effects in human iPSC-derived neurons is ameliorated by a small-molecule structure corrector. *Nat Med* **24**, 647-657 (2018).
19. Quancard J, *et al.* An allosteric MALT1 inhibitor is a molecular corrector rescuing function in an immunodeficient patient. *Nat Chem Biol* **15**, 304-313 (2019).
20. Andrei SA, *et al.* Stabilization of protein-protein interactions in drug discovery. *Expert Opin Drug Discov* **12**, 925-940 (2017).
21. Nozaki H, *et al.* Distinct molecular mechanisms of HTRA1 mutants in manifesting heterozygotes with CARASIL. *Neurology* **86**, 1964-1974 (2016).

REVIEWERS' COMMENTS

Reviewer #1 (Remarks to the Author):

The authors have carefully answered almost all my comments one by one by combining multiple approaches. As a result, the data and their explanations are very solid and clear. In addition, the novelty and limitations of this study in its attempts at clinical application are clearly stated.

Reviewer #2 (Remarks to the Author):

The authors addressed my questions mostly satisfactorily. They have also made a sincere effort to address other queries raised by different reviewers. So I am happy with the current manuscript.

Reviewer #3 (Remarks to the Author):

The Authors addressed all my comments, provided additional experimental data and the quality of the manuscript has been improved satisfactorily.

I have one additional remark concerning the Introduction.

Line 71: Activation is initiated by an interaction of the sensor loop L3 with the substrate bound to the active site or the PDZ domain 13,16. Subsequently, loop LD interacts with loop L3, which is provided by an adjacent protomer. This sentence is somewhat misleading.

Please, change to: "Activation is initiated by an interaction of the sensor loop L3 with the substrate bound to the active site or the PDZ domain 13,16. Subsequently, the loop L3 interacts with loop LD of an adjacent protomer. "

Reviewer #4 (Remarks to the Author):

The resubmitted manuscript by Ehrmann, Dichgans and coworkers is significantly improved over the previous version. All of my previous points have been addressed. Although the authors have not succeeded in acquiring quantitative data in all cases, the qualitative or semi-quantitative results are convincing and support the main conclusions.

I believe that the present manuscript is suitable for publication.

Reviewer #6 (Remarks to the Author):

Reviewer #6:

The authors have satisfactorily addressed the comments and suggestions and have made the appropriate changes. I commend them for the now in-depth characterization of the oligomeric state of the HTRA1 mutants in Suppl. Fig.1, and shedding light on the monomer-trimer equilibrium that is even present in the wildtype form; this was well done.

The corrections and the newly added experimental data in this revised version strongly improve the quality of the MS.

Some minor comments for the authors' consideration:

1. Fig.1a cartoon. Thank you for adding this cartoon. Please make the bar sizes the same for all three constructs. Also, the term "IGFBP7" for the N domain is not really correct since IGFBP7 consists of IGFBP, Kazal and an Ig domain, while the Ig domain is not present in the N domain. Therefore, the term IGFBP/Kazal-like would be more precise. However, for simplicity I would just label it N-domain and explain in the legend what it is composed of. Even though the authors explain in the methods for which experiments these different constructs were used, it would be much easier for the reader if the construct used would be indicated in the legends for each figure (HTRA1- Δ N, HTRA1- Δ N Δ PDZ, Full-length HTRA1). This is only a suggestion.

2. paragraph title: "Htra1R274Q mice display cerebrovascular abnormalities": this seems a bit of a stretch, as it implies phenotypic/functional changes, but the changes are only seen at the molecular level. I don't insist on changing, but the authors may want to consider to be more cautious with the title choice.

3. Fig.5: rates of b-casein cleavage: the methods section/fig. legend don't explain how they were determined. Also, based on the figure the rates don't seem linear in all cases, which would make rate calculation challenging.

4. page 11: I would recommend (but I don't insist) to clarify or add a reference on how the authors were able to categorize the peptides as PDZ-binding and non-PDZ binding.

5. Fig.6 Legend: "...such as trypsin and chymotrypsin that have no loop L3.

-the ref.#39 (Hedstrom et al.) is not appropriate here.

-it should be loop 3 or L3, not loop L3.

-the statement is incorrect as other trypsin-like serine proteases do indeed have a L3 (Perona, Craik JBC 272:29987, 1997), which is also termed the "175-loop" (Goettig et al. Biochimie, 166: 52, 2019).

6. p.12/line291; I would replace "as" with ", because"; otherwise it may cause confusion ("as" could be taken as meaning "to be").

Rational correction of pathogenic conformational defects in HTRA1

Rebuttal

REVIEWER 1:

The authors have carefully answered almost all my comments one by one by combining multiple approaches. As a result, the data and their explanations are very solid and clear. In addition, the novelty and limitations of this study in its attempts at clinical application are clearly stated.

AUTHORS: We thank the reviewer for the positive assessment of our manuscript.

REVIEWER 2:

The authors addressed my questions mostly satisfactorily. They have also made a sincere effort to address other queries raised by different reviewers. So I am happy with the current manuscript.

AUTHORS: We thank the reviewer for the positive feedback.

REVIEWER 3:

The Authors addressed all my comments, provided additional experimental data and the quality of the manuscript has been improved satisfactorily.

I have one additional remark concerning the Introduction.

Line 71: Activation is initiated by an interaction of the sensor loop L3 with the substrate bound to the active site or the PDZ domain 13,16. Subsequently, loop LD interacts with loop L3, which is provided by an adjacent protomer. This sentence is somewhat misleading. Please, change to: "Activation is initiated by an interaction of the sensor loop L3 with the substrate bound to the active site or the PDZ domain 13,16. Subsequently, the loop L3 interacts with loop LD of an adjacent protomer. "

AUTHORS: We thank the reviewer for the positive evaluation of the manuscript and revised the text exactly as suggested.

REVIEWER 4:

The resubmitted manuscript by Ehrmann, Dichgans and coworkers is significantly improved over the previous version. All of my previous points have been addressed. Although the authors have not succeeded in acquiring quantitative data in all cases, the qualitative or semi-quantitative results are convincing and support the main conclusions.

I believe that the present manuscript is suitable for publication.

AUTHORS: We thank the reviewer for the positive evaluation of our manuscript.

REVIEWER 6:

The authors have satisfactorily addressed the comments and suggestions and have made the appropriate changes. I commend them for the now in-depth characterization of the oligomeric state of the HTRA1 mutants in Suppl. Fig.1, and shedding light on the monomer-trimer equilibrium that is even present in the wildtype form; this was well done.

The corrections and the newly added experimental data in this revised version strongly improve the quality of the MS.

AUTHORS: We thank the reviewer for the positive feedback and for the constructive suggestions.

Some minor comments for the authors' consideration:

1. Fig.1a cartoon. Thank you for adding this cartoon. Please make the bar sizes the same for all three constructs. Also, the term "IGFBP7" for the N domain is not really correct since IGFBP7 consists of IGFBP, Kazal and an Ig domain, while the Ig domain is not present in the N domain. Therefore, the term IGFBP/Kazal-like would be more precise. However, for simplicity I would just label it N-domain and explain in the legend what it is composed of. Even though the authors explain in the methods for which experiments these different constructs were used, it would be much easier for the reader if the construct used would be indicated in the legends for each figure (HTRA1- Δ N, HTRA1- Δ N Δ PDZ, Full-length HTRA1). This is only a suggestion.

AUTHORS: We adapted the bar sizes and replaced 'IGFBP' by 'N' in the Fig. 1a. We revised the legend as accordingly: *"The N-terminal ('N') domain consists of a fragmented IGFBP7 domain where neither the incomplete IGFBP binds insulin nor the incomplete Kazal-like domain functions as a protease inhibitor"*.

We did not specify the constructs used in each figure legend to avoid distracting the reader with inflating information. As noted by the reviewer, this information is explicitly provided in the Methods section.

2. paragraph title: "Htra1R274Q mice display cerebrovascular abnormalities": this seems a bit of a stretch, as it implies phenotypic/functional changes, but the changes are only seen at the molecular level. I don't insist on changing, but the authors may want to consider to be more cautious with the title choice.

AUTHORS: We agree and revised the title and legend to Fig. 3) to: *"Htra1^{R274Q} mice show an altered cerebrovascular proteome"*.

3. Fig.5: rates of β -casein cleavage: the methods section/fig. legend don't explain how they were determined. Also, based on the figure the rates don't seem linear in all cases, which would make rate calculation challenging.

AUTHORS: We thank the reviewer for bringing this up and revised the Legend to Fig. 5 as follows: *"The rates of β -casein cleavage are presented as the maximum gradient of β -casein degradation after acceleration and before substrate exhaustion."*

4. page 11: I would recommend (but I don't insist) to clarify or add a reference on how the authors were able to categorize the peptides as PDZ-binding and non-PDZ binding.

AUTHORS: The text now reads: *"In contrast, peptides that strongly activate wt HTRA1 by binding to the PDZ domain¹ had no effect on the activity of the R274Q mutant. The same effect was observed for a peptide (CAPN2.1) that activates wt HTRA1 in a PDZ-independent manner¹."*

The rationale is provided in the reference.

5. Fig.6 Legend: "...such as trypsin and chymotrypsin that have no loop L3.

-the ref.#39 (Hedstrom et al.) is not appropriate here.

-it should be loop 3 or L3, not loop L3.

-the statement is incorrect as other trypsin-like serine proteases do indeed have a L3 (Perona, Craik JBC 272:29987, 1997), which is also termed the "175-loop" (Goettig et al. Biochimie, 166: 52, 2019).

AUTHORS: We agree and revised the legend to Fig. 6 as follows: *"Therefore, in active monomeric HTRA1, the activation domain resembles that of classic monomeric serine proteases^{2,3}"* (which excludes the ref. #39, Hedstrom et al).

Concerning the nomenclature of loops, we have in all our structural papers on HtrA proteases named loops as loop L... (2012 Nat Struct Mol Biol, 19:152-157, 2011 Nat Struct Mol Biol, 18:386-388, 2011 J Biol Chem, 286:30680-30690, 2011 Nat Struct Mol Biol, 18:728-731, 2008 Nature 453:885-890, 2007 Genes Dev 21: 2659-2670, 2004 Cell 117:483-494, 2002 Nature 416: 455-459), we therefore did not change throughout the text.

6. p.12/line291; I would replace “as” with “, because”; otherwise it may cause confusion (“as” could be taken as meaning “to be”).

AUTHORS: We revised the text accordingly.

REFERENCES

1. Rey J, *et al.* An allosteric HTRA1-calpain 2 complex with restricted activation profile. *Proc Natl Acad Sci U S A* **119**, e2113520119 (2022).
2. Perona JJ, Craik CS. Evolutionary divergence of substrate specificity within the chymotrypsin-like serine protease fold. *J Biol Chem* **272**, 29987-29990 (1997).
3. Goettig P, Brandstetter H, Magdolen V. Surface loops of trypsin-like serine proteases as determinants of function. *Biochimie* **166**, 52-76 (2019).